# SigmaDock: Untwisting Molecular Docking With Fragment-Based SE(3) Diffusion

**Alvaro Prat**,[*] **Leo Zhang, Charlotte M. Deane, Yee Whye Teh, Garrett M. Morris**
Department of Statistics, University of Oxford

## Abstract

Determining the binding pose of a ligand to a protein, known as molecular docking, is a fundamental task in drug discovery. Generative approaches promise faster, improved, and more diverse pose sampling than physics-based methods, but are often hindered by chemically implausible outputs, poor generalisability, and high computational cost. To address these challenges, we introduce a novel fragmentation scheme, leveraging inductive biases from structural chemistry, to decompose ligands into rigid-body fragments. Building on this decomposition, we present SigmaDock, an $SE(3)$ Riemannian diffusion model that generates poses by learning to reassemble these rigid bodies within the binding pocket. By operating at the level of fragments in $SE(3)$, SigmaDock exploits well-established geometric priors while avoiding overly complex diffusion processes and unstable training dynamics. Experimentally, we show SigmaDock achieves *state-of-the-art* performance, reaching Top-1 success rates (RMSD $< 2$ & PB-valid) above 79.9% on the PoseBusters set, compared to 12.7-32.8% reported by recent deep learning approaches, whilst demonstrating consistent generalisation to unseen proteins. SigmaDock is the *first deep learning approach* to surpass classical physics-based docking under the PB train-test split, marking a significant leap forward in the reliability and feasibility of deep learning for molecular modelling.

## 1 Introduction

The biological function of a protein is determined primarily by its 3D structure and the interactions it mediates. Thus, a central goal of drug discovery is to design small-molecule ligands that bind to a target protein and modulate its function to achieve a therapeutic effect. Since a change in protein function is highly correlated with the bound pose of the binding ligand, a consequence of energetically favourable protein-ligand interactions, the ability predict these structural conformations, which is the primary aim of molecular docking, is essential for reliable and accelerated drug discovery.

Deep learning approaches for molecular docking, in particular, diffusion-based methods, have been recently touted as providing superior accuracy over traditional, industry-standard physics-based methods (Halgren et al., 2004; Morris & Lim-Wilby, 2008; Trott & Olson, 2010). However, the need for such claims to be further qualified has been highlighted by Harris et al. (2023); Buttenschoen et al. (2024), who demonstrated that solely focusing on metrics, such as Root Mean Squared Deviation (RMSD) between the bound and predicted ligand poses, can obfuscate the actual predictive ability of deep learning-based docking tools. For instance, Buttenschoen et al. (2024) showed that when controlling for the chemical plausibility of generated samples, deep learning approaches performed far worse than traditional docking methods. While notable progress on this front has been made with the advent of co-folding models (Abramson et al., 2024; Boitreaud et al., 2024; Wohlwend et al., 2024) such as AlphaFold3 (AF3), which need not assume the protein to be a rigid structure, these models still have key limitations. Firstly, they require massive quantities of data and compute to train, hindering the ability of other researchers to contribute and improve such models. Secondly, as co-folding models jointly model the structure of proteins and ligands, they suffer from slow inference, which makes their practical applicability to drug discovery computationally prohibitive; especially for high-throughput virtual screening (HTVS), where it is often required to query millions of protein-ligand pairs.

---

[*]Correspondence: `alvaro.prat@stats.ox.ac.uk`

To address these issues, we revisit the commonly-used *torsional model* approach (Corso et al., 2022; Huang et al., 2024; Cao et al., 2025) for diffusion-based molecular docking. This approach defines a diffusion process over a ligand's global roto-translations and its torsional angles. Fundamentally, structural (geometric) chemical constraints imply that this low-dimensional manifold forms the principal degrees of freedom underlying any chemically feasible pose. By operating over a space with significantly reduced dimensionality, torsional models promise improved data efficiency, generalisation and faster inference over *all-atom* approaches (favoured by co-folding models). However, this has not been borne out empirically, with relatively disappointing results reported in the literature (Škrinjar et al., 2025).

In this work, we seek to resolve the discrepancy between the poor performance of torsional models and the desired benefits of exploiting inductive biases from structural chemistry. We suspect torsional models underperform because the score model must implicitly account for the mapping from 3D coordinates to torsional updates, a non-local, highly nonlinear, and sometimes ambiguous inverse problem. To bypass this burden, we propose a *fragment model*. We decompose a ligand into *molecular fragments* by breaking rotatable bonds; due to structural chemical constraints, we can treat each fragment's internal geometry as essentially *fixed*. The generative task thereby reduces to predicting an SE(3) rigid transformation for every fragment, from which any chemically feasible pose can be recovered by composing these transformations, obviating explicit modelling of torsional angles. Building on this, we introduce SIGMADOCK, an SE(3) Riemannian diffusion model that defines a diffusion process over the translation and orientation of rigid-body fragments. During sampling, SIGMADOCK iteratively reassembles the ligand's constituent fragments into a predicted bound pose. To reduce the additional degrees of freedom introduced from fragmentation (compared to the torsional model), we make the following novel contributions: (i) a fragmentation scheme that reduces the number of fragments required to represent a ligand; (ii) soft triangulation constraints to provide further inductive biases on the preserved bond lengths and angles across fragments; (iii) an SO(3)-equivariant architecture tailored for reasoning over fragment geometry and protein-ligand interactions.

We adopt the standard re-docking protocol in which the receptor is fixed (*holo-conformation*) and the binding pocket is known. This choice is deliberate for two reasons: first, it is the long-standing setting for benchmarking docking methods, and permits fair 'apples-to-apples' comparison with classical and recent deep learning baselines; second, it reflects industrial HTVS/lead-optimisation practice, where rigid-receptor docking is computationally tractable at scale. Prior generative methods have struggled in precisely this regime, and our aim is to close that gap first.

Empirically, we demonstrate SIGMADOCK surpasses prior deep learning approaches[1] and traditional physics-based docking, achieving *state-of-the-art* performance on the challenging Pose-Busters set (Buttenschoen et al., 2024), and generalising to unseen proteins (Figure 4). We highlight that, with a fraction of the training data, training/sampling time, and lower test-train leakage, we reach AF3-level performance and substantially outperform previous generative methods on the re-docking task. With this, we wish to state our main contribution as the careful and rigorous design of a well-characterised diffusion process, and a detailed construction of structural inductive biases which help learn simpler functions for the task of molecular docking. We highlight that our fragment $\mathrm{SE}(3)^m$ formulation naturally extends to flexible docking by treating selected side chains as additional fragments, and is also adaptable to co-folding, both of which we leave as future work.

## 2 METHOD

### 2.1 NOTATION

**Molecular notation.** The 3D graph of a molecular structure (e.g. ligand, protein or molecular fragment) is defined by the collection $\mathcal{G} = \{\mathbf{x}, \mathbf{v}, \mathbf{b}\}$ where $|\mathcal{G}|$ is the number of atoms in the structure, $\mathbf{x} \in \mathbb{R}^{|\mathcal{G}| \times 3}$ are atomic coordinates, $\mathbf{v} \in \mathbb{R}^{|\mathcal{G}| \times d_a}$ represents the features (e.g. atom type, atomic charge etc.) associated with each atom, and $\mathbf{b} \in \mathbb{R}^{|\mathcal{G}| \times |\mathcal{G}| \times d_b}$ represents the features (e.g. bond type, bond conjugation etc.) associated with each bond. Furthermore, we define the corresponding 2D graph of a molecular structure by $\mathcal{G}^{\mathrm{2D}} = \{\mathbf{v}, \mathbf{b}\}$. For further details on our choice of molecular featurisation see Appendix G.1.

---

[1]Under fair comparison with models trained on the PoseBusters train-test split.

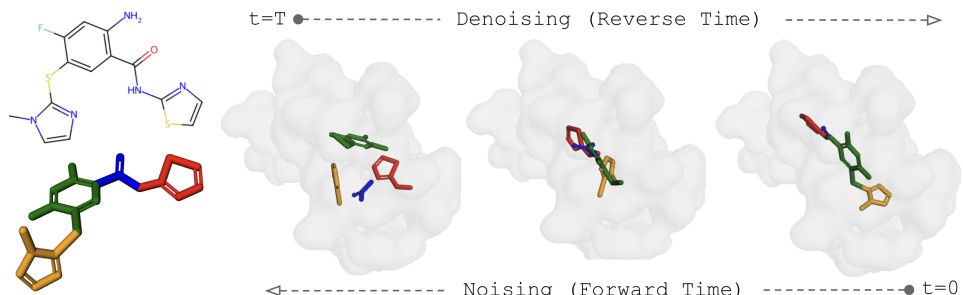

Figure 1: Illustration of SIGMADOCK using PDB 1V4S and ligand MRK. We create an initial conformation of a query ligand where we define our $m$ rigid body fragments (colour coded). The corresponding forward diffusion process operates in $SE(3)^m$ via independent roto-translations.

**SE(3) notation.** Let $\mathbf{x} = [x_1, \ldots, x_N] \in \mathbb{R}^{N \times 3}$ be a collection of points $x_i \in \mathbb{R}^3$ in a rigid-body system where $[\cdot]$ denotes row-wise concatenation. The pose (*i.e.* position and orientation) of $\mathbf{x}$ can be parametrised by elements $(p, R)$ from the Lie group $SE(3)$ via the standard group action:

$$(p, R) \cdot \mathbf{x} = [p + R \cdot x_1, \ldots, p + R \cdot x_N],$$

where $p \in \mathbb{T}(3) \cong \mathbb{R}^3$ is an element of the translation group and $R \in SO(3)$ is a special orthogonal matrix representing a rotation in $\mathbb{R}^3$. For further details about $SE(3)$, see Appendix B.

**Fragment notation.** In this work, we develop a novel fragmentation scheme which allows us to represent the 2D graph of a ligand $\mathcal{G}_{\text{ligand}}^{\text{2D}} = \{\mathbf{v}, \mathbf{b}\}$ in terms of *rigid-body fragments* $\{\mathcal{G}_{F_i}\}_{i=1}^m$ where $\mathcal{G}_F = \{\tilde{\mathbf{x}}_F, \mathbf{v}_F, \mathbf{b}_F\}$ for each $F \in \{F_i\}_{i=1}^m$. In particular, we take $\tilde{\mathbf{x}}_F \in \mathbb{R}^{|\mathcal{G}_F| \times 3}$ to represent the *local coordinates* of the fragment centered at the origin[2], *i.e.* $\frac{1}{|\mathcal{G}_F|} \mathbf{1}^T \tilde{\mathbf{x}}_F = 0$ where $\mathbf{1}$ is a vector of unit entries; at a high level, this can be viewed as the predetermined coordinates of a local, rigid substructure within the ligand due to structural chemical constraints from $\mathcal{G}_{\text{ligand}}^{\text{2D}}$. Hence, the 3D coordinates $\mathbf{x}$ of the ligand can only be constructed from some arrangement of the rigid-body translations and rotations of $\{\tilde{\mathbf{x}}_{F_i}\}_{i=1}^m$. Formally, we identify $\mathbf{z} = (\mathbf{p}, \mathbf{R}) \in SE(3)^m$ with the *global coordinates* $\{\mathbf{x}_{F_i}\}_{i=1}^m$ of the fragments through the usual group action: $\mathbf{x}_{F_i} = (p_{F_i}, R_{F_i}) \cdot \tilde{\mathbf{x}}_{F_i}$ where $\mathbf{p} = (p_{F_1}, \ldots, p_{F_m}) \in \mathbb{T}(3)^m$ and $\mathbf{R} = (R_{F_1}, \ldots, R_{F_m}) \in SO(3)^m$. This allows us to parametrise the pose of the ligand $\mathbf{x}$ in terms of $SE(3)^m$ by $\mathbf{x} = [\mathbf{x}_{F_1}, \ldots, \mathbf{x}_{F_N}]$ and we denote this mapping[3] by $\varphi : SE(3)^m \to \mathbb{R}^{|\mathcal{G}_{\text{ligand}}| \times 3}$.

## 2.2 STRUCTURALLY-AWARE FRAGMENTATION FOR SE(3) DIFFUSION

The task of molecular docking can be summarised as predicting a ligand's bound pose $\mathbf{x} \in \mathbb{R}^{|\mathcal{G}_{\text{ligand}}| \times 3}$ for some query protein, given the 2D graph of the ligand $\mathcal{G}_{\text{ligand}}^{\text{2D}} = \{\mathbf{v}, \mathbf{b}\}$ and the 3D graph of the protein $\mathcal{G}_{\text{protein}} = \{\mathbf{y}, \mathbf{v}_y, \mathbf{b}_y\}$. The guiding idea behind SIGMADOCK is to exploit the inherent structure of a ligand's topology to simplify and condition a smooth and well-defined rigid-body diffusion process in $SE(3)$. In particular, we rely on a novel fragmentation strategy that preserves common stereochemical symmetries, reduces degrees of freedom, and creates a set of geometric priors which help SIGMADOCK learn more general physicochemical correlations. A visual overview of the $SE(3)$ diffusion process described in this section is outlined in Figure 1.

### 2.2.1 THE CONFORMATIONAL MANIFOLD

SIGMADOCK makes use of well-established thermodynamic priors in structural chemistry. Namely, it leverages the fact that the conformational space $\mathbf{x}_c$ defining the local geometry of a ligand (in a vacuum) with known topological structure $\mathcal{G}_{\text{ligand}}^{\text{2D}}$ follows a Boltzmann distribution with probability mass concentrated about a manifold $\mathcal{M}_c$ with holonomic constraints[4] of the form:

$$\mathcal{M}_c = \{\mathbf{x}_c \in \mathbb{R}^{|\mathcal{G}_{\text{ligand}}| \times 3} : g(\mathbf{x}_c) \approx 0\},$$

---

[2]We defer the issue of choosing the orientation of the local coordinates to Section 2.4.

[3]We abuse notation by leaving the dependence on $\{\tilde{\mathbf{x}}_{F_i}\}_{i=1}^m$ from the fragmentation $\{\mathcal{G}_{F_i}\}_{i=1}^m$ implicit.

[4]Holonomic constraints (see (Ryckaert et al., 1977)) restrict the configuration space (position) of a body.

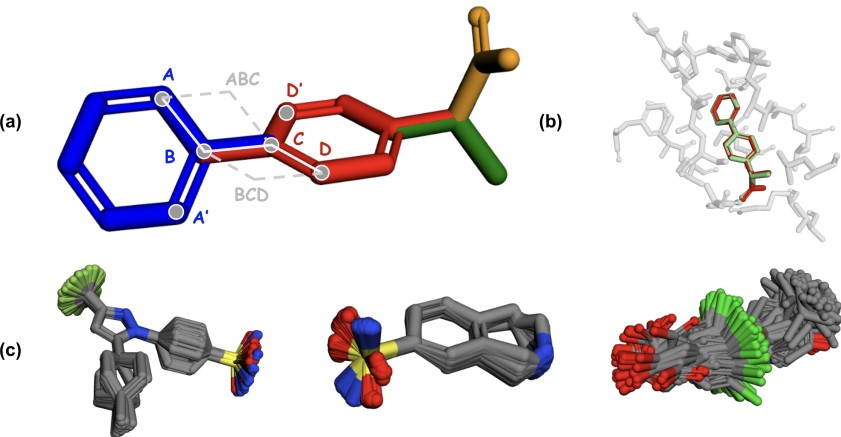

Figure 2: **A**: Illustration of a dihedral $\phi_{ABCD}$ across torsional bond $\overline{BC}$, defined as the angle between planes $\overline{ABC}$ and $\overline{BCD}$, across two adjacent benzene rings in ligand BFL; **B**: Bound (red) and aligned (green) poses for BFL in PDB 1Q4G with an optimised alignment RMSD of 0.11Å; **C**: Conformational ensembles generated from $\pi_{\mathcal{M}_c}$ for ligands SKF, CEL, and IH5 respectively. Notably, the most significant structural changes are derived from torsions across the rotatable bonds.

where $g : \mathbb{R}^{|\mathcal{G}_{\text{ligand}}| \times 3} \to \mathbb{R}_+^m$ maps from Cartesian coordinates to an $m$-vector of scalar holonomic constraints, encoding $m$ independent (soft) boundary conditions. The latter represent locally conserved geometrical priors such as bond lengths ($d_{AB} = d_0$) and bond angles ($\tau_{ABC} = \tau_0$), and exclude dihedral angles/torsions ($\phi_{ABCD} = \phi$); although the distribution $\phi_{ABCD}$ is anisotropic, bodies adjacent to a torsional bond defined over atoms $B, C$ are free to rotate (Figure 2a). Importantly, we can safely assume the holonomic constraints implicit in $\mathcal{M}_c$ can be derived from $\mathcal{G}_{\text{ligand}}^{2D}$ and thermodynamic equilibria. A more formal definition of the form of $g(\cdot)$ is detailed in Appendix D.1. Under this construct, the probability measure supported on $\mathcal{M}_c$ is the constrained Boltzmann measure $\pi_{\mathcal{M}_c}$ which we may sample from: $\mathbf{x}_c \sim \pi_{\mathcal{M}_c}(\mathcal{G}_{\text{ligand}}^{2D})$. As shown in Figure 2c, although this distribution is complex and multimodal, there are abundant preserved symmetries: with RDKit's ETKDGv3 (Landrum (2025)) as our proxy for $\pi_{\mathcal{M}_c}$, we observe that, excluding global rigid motions, the 3D structural differences in $\mathbf{x}_c$ are effectively dominated by changes in dihedrals (torsion).

To faithfully sample rigid body fragments, we first need to justify that samples in $\text{SE}(3)^m$ drawn from the conformational manifold $\mathcal{M}_c$ can be consistently aligned to the bound manifold ($\mathcal{M}_b$), whose distribution $\pi_{\mathcal{M}_b}$ represents the target (data) distribution of bound states. We verify this by aligning ligand conformations from $\mathcal{M}_c$ to $\mathcal{M}_b$ via joint roto–translational and torsional registration. Concretely, let $\mathbf{x}_c \sim \pi_{\mathcal{M}_c}(\cdot)$ denote a conformer on $\mathcal{M}_c$ and let $\boldsymbol{\phi}_c \in \mathbb{T}^k$ represent the corresponding dihedral angles which define the $k$ torsional bonds in $\mathbf{x}_c$. Let $\psi(\boldsymbol{\phi}_c) : \mathbb{T}^k \to \mathbb{R}^{|\mathcal{G}_{\text{ligand}}| \times 3}$ denote the invertible map, where $\mathbb{T}^k = (S^1)^k \cong SO(2)^k$ is the hypertorous defining the space of dihedrals. With $(p, R) \in \text{SE}(3)$ representing global rigid motion, we use the total map

$$\Psi(p, R, \boldsymbol{\phi}) = (p, R) \cdot \psi(\boldsymbol{\phi}) \in \mathbb{R}^{|\mathcal{G}_{\text{ligand}}| \times 3},$$

and perform Kabsch alignment jointly with torsional adjustment by solving

$$\min_{p \in \mathbb{R}^3, \, R \in \text{SO}(3), \, \boldsymbol{\phi} \in \mathbb{T}^k} \text{RMSD}(\mathbf{x}_b, \, \Psi(p, R, \boldsymbol{\phi})), \qquad \mathbf{x}_b \sim \pi_{\mathcal{M}_b}(\mathcal{G}_{\text{ligand}}^{2D})$$

Empirically, we find that the RMSDs between experimentally bound poses $\mathbf{x}_b$ and their corresponding aligned conformers $\mathbf{x}_b' = \Psi(p^\star, R^\star, \boldsymbol{\phi}^\star)$ are substantially below both (i) standard error rates reported by docking baselines (Buttenschoen et al., 2024; Harris et al., 2023), and (ii) the commonly used success threshold of 2Å. This provides sufficient support to claim that the variability in bond lengths and bond angles subsumed in $\mathcal{M}_c$ can be generally ignored in the task protein-ligand docking. Crucially, this allows us to treat bound states as being approximately contained in the set of structures reachable by torsions and SE(3) transforms on conformers drawn from $\pi_{\mathcal{M}_c}$: for any $\mathbf{x}_c \in \mathcal{M}_c$ and $\mathbf{x}_b \in \mathcal{M}_b$, we can align $\mathbf{x}_c$ to $\mathbf{x}_b$ with negligible error such that $\text{RMSD}(\mathbf{x}_b, \mathbf{x}_b') \ll 2$Å. See Figure 2b for an aligned example and Appendix D.3 for additional results and empirical analysis. This empirical inclusion is paramount to our approach as it justifies assembling our stationary

distribution from fragments sampled from $\mathcal{M}_c$, without falling out of distribution. Throughout the remainder of the paper, we will absorb the alignment into the notation by writing $\mathbf{x} \leftarrow \mathbf{x}_b \leftarrow \mathbf{x}_b'$, where we use the aligned conformation $\mathbf{x}_b'$ at the start of forward noising process.

### 2.2.2 Challenges of Torsional Models & Motivation for Our Method

The idea behind directly modelling dihedral angles has been adopted as the standard approach to modelling small molecules (Corso et al., 2022; Jing et al., 2023) and amino acid side-chains in proteins (Jumper et al., 2021). However, formulations that directly model time-dependent dihedrals $\phi_{ABCD}(\mathbf{x}_t, t)$ via torsional updates suffer from fundamental caveats which we aim to resolve in our approach. In Theorem 1 we show that the induced torsional density in Cartesian space is generally not a product distribution, leading to highly entangled implicit dynamics. For further details and a proof of Theorem 1, see Appendix C.2.

**Theorem 1.** *For standard molecular topologies, torsional models define nonlinear mappings from torsion angles to Cartesian coordinates, producing highly entangled, non-product induced measures. In contrast, disjoint rigid fragments yield a factorised product of Haar measures on* $\mathrm{SE}(3)^m$.

Consequently, we argue that diffusing a molecule in fragment space $\mathrm{SE}(3)^m$ offers a simpler learning task than diffusing in torsion space $\mathbb{T}^k \times \mathrm{SE}(3)$. Intuitively, local changes in torsional angles often produce non-local Cartesian displacements, creating strong geometric coupling along torsional chains: a change in a single torsion can substantially displace remote atoms. Therefore, independent torsional perturbations become correlated once mapped to Cartesian coordinates (where the model observes the data) under the induced measure, breaking the product structure. This leads to an ill-conditioned learning problem and stiff sampling dynamics. In contrast, our forward kernel factorises over disjoint $\mathrm{SE}(3)^m$ fragments (product); inter-fragment correlations enter only via the learnt score, rather than being induced by the noise, yielding simpler, better-conditioned mappings, and more stable reverse-time integration.

Furthermore, mapping a torsional increment $\Delta\phi_i$ to a Cartesian displacement $\Delta\mathbf{x}$ is intrinsically ambiguous: one must chose an extrinsic gauge (which side of the torsional bond is rotated, or which combination). Implementations often apply heuristics such as RMSD alignment to remove the net rigid motion caused by torsional updates, which would otherwise break the product-space structure. However, this does not mitigate the ambiguity of the intrinsic to extrinsic mapping, especially when torsional steps are large. Practical solutions often commit an extrinsic realisation (rotate left, rotate right, or a combination), and the model must learn a score consistent with that convention; this choice cannot guarantee consistency during sampling as the selected torsional realisation may not align with the true score direction. Moreover, as $k$ (and thus molecular size and flexibility) increases, torsions produce amplified nonlocal Cartesian displacement (lever effect), coupling distant degrees of freedom. The combinatorial growth of possible extrinsic realisations exacerbates this geometric entanglement, making this framework unscalable. We hypothesise that, in general settings, these issues make torsional frameworks can become poorly conditioned and unnecessarily complex to model. These shortcomings motivate our approach of representing molecules via independent rigid fragments, allowing us to operate over a well defined and geometrically independent product space.

### 2.2.3 Irreducible Fragmentation & Soft Geometric Constraints on $\mathrm{SE}(3)$

The naive choice to define our fragments is to break the molecular graph obtained from $\pi_{\mathcal{M}_c}$ at the torsional bonds, producing a set $\{\mathcal{G}_{F_i}\}_{i=1}^{\hat{m}}$ of torsion-free rigid-body fragments with global coordinates parametrised by $\mathrm{SE}(3)^{\hat{m}}$. This approach yields a set of $\hat{m} = (k+1)$ fragments with a total of $6\hat{m}$ DoFs[5]. In contrast, we note torsional models have $(k+6)$ DoFs ($k$ torsional bonds in $\mathcal{S}^1$ and 6 for rigid body $\mathrm{SE}(3)$). Thus, the natural question arises: *How can we reduce the DoFs of the system and in turn abstract the problem in a general form?* In SigmaDock we tackle this problem by creating a simple yet effective molecular fragmentation reduction (FR3D) that recursively merges adjacent fragments from $\hat{m} = (k+1)$ down to $m$ (Figure 3).

Instead of biasing the fragmentation order, FR3D performs a stochastic search, starting from the torsion-free $\hat{m}$ fragments and branching through candidate neighbour proposing merge actions until reaching an irreducible set of size $m$. Hence, FR3D not only reduces the learnable DoFs by reducing

---

[5]Assuming $|\mathcal{G}_F| > 1$, each fragment is defined by its $\mathbb{T}(3) \cong \mathbb{R}^3$ and $\mathrm{SO}(3)$ parametrisation.

the number of fragments, but also provides a promising stream for data augmentation. Merging is possible in molecular graphs where two or more consecutive torsional bonds are linked, so there are topologies that are irreducible. Hence, we are upper-bounded in the number of fragments: $1 \leq m \leq k+1$. For a fragment hyper-graph with $m$ fragments (and no loop closures), the effective DoFs concentrate between $k+6$ (triangulation-induced lower bound)[6] and $6m$ (unconstrained). Under naïve fragmentation ($\hat{m}=k+1$), this becomes $k+6 \leq \text{DoF} \leq 6\hat{m}$. As FR3D reduces $m$ (empirically we find $m \approx \frac{2}{3}\hat{m}$), the upper bound shrink in practice. We refer the reader to Appendix D.4 to a more extensive analysis and empirical results.

During fragmentation, we retain torsional bond length and angle information by introducing dummy atoms at either side of the bond. Hence, $|\mathcal{G}_{\text{ligand}}| \leq \sum_{i=1}^{m} |\mathcal{G}_{F_i}|$. Importantly, we only retain *free* dummy atoms in $\mathcal{G}_{F_i}$ and otherwise prune dummies which are *over-constrained*. Here, we label a dummy as over-constrained whenever FR3D merges the torsional bond it belongs with a neighbouring fragment, as it naturally over-defines a dihedral angle (Figure 3b). Removing over-constrained atoms is fundamental since an immutable dihedral sampled from $\pi_{\mathcal{M}_c}$ would violate the free torsional requirements outlined in Section 2.2.1, forcing our generator $\pi_{\mathcal{M}_c}$ to yield structures that do not strictly overlap with the bound manifold under optimal alignment: $\Psi \cdot \pi_{\mathcal{M}_c}(\cdot) \neq \pi_{\mathcal{M}_b}(\cdot)$.

We refer the reader to Algorithm 1 in Appendix D.4 for an overview of FR3D and further analysis.

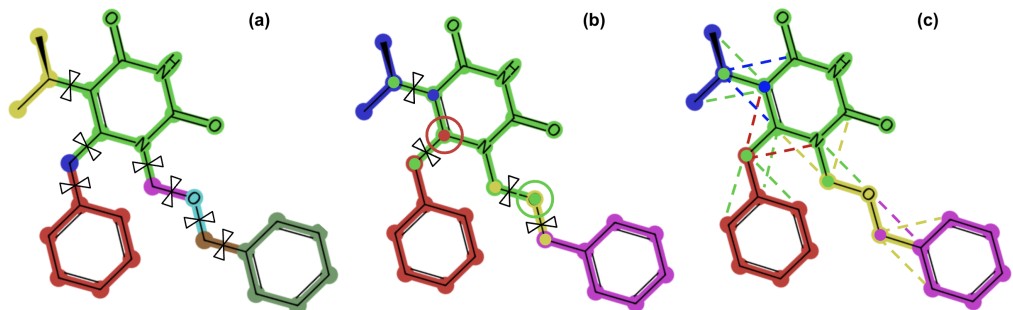

Figure 3: Illustrative example of how FR3D reduces the number of fragments (colour coded) required to represent rigid bodies on ligand TNK into irreducible form. **A**: Defining fragments by snapping all torsional bonds (ribbons); **B**: FR3D recursively attempts to reduce the $k$ torsional bonds and removes over-constrained dummies in the process (denoted by the coloured rings), which otherwise define a dihedral across the merged fragment; **C**; Over-constrained dummies removed and triangulation edges displayed under a different stochastic reduction (equiprobable to solution **b**).

**Soft geometric constraints.** A core ingredient of our method is the inclusion of geometric priors as a mechanism to provide soft (implicit) boundary conditions. Specifically, although FR3D produces irreducible fragments, we define a triangulation distance conditioning scheme which enables pseudo-reductions to the observable DoFs. Concretely, for any torsional bond $\overline{BC}$ connecting adjacent fragments $\mathcal{A}$ and $\mathcal{D}$, we define triangles $(A, B, C)$ and $(B, C, D)$ using neighbouring atoms $A \in \mathcal{A}$ and $D \in \mathcal{D}$ on either side of the set of dihedrals $\phi_{ABCD}$ across $\overline{BC}$. Through Lemma 1, we show that by defining cross-fragment distances $||A - C||$ and $||B - D||$ on top of the rigid fragment template, the corresponding bond angles $\angle(A, B, C)$ and $\angle(B, C, D)$ become uniquely determined. See Figure 3c for an illustration, and for a proof of Lemma 1, see Appendix D.2.

**Lemma 1.** $\forall(A, B) \in (\mathcal{A}, \mathcal{D})$ *bond lengths* $||A - B||, ||B - C||, ||C - D||$ *and bond angles* $\angle(A, B, C), \angle(B, C, D)$ *are fully determined with triangulation conditioning, without restricting changes in the dihedral angles* $\Delta\phi_{ABCD}$.

## 2.3 SE(3) DIFFUSION

From our identification of ligand poses $\mathbf{x}$ with $\mathbf{z} = (\mathbf{p}, \mathbf{R}) \in \text{SE}(3)^m$ via the fragmentation $\{\mathcal{G}_{F_i}\}_{i=1}^{m}$, we adopt the SE(3) diffusion model framework introduced in Yim et al. (2023) to construct a generative model $p_\theta(\mathbf{z}|\mathcal{G}_{\text{dock}})$ for sampling the docked pose of some ligand,

---

[6]The resulting DoFs are lower-bounded because the triangulation scheme imparts soft boundary constraints; it provides a strong signal for SIGMADOCK to reduce $\Delta d_{A,C}$ to 0 as $t \to 0$.

given its 2D graph $\mathcal{G}_{\text{ligand}}^{\text{2D}}$, its fragmentation $\{\mathcal{G}_{F_i}\}_{i=1}^m$ and some query protein $\mathcal{G}_{\text{protein}}$; we use $\mathcal{G}_{\text{dock}} = (\mathcal{G}_{\text{ligand}}^{\text{2D}}, \{\mathcal{G}_{F_i}\}_{i=1}^m, \mathcal{G}_{\text{protein}})$ to denote this conditioning information. We provide an overview of this framework below, and for further details, see Appendix C.

**Forward process.** For each protein-ligand pair $(\mathcal{G}_{\text{ligand}}, \mathcal{G}_{\text{protein}})$ in our dataset $p_{\text{data}}$, with an associated fragmentation $\{\mathcal{G}_{F_i}\}_{i=1}^m$, we define the forward process $(\mathbf{Z}^{(t)})_{t \in [0,T]} = ((\mathbf{p}^{(t)}, \mathbf{R}^{(t)}))_{t \in [0,T]}$ with $\mathbf{Z}^{(t)} \sim p_t(\mathbf{z}|\mathcal{G}_{\text{dock}})$ via the SDE:

$$d\mathbf{Z}^{(t)} = \left[-\tfrac{1}{2}\mathbf{p}^{(t)}, 0\right] dt + \left[d\mathbf{B}_{\mathbb{R}^{m \times 3}}^{(t)}, d\mathbf{B}_{\text{SO}(3)^m}^{(t)}\right], \tag{1}$$

where $\mathbf{B}_{\mathbb{R}^{m \times 3}}^{(t)}, \mathbf{B}_{\text{SO}(3)^m}^{(t)}$ denotes Brownian motion on $\mathbb{R}^{m \times 3}$ and $\text{SO}(3)^m$ respectively, with the initial condition $\mathbf{Z}^{(0)} = (\mathbf{p}^{(0)}, \mathbf{R}^{(0)}) = \varphi^{-1}(\mathbf{x})$ where $\mathcal{G}_{\text{ligand}} = \{\mathbf{x}, \mathbf{v}, \mathbf{b}\}$ contains the ground-truth docked pose $\mathbf{x}$. We note that the SDE is designed for the forward kernel $p_{t|0}(\mathbf{z}^{(t)}|\mathbf{z}^{(0)})$ to be tractably sampled from, and we take $T > 0$ large enough for $p_T$ to be close to the stationary distribution $q(\mathbf{z}) = \mathcal{N}(\mathbf{p}; 0, \mathbf{I}) \otimes \mathcal{U}_{\text{SO}(3)^m}(\mathbf{R})$, where $\mathcal{U}_{\text{SO}(3)^m}$ denotes the uniform distribution on $\text{SO}(3)^m$.

**Backward process.** The associated backward process $(\overleftarrow{\mathbf{Z}}^{(t)})_{t \in [0,T]}$ is then given by the SDE:

$$d\overleftarrow{\mathbf{Z}}^{(t)} = \left[\tfrac{1}{2}\overleftarrow{\mathbf{p}}^{(t)} + \nabla_p \log p_{T-t}(\overleftarrow{\mathbf{Z}}^{(t)}|\mathcal{G}_{\text{dock}}), \nabla_R \log p_{T-t}(\overleftarrow{\mathbf{Z}}^{(t)}|\mathcal{G}_{\text{dock}})\right] dt + \left[d\mathbf{B}_{\mathbb{R}^{m \times 3}}^{(t)}, d\mathbf{B}_{\text{SO}(3)^m}^{(t)}\right], \tag{2}$$

where $\nabla_z \log p_t(\mathbf{z}|\mathcal{G}_{\text{dock}}) = [\nabla_p \log p_t(\mathbf{z}|\mathcal{G}_{\text{dock}}), \nabla_R \log p_t(\mathbf{z}|\mathcal{G}_{\text{dock}})]$ denotes the score function of the induced probability path $p_t$; we note that this should be understood as a Riemannian gradient which lives in the tangent space $\text{Tan}_{\mathbf{z}} \text{SE}(3)^m$.

**Training and sampling.** We see that we can generate bound poses under $\mathcal{G}_{\text{dock}}$ from simulating the backward SDE in Equation 2, however, the true score function $\nabla_z \log p_t$ is intractable. Therefore, we train a neural network approximation $s_\theta(\mathbf{z}, t, \mathcal{G}_{\text{dock}})$ via the score matching objective:

$$\mathcal{L}(\theta) = \mathbb{E}_{p(t), p_{\text{data}}(\mathcal{G}_{\text{ligand}}, \mathcal{G}_{\text{protein}}), p_{t|0}(\mathbf{Z}^{(t)}|\mathbf{Z}^{(0)})} \left[\left\|s_\theta(\mathbf{Z}^{(t)}, t, \mathcal{G}_{\text{dock}}) - \nabla_z \log p_{t|0}(\mathbf{Z}^{(t)}|\mathbf{Z}^{(0)})\right\|_{\text{SE}(3)^m}^2\right]. \tag{3}$$

Hence, we denote $p_\theta(\mathbf{z}|\mathcal{G}_{\text{dock}})$ as the distribution of generated samples, from first sampling $\overleftarrow{\mathbf{Z}}^{(0)} \sim q$ and then simulating the backward SDE with our learnt approximation $s_\theta$, which approximates the true distribution $p_0(\mathbf{z}|\mathcal{G}_{\text{dock}})$. The corresponding 3D coordinates $\hat{\mathbf{x}}$ of samples $\hat{\mathbf{z}} \sim p_\theta(\mathbf{z}|\mathcal{G}_{\text{dock}})$ can then be recovered by the mapping $\hat{\mathbf{x}} = \varphi(\hat{\mathbf{z}})$.

## 2.4 ARCHITECTURE

A significant contribution of SIGMADOCK is the design of our architecture $s_\theta(\mathbf{z}, t, \mathcal{G}_{\text{dock}})$ which parametrises the score function. In particular, we augment EquiformerV2 (Liao et al., 2023) to handle protein-ligand (and other molecular) diffusion; we use this as the backbone for our model to ensure $\text{SO}(3)$-equivariance[7]. Our main innovations are: (i) we augment the input graph with virtual nodes and edges on top of the original chemical graph $\mathcal{G}_{\text{dock}}$, creating a hierarchical topology. This reduces risk of over-squashing by reducing the average node degree, whilst promoting global information flow and mitigating over-smoothing: less layers needed to pass global information; (ii) we tailor our featurisations of nodes and edges according to their structural role; (iii) we ensure messages and gradients along the edges, which represent local interactions (present on proximity), smoothly decay to zero as the distance between the neighbouring nodes approaches some cutoff; this prevents instabilities from sudden changes in the input graph's topology as we perturb $\mathbf{z}$.

Moreover, we note that a critical issue for the design of our architecture is that the parametrisation of the global coordinates $\mathbf{x}_F$ in terms of $(p, R) \in \text{SE}(3)$ is *not uniquely* defined. This is due to the fact that we do not have a canonical choice for the orientation of the local coordinates $\tilde{\mathbf{x}}_F$. For instance, we have the equally valid choices $\tilde{\mathbf{x}}_F, \tilde{\mathbf{x}}_F'$ for the local coordinates of $\mathcal{G}_F$ if $\tilde{\mathbf{x}}_F' = R_0 \cdot \tilde{\mathbf{x}}_F$ where $R_0 \in \text{SO}(3)$. Hence, we can have two different representations of global coordinates $\mathbf{x}_F$ from $\mathbf{x}_F =$

---

[7]EquiformerV2 is also translation invariant but we do not require this property since our problem setting has a canonical centre of mass given by the binding pocket.

$(p, R) \cdot \tilde{\mathbf{x}}_F = (p, R R_0^{-1}) \cdot \tilde{\mathbf{x}}'_F$ depending on the initial choice of orientation. To resolve this issue, we adapt the SO(3)-equivariant prediction head introduced in Jin et al. (2023), based on the Newton-Euler equations from rigid-body mechanics, to which we pass the outputs of our backbone model into. Particularly, we predict pseudo-forces for all atoms pertaining to the $m$ fragments and use these as a basis to define our scores in the tangent space of SE(3) (more details in Appendix G.4). With this choice, Theorem 2 shows that SIGMADOCK is invariant to the choice of local coordinate axes.

**Theorem 2.** *Our training objective and sampling procedure are invariant with respect to the choice of orientations for local coordinates. Moreover, our score model is* SO(3)-*equivariant which ensures* $p_\theta(\mathbf{z}|\mathcal{G}_{dock})$ *is a stochastically* SO(3)-*equivariant kernel.*

**Conditioning.** We define the triangulation conditioning by feeding the relative distance mismatch as an edge feature (compact notation): $\Delta d_{A,C}(\mathbf{x}_t, t|\mathcal{G}_{\text{dock}}) = ||A(t) - C(t)|| - d^{\text{ref}}_{A,C}$, (with $d^{\text{ref}}_{A,C}$ defined from the initial RDKit conformer), such that $\lim_{t \to 0} \Delta d_{A,C}(\mathbf{x}_t, t|\mathcal{G}_{\text{dock}}) = 0$ across all cross-fragment triangulation edges. Dummy atoms which define triangular geometry are discarded after sampling; torsional bonds are reconstructed from anchors so there is no discrepancy between conditioning inputs and the final conformation. With this conditioning, only dihedral angles and rigid body roto-translations remain free as $t \to 0$.

For further architectural details, see Appendix G, and for a proof of Theorem 2, see Appendix H.1.

## 2.5 Training and Inference

We outline our training setup for SIGMADOCK in Appendix E. In particular, we discuss how we preprocess our data for fragmentation and training, as well as computational tricks for increasing training throughput. We detail the definition of the binding pocket in Appendix E.1. Briefly, the pocket includes all residues with any atom within a stochastic cutoff $d_r$ of any ligand atom. Formally, $d_r := d_0 + \mathcal{N}(0, \sigma_r)$, where $d_0$ and $\sigma_r$ default to 5Å and 1Å respectively during training. Our sampling procedure is outlined in Appendix F, where we discuss the fact that, due to the reliability of SIGMADOCK in generating chemically plausible samples, SIGMADOCK *does not* require the use of a separately trained confidence model to filter out poor generations. Instead, we propose using the simple and cheap heuristic of evaluating both the (pseudo) binding energy of the generated protein-ligand system, as well as a set of physicochemical checks (such as, bond angles, bond lengths, internal energy) to rank our $N_{\text{seeds}}$ samples for evaluation.

## 3 Experiments

### 3.1 Data and Metrics

**Datasets.** We use PDBBind(v2020) (Wang et al., 2005), a curated set of 19,443 protein-ligand complexes obtained through crystallography, as our training set. Crucially, we deliberately restrict ourselves to this dataset for fair comparison[8], isolating any increase in performance obtained in this study to our proposed framework. For validation, we use the well established PoseBusters (Buttenschoen et al., 2024) and Astex (Hartshorn et al., 2007) datasets. PoseBusters(v2) (PB) acts as our temporal-split validation set containing 308 protein-ligand complexes with unseen protein sequences realised from 2021 onwards. The Astex (AX) dataset consists of an additional 85 diverse and highly curated protein-ligand complexes originally designed to faithfully evaluate the quality of protein-ligand docking algorithms.

**Metrics.** We evaluate a generated pose $\hat{\mathbf{x}}$ by measuring the symmetry-corrected RMSD (Meli & Biggin, 2020) between the crystallographic (bound) pose $\mathbf{x}$ and the generated pose $\hat{\mathbf{x}}$ obtained by the mapping $\hat{\mathbf{x}} = \varphi(\hat{\mathbf{z}})$ where $\hat{\mathbf{z}} \sim p_\theta(\mathbf{z}|\mathcal{G}_{\text{dock}})$. To account for sampling variability, either from different conformers (inducing differences in fragment local coordinates), or by resampling $\mathbf{z}^{(1)} \sim q(\mathbf{z})$, we report the Top-$k$ success rate, i.e., the fraction of complexes where at least one of the top $k$ poses (from $N_{\text{seeds}}$ samples) has RMSD $< 2$Å. We also use PoseBuster to assess PB-validity, indicating whether generated structures also satisfy standard physicochemical tolerances.

---

[8]It is unfair yet unfortunately common in the literature to compare the same held out set with models trained with larger and more diverse datasets without assessing train-test overlap (Abramson et al., 2024).

## 3.2 RESULTS

Using the sampling algorithm described in Appendix F, we benchmark the base performance of SIGMADOCK and present our main results in Figure 4. To the best of our knowledge, SIGMADOCK is the first deep learning-based method to surpass classical physics-based approaches in the PB and AX sets using the intended train-test split on the re-docking task[9]. Not only does SIGMADOCK achieve a $6.3\times$ higher PB-validity than DiffDock, the best open-source alternative tested on the same split, but it also excels on proteins with low sequence similarity, overcoming the common critique that deep learning models memorise rather than learn physics. Notably, Corso et al. (2024) train DiffDock-L on a significantly larger corpus (PDBBind(v2020) ∪ BindingMOAD) and report 50% Top-1 (RMSD only) on PB, whereas SIGMADOCK attains a Top-1 (RMSD & PB validity) of 79.9%. Overall, these results support our main contribution: a theoretically-grounded $\mathrm{SE}(3)^m$ Riemannian diffusion framework with strong generalisation. Conversely, torsional-space baselines and point-cloud docking models evaluated under identical conditions do not attain comparable results.

We also highlight SIGMADOCK does *not* require minimisation to achieve high PB validity, a common yet computationally expensive hack used to artificially improve deep learning methods. Notably, we achieve AF3-level performance (Top-1 of 84%: see Extended Data Fig. 4e in Abramson et al. (2024)) with just 19k training data-points, *significantly lower train-test leakage* (see App. J), and $50\times$ faster sampling. Together with an outstanding performance in the AX set, reaching near-perfect Top-1 (above 90%), we believe these results mark a *major leap forward* in the feasibility and reliability of deep learning for molecular modelling.

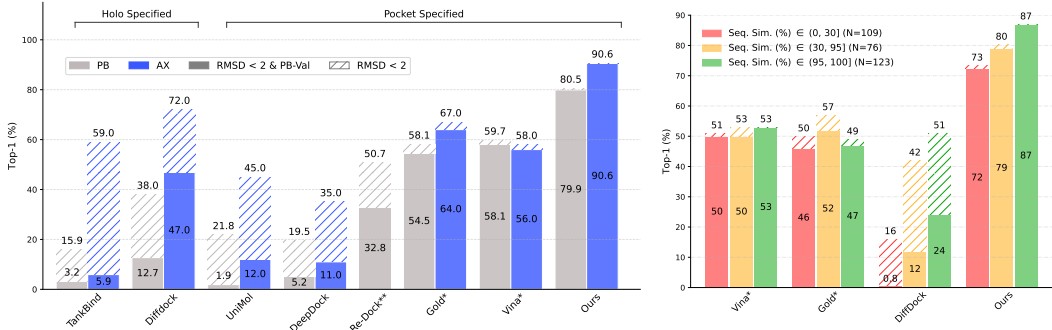

Figure 4: Performance benchmarks. *Left*: Comparative performance of SIGMADOCK on the PB and AX diverse sets against prior methods. Extracted from Abramson et al. (2024); Buttenschoen et al. (2024). (*) Denotes classical docking; (**) Are not open-sourced. *Right*: Performance breakdown across sequence similarity splits in the PB set.

**Ablations.** To better characterise and highlight the contribution of some key components in our method, we perform an ablation study covering a set of training-time and test-time variables (see Table 1). Namely, we report the influence of our fragmentation merging strategy and triangulation conditioning, as well as the effect of including protein-ligand interactions as part of the computational graph, and give empirical evidence of their relevance (4-12% relative improvement). In addition, we show how sampling fragments from $\mathcal{M}_c$ vs. $\mathcal{M}_b$ leads to a small but expected decrease in sample quality. By excluding PB-checks in the heuristic, SIGMADOCK maintains a high PB-valid Top-1. Finally, we show how increasing $N_{\text{seeds}}$ improves performance at the expense of more computational overhead, and highlight the importance of our simple yet effective heuristic for ranking our samples and picking our best candidate(s).

On top of sequence similarity, we stratify the PB set into distinct chemical environments determined by the nature of ligand interactions with additional co-factors (ions, crystallisation aids, natural ligands, or other co-factors). We hypothesise that, since SIGMADOCK is deliberately designed (for simplicity) to exclude co-factors, higher failure rates should be observed when the true bound pose is realised in conjunction to additional artefacts (co-binding event), as the setup is partially observable.

---

[9]For fairness, we compare our method in the main body against models trained on the same train-test split. We do not include DiffDock-L or AF3 in Figure 4 under fair scientific practice.

After isolating the protein-ligand pairs for which SIGMADOCK fails to generate accurate poses[10], we find this hypothesis to hold true, as per Table 2. This result provides additional confidence that SIGMADOCK does not blindly memorise and hallucinate protein-ligand poses.

Table 1: Ablation results (Top-1 accuracy (%) across the PB set) for different configurations. **A-C** are re-trained from scratch; (*): default.

| Conf. | Description | RMSD $< 2$ | PB Val. |
|---|---|---|---|
| **A** | $(-)$ Tri. Cond. | 71.9 | 67.1 |
| **B** | $(-)$ PL Interactions | 79.2 | 76.3 |
| **C** | $(-)$ Frag. Merging | 74.4 | 73.7 |
| **G** | Sampling from $\mathcal{M}_b$ | 86.4 | 85.4 |
| **D** | $(-)$ Energy Scoring | 67.2 | 66.1 |
| **E** | $(-)$ PB Scoring | **82.1** | 70.8 |
| **H** | SIGMADOCK ($N_{seeds} = 10$) | 74.7 | 72.2 |
| **I*** | SIGMADOCK ($N_{seeds} = 40$) | 80.5 | **79.9** |

Table 2: Performance analysis (Top-1 accuracy (%)) across PB subsets according to the presence of various co-factors. The subset size is shown next to the co-factor species key. The failure rate represents the sample failure rate, averaged across 40 seeds, for all complexes in the subset.

| Co-factor Presence | RMSD $< 2$ | PB Val. | Fail. Rate |
|---|---|---|---|
| Natural Ligands (17) | 58.8 | 58.8 | 41.2 |
| Ions (57) | 75.4 | 75.4 | 23.6 |
| Other (60) | 76.7 | 76.7 | 28.1 |
| Crystallisation Aids (37) | 81.1 | 81.1 | 35.0 |
| **None (165)** | **84.2** | **83.0** | **16.2** |

Docking tools are typically provided with a search region (e.g. bounding box or centre-radius) defining the pocket search space. To assess robustness to larger pockets, corresponding greater uncertainty of the binding region, we sweep the deterministic cutoff $d_0$, whilst reducing the jitter $\sigma_r \to 0$. As presented in Table 3, SIGMADOCK remains robust to larger pockets, with a moderate drop when operating outside the training support (e.g. $\varnothing_{pocket} = 7$Å is $2\sigma$ relative to the training mean $d_0 = 5, \sigma_r = 1$). Notably, reducing the pocket size (`--autobox` + 5Å) does not improve Vina's Top-1 (57.2% vs. 56.0%), indicating that gains over classical methods are not attributable to smaller pocket definitions. Although we cannot directly compare SIGMADOCK to co-folding methods, we show competitive performance relative to AF3 with a fraction of the training data and lower test-train leakage (Table 4). We leave a more detailed comparison in Appendix J.2.

Table 3: Sensitivity to pocket definition (PB set). Pocket diameter $\varnothing_{pocket}$ is the maximum pairwise distance between $C_\alpha$'s across the selected $N_{res}$ residues; we report dataset means.

| Metric $\|| d_0$(Å) | **4** | **5** | **6** | **7** |
|---|---|---|---|---|
| $\varnothing_{pocket}$ (Å) | 20.9 | 22.4 | 24.2 | 26.2 |
| $N_{res}$ | 15.6 | 20.3 | 26.5 | 37.8 |
| RMSD $< 2$ | 80.5 | **81.5** | 78.3 | 69.8 |
| PB-Val. | 80.2 | **80.5** | 77.3 | 68.2 |

Table 4: Per-sequence-similarity comparison between SIGMADOCK (left) and AF3 (right) on the PB set. Values for AF3 are extracted from Extended Data 4c (Abramson et al., 2024).

| Seq. Sim. (%) | Count | PB-Val. |
|---|---|---|
| $[0, 30)$ | 109 \| 38 | 72 \| 87 |
| $[30, 95)$ | 76 \| 83 | 79 \| 82 |
| $[95, 100]$ | 123 \| 187 | 87 \| 78 |
| Total / Avg. | 308 \| 308 | 79.9 \| 80.2 |

We refer the reader to Appendix I for extended results, and Appendix J for a detailed discussion on the current limitations of our method and future work.

## 4 CONCLUSION

We believe SIGMADOCK represents a major step forward in the reliability and feasibility of deep learning as a promising tool for accelerating drug discovery. Moving away from torsional parametrisation, our proposed $SE(3)^m$ fragment-space Riemannian diffusion model is, to our knowledge, the first generative method trained on the intended PB split to surpass classical dockers on the redocking task. We extensively lay out the key components of our framework in the Appendices and open-source our codebase to proliferate reproducibility and further development, and we view extensions to flexible-receptor docking and co-folding as natural next steps to make SIGMADOCK a more general and practical tool. We demonstrate the critical role of principled inductive biases in enabling superior generalisation and data efficiency, and hope our work encourages rethinking progress on geometry and conditioning, as opposed to relying on scale alone (Abramson et al., 2024).

---

[10]Here we define a failure if the majority of samples generated across $N$ seeds have an RMSD above 2Å.

AUTHOR CONTRIBUTIONS

As the main contributor, A.P. instigated and led the formulation, development, and analysis of SIG-MADOCK. A.P and L.Z. both developed the mathematical framework, proofs, and the key components of SIGMADOCK. A.P and L.Z. organised and wrote the manuscript. G.M.M, C.M.D and Y.W.T supervised the project, partaking in relevant scientific discussions and proof-reading the manuscript.

ACKNOWLEDGMENTS

AP is funded by EPSRC and AstraZeneca via an iCASE award for a DPhil in Machine Learning. LZ is supported by the EPSRC CDT in Modern Statistics and Statistical Machine Learning (EP/S023151/1). AP thanks Kathryn Giblin for insightful scientific discussions, as well as Jochem Nelen and other PhD students in the OPIG and OxCSML groups for valuable exchanges. AP also thanks Yasmin Baba for her unwavering support and for compelling him to take a rare break from research, during which the core idea of this work emerged. LZ thanks Jessica Harrison for helpful witticisms in informing (subliminally) our work's moniker. The authors declare no competing interests.

REPRODUCIBILITY STATEMENT & CODE AVAILABILITY

All training and evaluation data used in this study are publicly available. The full codebase for training, sampling, and evaluation used to generate the reported results will be released open source upon publication at github.com/alvaroprat97/sigmadock.

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

# APPENDICES

# A    RELATED WORK

**Molecular docking.**    Traditional approaches to molecular docking rely on the combination of assessing ligand poses with hand-crafted physics-based scoring functions and search-based optimisation (Halgren et al., 2004; Morris & Lim-Wilby, 2008; Trott & Olson, 2010). Despite the widespread use of such tools, these approaches have relatively slow run times and lower than desired accuracy. There is growing interest in the application of deep learning-based methods to address these shortcomings. These can be broadly organised into learning scoring functions (Prat et al., 2024; 2023; Méndez-Lucio et al., 2021; Zhou et al., 2023a), regression-based prediction (Stärk et al., 2022; Lu et al., 2022) and generative modelling of docked poses. The latter can be roughly grouped into methods which condition on pocket-specific interactions (Plainer et al., 2023) (as opposed to blind docking), restricts the generative dynamics to the degrees of freedom spanned by global transformations and updates to torsional angles (Corso et al., 2022; Cao et al., 2025), allow for the unconstrained atom-level generation of poses (Stärk et al., 2023), and the flexible modelling of protein side chains (Huang et al., 2024). Closely related are co-folding models (Abramson et al., 2024; Boitreaud et al., 2024; Wohlwend et al., 2024) which aim to jointly generate the 3D structure of a protein-ligand complex for docking, as opposed to only modelling the ligand. Despite the impressive results claimed over traditional docking tools, as measured in terms of RMSD, Buttenschoen et al. (2024) demonstrated that when controlling for the chemical plausibility of generated samples, performance falls below even traditional approaches. We note that co-folding models have reported good performance on the PB benchmark, which are higher than traditional docking methods, used by Buttenschoen et al. (2024), but the comparison with the specialised deep learning-based docking models mentioned above is not exactly fair, due to the large gap in data and compute used to train co-folding models, which also results in substantially longer inference times. SIGMADOCK improves over previous methods in that it is able to quickly and reliably generate poses with high chemical plausibility and accuracy—even without the need for a separate neural network-based confidence model to filter out poor quality samples.

$SE(3)$ **diffusion.**    Euclidean diffusion models (Ho et al., 2020; Song et al., 2020) construct a generative model of data by learning the time reversal of some fixed Gaussian noising process. This class of models have been generalised to data residing on non-Euclidean manifolds (Leach et al., 2022; Huang et al., 2022; De Bortoli et al., 2022), in particular the Lie group $SE(3)$ (Yim et al., 2023) which is commonly used to describe the position and orientation of rigid-body systems. By formulating the generative dynamics for a given ligand over the position and orientation of its fragments in $SE(3)$, SIGMADOCK avoids the complexity and poor training dynamics of the torsional model approach popularised by Corso et al. (2022) for generative molecular docking.

**Fragment-based models.**    Working with molecules in terms of their constituent molecular fragments is a popular approach due to the close relation between the properties and geometry of a molecule and the composition of their fragments. Examples include lead discovery for drug development (Xu & Kang, 2025), modelling proteins in terms of rigid backbone frames (Jumper et al., 2021; Yim et al., 2023; Watson et al., 2023), generating synthesizable molecules in terms of building blocks and their reaction pathways (Koziarski et al., 2024; Gao et al., 2024), and linker design (Guan et al., 2023). Despite the aforementioned popularity of fragmentation, to the best of our knowledge, SIGMADOCK is the first work to use a fragment-based framework for generative molecular docking.

# B    FURTHER DETAILS ON SE(3)

In this section, we provide an extended discussion on SE(3) and Lie group theory.

**Group structure of** SE(3). Let $\mathbb{T}(3)$ be the translation group in $\mathbb{R}^3$ with the addition operation (it is trivial to see that $\mathbb{T}(3)$ is isomorphic to $\mathbb{R}^3$ with vector addition) and let SO(3) be the group of special orthogonal matrices in $\mathbb{R}^{3\times3}$. The group SE(3) is defined by the following semi-direct product: $SE(3) := \mathbb{T}(3) \rtimes SO(3)$ where elements from SE(3) can be written as $(p, R) \in \mathbb{T}(3) \times SO(3)$. We recall the following properties of SE(3):

- The identity element $e \in SE(3)$ is defined as $e = (0, \mathbf{I})$;
- The group operation is defined by $(p, R) \cdot (p', R') = (p + Rp', RR')$;
- For any $(p, R) \in SE(3)$, its inverse is defined by $(p, R)^{-1} = (-R^{-1}p, R^{-1})$.

Furthermore, we define $SE(3)^m := \prod_{i=1}^m SE(3)$ as a product group.

**Lie groups.** A Lie group is a smooth manifold $G$ equipped with a group structure, such that the group operation $\cdot : G \times G \to G$ and the inverse operation $(\cdot)^{-1} : G \to G$ are both smooth maps, considered under the smooth structure of $G$. Common examples of Lie groups include $\mathbb{T}(3), SO(3)$ and SE(3). Additionally, for each Lie group $G$, we have its associated Lie algebra $\mathfrak{g}$ defined as the tangent space $\mathrm{Tan}_e G$ of $G$ at the identity element $e \in G$[11]. For a mathematical introduction to smooth manifolds and Lie groups, we recommand Lee (2003).

## B.1    FURTHER DETAILS ON SO(3)

We provide a further discussion on the Lie group SO(3) which is relevant for our construction of a SE(3) diffusion model.

**Lie algebra basis.** The Lie algebra $\mathfrak{so}(3)$ of the Lie group SO(3) has a canonical basis described by the following skew-symmetric matrices:

$$\mathbf{e}_1 = \begin{pmatrix} 0 & 0 & 0 \\ 0 & 0 & -1 \\ 0 & 1 & 0 \end{pmatrix}, \qquad \mathbf{e}_2 = \begin{pmatrix} 0 & 0 & 1 \\ 0 & 0 & 0 \\ -1 & 0 & 0 \end{pmatrix}, \qquad \mathbf{e}_3 = \begin{pmatrix} 0 & -1 & 0 \\ 1 & 0 & 0 \\ 0 & 0 & 0 \end{pmatrix}.$$

We use $[\cdot]_{\mathbb{R}^3} : \mathfrak{so}(3) \to \mathbb{R}^3$ to denote the standard coordinate representation map which is a linear isomorphism (*i.e.* maps the canonical basis to the canonical basis of $\mathbb{R}^3$), and we denote $[\cdot]_\times : \mathbb{R}^3 \to \mathfrak{so}(3)$ as its inverse.

The above basis can be derived from differentiating $A(t)^\top A(t) = \mathbf{I}$ where $t \mapsto A(t)$ is any smooth curve in SO(3) such that $A(0) = \mathbf{I}$, and using the fact that SO(3) is an embedded submanifold of $\mathbb{R}^9$ (by the regular value theorem) so that the standard Euclidean derivative $A'(t)$ of $A(t)$ can be considered as an element in $\mathrm{Tan}_{A(t)}\mathbb{R}^9 \cong \mathbb{R}^9$, which contains as a linear subspace, the tangent space of SO(3) (considered as the image of the differential of the inclusion map). Moreover, for any $R \in SO(3)$, we can define the canonical basis for $\mathrm{Tan}_R SO(3)$ as $\{R\mathbf{e}_1, R\mathbf{e}_2, R\mathbf{e}_3\}$ by mapping $\{\mathbf{e}_1, \mathbf{e}_2, \mathbf{e}_3\}$ through the linear isomorphism given by the differential of the left-multiplication map $L_R(R') = RR'$ (this map is a diffeomorphism as smoothness is given by the definition of the Lie group and we have the smooth inverse $L_R^{-1}(R') = R^{-1}R$).

We note that another common representation of $\mathfrak{so}(3)$ is the axis-angle parametrisation - *i.e.* for any $S \in \mathfrak{so}(3)$, there exists some $u \in S^2 \subset \mathbb{R}^3$ (the axis) and $\omega \in \mathbb{R}^+$ (the angle) such that $S = [\omega u]_\times$.

**Exponential and logarithmic maps.** The exponential map $\exp : \mathfrak{g} \to G$ and its inverse, the logarithmic map $\log : G \to \mathfrak{g}$ are fundamental tools for studying Lie groups. They provide a canonical (local) diffeomorphism between the flat Lie algebra and the curved Lie group, thus allowing for the parametrisation of the manifold by an easy-to-work-with vector space and translating geometric problems to problems of linear algebra.

---

[11]Strictly, this should also be considered as equipped with a Lie bracket.

In the case of $\mathrm{SO}(3)$, the exponential map $\exp : \mathfrak{so}(3) \to \mathrm{SO}(3)$ has an analytic form, given by Rodrigues' formula (considering $S \in \mathfrak{so}(3)$ in terms of its angle-axis parametrisation):

$$\exp(S) = \mathbf{I} + \frac{\sin \omega}{\omega} S + \frac{1 - \cos \omega}{\omega^2} S^2,$$

Furthermore, the logarithmic map $\log : \mathrm{SO}(3) \to \mathfrak{so}(3)$ has the form:

$$\log R = \frac{\theta}{2 \sin \theta} (R - R^\top), \quad \text{where } \mathrm{Tr}(R) = 1 + 2 \cos \theta.$$

## C   FURTHER DETAILS ON SE(3) DIFFUSION

In this section, we provide an extended discussion on the construction of diffusion models on SE(3); this is mainly recapped from the seminal work: Yim et al. (2023).

**Riemannian metric.** In order to define a diffusion process on the Lie group SE(3), we note that the generator for Euclidean Brownian motion is given by the Laplace operator $\Delta$. Therefore, to define an analogous (diffusion) Markov process on SE(3), we can use the Laplace–Beltrami operator, which is a generalisation of the Laplace operator to Riemannian manifolds. This requires endowing SE(3) with a Riemannian metric.

First, we note that tangent space to SE(3) can be expressed as a direct sum: $\text{Tan}_{(p,R)}\, \text{SE}(3) = \text{Tan}_p\, \mathbb{T}(3) \oplus \text{Tan}_R\, \text{SO}(3)$ which we equip with their canonical bases. Yim et al. (2023) then proposes the choice of metric $\langle \cdot, \cdot \rangle_{\text{SE}(3)}$ defined by

$$\langle (a, S), (a', S') \rangle_{\text{SE}(3)} = \langle a, a' \rangle_{\mathbb{T}(3)} + \langle S, S' \rangle_{\text{SO}(3)},$$

where $(a, S), (a', S') \in \text{Tan}_{(p,R)}\, \text{SE}(3)$ for some $(p, R) \in \text{SE}(3)$, and $\langle a, a' \rangle_{\mathbb{T}(3)} = \sum_{i=1}^{3} a_i \cdot a_i'$ and $\langle S, S' \rangle_{\text{SO}(3)} = \frac{1}{2} \text{Tr}(SS'^{\top})$ are the canonical metrics for $\mathbb{T}(3)$ and SO(3) respectively. We note that the canonical basis of $\text{Tan}_{(p,R)}\, \text{SE}(3)$ is orthonormal under this metric. Essentially, this choice of metric allows us to view SE(3) as the Riemannian product manifold $\mathbb{T}(3) \times \text{SO}(3)$ and allows for the factorisation of diffusion processes over translations and rotations. The Riemannian metric on SE(3)$^m$ is given by the standard extension.

**Noise schedules.** For ease of presentation, we only consider a non-time dependent forward process in Section 2.3, however, following Yim et al. (2023), our actual implementation fixes $T = 1$ and uses the following decoupled time-scaling of the forward translational and rotational SDEs:

- Our translation SDE is given by

$$\mathrm{d}\mathbf{p}^{(t)} = -\frac{1}{2}\beta(t)\mathbf{p}^{(t)}\mathrm{d}t + \sqrt{\beta(t)}\mathrm{d}\mathbf{B}^{(t)}_{\mathbb{R}^{m \times 3}},$$

where $\beta(t) = \beta_{\min} + t(\beta_{\max} - \beta_{\min})$.

- Our rotational SDE is given by

$$\mathrm{d}\mathbf{R}^{(t)} = \sqrt{g(t)}\mathrm{d}\mathbf{B}^{(t)}_{\text{SO}(3)^m},$$

where $g(t) = \frac{d}{dt}\sigma^2(t)$ and $\sigma(t) = \log(t \exp(\sigma_{\max}) + (1 - t)\exp(\sigma_{\min}))$.

The backward SDE then has the form:

$$\mathrm{d}\overleftarrow{\mathbf{Z}}^{(t)} = \begin{bmatrix} \frac{1}{2}\beta(1-t)\overleftarrow{\mathbf{p}}^{(t)} + \beta(1-t)\nabla_p \log p_{1-t}(\overleftarrow{\mathbf{Z}}^{(t)}) \\ g(1-t)\nabla_R \log p_{1-t}(\mathbf{Z}^{(t)}) \end{bmatrix} \mathrm{d}t + \begin{bmatrix} \sqrt{\beta(1-t)}\mathrm{d}\mathbf{B}^{(t)}_{\mathbb{R}^{m \times 3}} \\ \sqrt{g(1-t)}\mathrm{d}\mathbf{B}^{(t)}_{\text{SO}(3)^m}. \end{bmatrix}$$

Going ahead, we will refer to these as the definition of our forward and backward SDEs.

**Forward kernels.** By the structure of the metric $\langle \cdot, \cdot \rangle_{\text{SE}(3)}$, the forward kernel $p_{t|0}(\mathbf{z}^{(t)}|\mathbf{z}^{(0)})$ (for ease of presentation, we only consider $m = 1$ here) can be factorised into independent kernels over $\mathbb{T}(3)$ and SO(3).

- For the translational component, we have the standard VP forward kernel:

$$p_{t|0}(p^{(t)}|p^{(0)}) = \mathcal{N}(p^{(t)}; \alpha_t p^{(0)}, (1 - \alpha_t^2)\mathbf{I}), \text{ where } \alpha_t = \exp\left(-\frac{1}{2}\int_0^t \beta(s)\mathrm{d}s\right).$$

- For the rotational component, the forward kernel is given by:

$$p(R^{(t)}|R^{(0)}) = \mathcal{IG}_{\text{SO}(3)}(R^{(t)}; R^{(0)}, \sigma^2(t)).$$

To sample from $R^{(t)} \sim \mathcal{IG}_{\mathrm{SO}(3)}(R^{(t)}; R^{(0)}, \sigma^2)$, we can first sample $R \sim \mathcal{IG}_{\mathrm{SO}(3)}(R; \mathbf{I}, \sigma^2)$ through independently sampling $\omega \in [0, \pi]$ according to the following density:

$$f(\omega, \sigma) = \frac{1 - \cos \omega}{\pi} \sum_{l=0}^{\infty} (2l + 1) e^{-l(l+1)\sigma^2/2} \frac{\sin((l + \frac{1}{2})\omega)}{\sin(\omega/2)},$$

and sampling $u \sim \mathcal{U}_{S^2}$ where $\mathcal{U}_{S^2}$ denotes the uniform distribution on the sphere (e.g. sampling from isotropic Gaussian and then dividing by the norm). We then compute $R = \exp([\omega u]_\times) \sim \mathcal{IG}_{\mathrm{SO}(3)}(R; \mathbf{I}, \sigma^2)$ and return the sample: $R^{(t)} = RR^{(0)} \sim \mathcal{IG}_{\mathrm{SO}(3)}(R^{(t)}; R^{(0)}, \sigma^2)$.

To sample from $f(\omega, \sigma)$, following previous work (Leach et al., 2022; Yim et al., 2023), we truncate the infinite series in order to compute an approximation to the distribution's CDF. We then cache these values and draw samples with inverse transform sampling.

**Stationary distribution.** To sample from the uniform distribution $\mathcal{U}_{\mathrm{SO}(3)}$ over $\mathrm{SO}(3)$, we independently sample $\omega \in [0, \pi]$ according to the following density:

$$p(\omega) = \frac{1 - \cos \omega}{\pi},$$

and sample $u \in \mathcal{U}_{S^2}$. We then return $R = \exp([\omega u]_\times) \sim \mathcal{U}_{\mathrm{SO}(3)}$. We note that this is equivalent to sampling from $\mathcal{IG}_{\mathrm{SO}(3)}(0, \sigma^2)$ in the limit as $\sigma \to \infty$.

**Conditional scores.** For the score matching objective, we require the form of the conditional scores over translations and rotations. For translations, this is a standard computation, and for rotations, we first require the modified function: $f_0(\omega, \sigma) = f(\omega, \sigma)/p(\omega)$.

This is due to the fact that $f(\omega, \sigma)$ denotes the marginal density of the angle $\omega$, hence the density $\mathcal{IG}_{\mathrm{SO}(3)}(R^{(t)}; R^{(0)}, \sigma^2)$, which has support over $\mathrm{SO}(3)$ instead, requires a different factor. We then have the following formula:

$$\nabla_R \log p_{t|0}(R^{(t)} | R^{(0)}) = \frac{R^{(t)}}{\omega(t)} \log R^{(0,t)} \frac{\partial_\omega f_0(\omega(t), \sigma(t))}{f_0(\omega(t), \sigma(t))},$$

where $R^{(0,t)} = (R^{(0)})^\top R^{(t)}$ and $\omega(t) = \omega(R^{(0,t)})$ is the angle in the axis-angle representation of $R^{(0,t)}$ - this can be computed from the equation: $\mathrm{Tr}\, R^{(0,t)} = 1 + 2\cos(\omega(R^{(0,t)}))$. Similarly, we truncate the infinite sum in $f_\theta(\omega, t)$ in order to compute the above score.

**Loss scaling.** We base our loss scaling off of the standard heuristic given in (Song et al., 2020) from the reciprocal of the expected value of the squared norm of the conditional score. For the translational loss, it is a standard calculation to show $\mathbb{E} \left\| \nabla_p \log p_{t|0}(p^{(t)} | p^{(0)}) \right\|_{\mathbb{T}(3)}^2 \propto \hat{\lambda}_t^p = \frac{1}{1 - \alpha_t^2}$. For the rotational loss, we have the following proposition. For the proof[12], see Appendix C.1.

**Proposition 1.** *The expected squared norm of the conditional rotational score is given by*

$$\hat{\lambda}_t^R = \mathbb{E} \left\| \nabla_R \log p_{t|0}(R^{(t)} | R^{(0)}) \right\|_{\mathrm{SO}(3)}^2 = \mathbb{E}_{\omega \sim f(\omega, \sigma^2(t))} \left[ \left( \frac{\partial_\omega f_0(\omega, \sigma(t))}{f_0(\omega, \sigma(t))} \right)^2 \right].$$

We then define our final loss scaling by $\lambda_t^p = C^p / \hat{\lambda}_t^p$ for translations and $\lambda_t^R = C^R / \hat{\lambda}_t^R$ for rotations where $C^p, C^R > 0$ are some chosen hyper-parameters. We note that we numerically compute the expectation in $\hat{\lambda}_t^R$ from our truncated approximation to $f_0(\omega, \sigma)$.

---

[12]We note that it is surprising that, to the best of our knowledge, this proof is not seen in the literature.

## C.1 Proof of Proposition 1

*Proof.* We first note that

$$\left\|\nabla_R \log p_{t|0}(R^{(t)}|R^{(0)})\right\|^2_{SO(3)}$$

$$= \left(\frac{\partial_\omega f_0(\omega(t), \sigma(t))}{\omega(t) f_0(\omega(t), \sigma(t))}\right)^2 \frac{1}{2} \mathrm{Tr}\left(R^{(t)} \log R^{(0,t)} (\log R^{(0,t)})^\top (R^{(t)})^\top\right)$$

$$= \left(\frac{\partial_\omega f_0(\omega(t), \sigma(t))}{\omega(t) f_0(\omega(t), \sigma(t))}\right)^2 \frac{1}{2} \mathrm{Tr}\left(\log R^{(0,t)} (\log R^{(0,t)})^\top\right),$$

due to $\mathrm{Tr}(AB) = \mathrm{Tr}(BA)$. Moreover, since $R^{(t)} \sim \mathcal{IG}_{SO(3)}(R^{(t)}; R^{(0)}, \sigma^2(t))$, we know that $R^{(t)} = \exp([\omega u]_\times) R^{(0)}$ with $\omega \sim f(\omega, \sigma^2(t))$ and $u \sim \mathcal{U}_{S^2}$, which implies that $R^{(0,t)} = \exp([\omega u]_\times)^\top = \exp([-\omega u]_\times)$ by the definition of $[\cdot]_\times$ and using the fact that $\exp(S)^\top = \exp(S^\top)$ by the definition of the exponential map. As we have $u \sim -u$, we can conclude that $R^{(0,t)} \sim \exp([\omega u]_\times)$. Moreover, we have $\omega(\exp([\omega u]_\times)) = \omega$ as $\omega(\cdot)$ returns the angle of the input rotation[13].

This allows us to rewrite $\mathbb{E}\left\|\nabla_R \log p_{t|0}(R^{(t)}|R^{(0)})\right\|^2_{SO(3)}$ as

$$\mathbb{E}\left\|\nabla_R \log p_{t|0}(R^{(t)}|R^{(0)})\right\|^2_{SO(3)} =$$

$$\mathbb{E}_{\omega,u}\left[\left(\frac{\partial_\omega f_0(\omega, \sigma(t))}{\omega f_0(\omega, \sigma(t))}\right)^2 \frac{1}{2} \mathrm{Tr}\left(\log \exp([\omega u]_\times)(\log \exp([\omega u]_\times))^\top\right)\right].$$

Further, we see that $\log \exp([\omega u]_\times) = [\omega u]_\times$ and that $\frac{1}{2}\mathrm{Tr}([\omega u]_\times [\omega u]_\times^\top) = \omega^2 \|u\|_2^2 = \omega^2$ as $u \in S^2$. Putting this together, we can conclude

$$\mathbb{E}\left\|\nabla_R \log p_{t|0}(R^{(t)}|R^{(0)})\right\|^2_{SO(3)} = \mathbb{E}_{\omega \sim f(\omega, \sigma^2(t))}\left[\left(\frac{\partial_\omega f_0(\omega, \sigma(t))}{f_0(\omega, \sigma(t))}\right)^2\right].$$

$\square$

## C.2 Proof of Theorem 1

Let a molecule with $N$ atoms have Cartesian coordinates $\mathbf{x} = (x_1, \ldots, x_N) \in \mathcal{X} : \mathbb{R}^{N \times 3}$, where each $x_i \in \mathbb{R}^3$. Let $p_{\mathcal{X}} := \mathcal{X} \to \mathbb{R}^+$ be any target density on Cartesian space (with respect to Lebesgue measure $d\mathbf{x}$), covering equilibrium Boltzmann distributions, forward-diffusion marginals $p_t$, empirical data density, *etc.*

$$p_{\mathcal{X}}(\mathbf{x}) = \frac{1}{Z} \exp(-\beta E(\mathbf{x})),$$

where $\beta$ is a thermal constant and $E(\mathbf{x}) : \mathbb{R}^{N \times 3} \to \mathbb{R}$ is the energy (scalar) function.

The purpose of this proof is *geometric*: compare the Riemannian (Jacobian) base measures induced by two different parametrisations of the same constrained Cartesian manifold and show that, under standard molecular topologies, the torsional parametrisation generically does *not* induce a product base measure while a disjoint rigid-fragment parametrisation does.

**Torsions.** Fix all bond lengths and bond angles and parametrize the remaining internal degrees of freedom by $k$ torsional angles $\boldsymbol{\phi} = (\phi_1, \ldots, \phi_k) \in \Phi := \mathbb{T}^k$, where $\phi_i \in S^1 \cong SO(2)$. Together with a global rigid-body pose $(R, p) \in SE(3)$ this gives the parameter space $\mathcal{U} = SE(3) \times \mathbb{T}^k$. Let

$$\psi : \mathcal{U} = SE(3) \times \mathbb{T}^k \to \mathbb{R}^{N \times 3}, \qquad u = (R, p, \boldsymbol{\phi}) \mapsto \mathbf{x}$$

denote the parametrisation of Cartesian configurations with prescribed fixed bonds and angles.

---

[13]This is trivial to show.

**Rigid fragments.** Split the molecule into $m$ disjoint rigid fragments with local coordinates $\tilde{\mathbf{x}}_{F_i} \in \mathbb{R}^{|\mathcal{G}_{F_i}| \times 3}$ for $i = 1, \ldots, m$. Let $\mathbf{z} = (z_1, \ldots, z_m) \in \mathcal{Z} = \mathrm{SE}(3)^m$ with $z_i = (p_i, R_i) \in \mathrm{SE}(3)$, and define

$$\varphi : \mathcal{Z} \to \mathbb{R}^{N \times 3}, \qquad z \mapsto \mathbf{x},$$

by placing each fragment via the group action

$$\mathbf{x}_{F_i} = R_i \tilde{\mathbf{x}}_{F_i} + p_i, \qquad i = 1, \ldots, m.$$

Assuming the fragments are disjoint (no shared atoms) this yields an explicit parametrisation of the constrained Cartesian configurations obtained by rigidly placing each fragment.

**Important distinction.** The data density $p_{\mathcal{X}}$ at $t = 0$ is identical under any parametrisation, since it reflects the true molecular distribution. What differs is the forward (noising) process: the marginals $p_t(\mathbf{x}_t)$ depend on the parameter space in which the dynamics are defined. In other words, while the starting distribution $p_0(\mathbf{x}_0)$ is fixed in Cartesian space, the choice of coordinates (e.g. torsions or rigid fragments) determines how the forward dynamics evolve and hence the form of the intermediate marginals.

**Preliminaries.** For a parametrisation $\Omega$ denoting either parametrisation map ($\psi$ or $\varphi$) and a parameter $\mathbf{y}$ in the corresponding parameter space, define the Jacobian

$$J_\Omega(\mathbf{y}) = \frac{\partial \Omega}{\partial \mathbf{y}} \in \mathbb{R}^{3N \times d},$$

where $d$ is the parameter-space dimension. We define the Gram matrix as $G_\Omega(\mathbf{y}) = J_\Omega(\mathbf{y})^\top J_\Omega(\mathbf{y}) \in \mathbb{R}^{d \times d}$. The induced Riemannian volume element on parameter space is

$$d\mu_\Omega(\mathbf{y}) = \sqrt{\det G_\Omega(\mathbf{y})}\, d\mathbf{y},$$

where $d\mathbf{y}$ is the Lebesgue element in the local parameter space. For densities $p_{\mathcal{X}}$ on $\mathcal{X}$ and $p_{\mathcal{Y}}$ on parameter space $\mathcal{Y}$, the change-of-variables (push-forward) relation is the equality of measures

$$p_{\mathcal{Y}}(\mathbf{y}) d\mu_\Omega(\mathbf{y}) = p_{\mathcal{X}}(\mathbf{x}) d\mathbf{x}$$

which simplifies to

$$p_{\mathcal{Y}}(\mathbf{y}) = p_{\mathcal{X}}(\Omega(\mathbf{y})) \sqrt{\det G_\Omega(\mathbf{y})}.$$

Equivalently

$$p_{\mathcal{X}}(\Omega(\mathbf{x})) = \frac{p_{\mathcal{Y}}(\mathbf{y})}{\sqrt{\det G_\Omega(\mathbf{y})}}.$$

### C.2.1 TORSIONAL ENTANGLEMENT

*Proof.* Torsional models suffer from geometric coupling and a highly intricate non-product induced measure.

Consider the torsional parametrisation $\psi$. The parameter space is

$$\mathcal{U} = \mathrm{SE}(3) \times \mathbb{T}^k, \qquad u = (R, p, \boldsymbol{\phi})$$

and the torsional parametrisation is the map $\psi : \mathcal{U} \to \mathbb{R}^{N \times 3}, \ u \mapsto \mathbf{x}$. Its Jacobian and Gram matrix are

$$J_\psi(u) = \frac{\partial \psi}{\partial u} \in \mathbb{R}^{3N \times (k+6)}, \qquad G_\psi(u) = J_\psi^\top(u) J_\psi(u) \in \mathbb{R}^{(k+6) \times (k+6)}$$

and the induced Riemannian volume on parameter space is $d\mu_\psi(u) = \sqrt{\det G_\psi(u)}\, du$.

Index torsional coordinates by $i, j \in \{1, \ldots, k\}$. The column $J_\psi$ corresponding to $\phi_i$ is the instantaneous Cartesian displacement vector obtained by varying torsion $\phi_i$. Writing atomic positions $x_a \in \mathbb{R}^3, a \in \{1, \cdots, N\}$, the torsion-torsion Gram matrix is

$$G_\psi(u)_{ij} = \langle (J_\psi)_i, (J_\psi)_j \rangle = \sum_{a=1}^{3N} \frac{\partial x_a}{\partial \phi_i}(u) \frac{\partial x_a}{\partial \phi_j}(u).$$

In standard bonded molecular topologies (chains, branched trees, rings with shared downstream atoms) a change in a single torsion displaces multiple downstream atoms. Additionally, the sets of atoms that move when different torsions are varied typically overlap (*i.e.*, the same downstream atoms are moved by multiple torsions). Hence, there generally exist pairs $i \neq j$ and atoms $a$ for which both derivatives are non-zero, producing off-diagonal entries $(G_\psi)_{ij} \neq 0$. Thus, the torsion block $G_\psi(u)$ is generically non-diagonal. Since the determinant of a matrix with non-zero off-diagonals does not factor as a product of single-coordinate functions, $\det G_\psi(u)$ does not decompose as $\prod_i g_i(\phi_i)$. Consequently, the induced volume form

$$d\mu_\psi(u) = \sqrt{\det G_\psi(u)}\, du$$

is *not a product measure* across torsions.

Therefore, independent perturbations in torsion space is mapped to Cartesian marginals with metric-induced correlations: torsional independence does not imply Cartesian independence. As a result, forward-backward kernels in torsion space correspond to correlated and anisotropic displacements in Cartesian space, complicating diffusion processes. In addition, a model that observes Cartesian coordinates must implicitly learn this metric-induced coupling, yielding unstable training dynamics, and degeneracy during inference. $\qquad\square$

**Gauge ambiguity (complementary issue).**  A torsion angle by itself does not uniquely determine which rigid sub-body rotates about a bond; mapping a torsional increment $\Delta\phi_i$ to Cartesian updates requires choosing a canonical extrinsic gauge (heuristics such as RMSD alignment are often situation-dependent and non-differentiable): An update $\Delta\phi_i^{ABCD}$ defined in $\mathcal{S}^1$ can rotate bodies $A_i$ and $B_i$ equiprobably such that $\Delta\phi_i^{ABCD} = \theta_i^{ABC} - \theta_i^{BCD}$. Since we have 2 loose variables that are not determined by the torsion alone, the mapping from $\Delta\phi_i^{ABCD}$ to Cartesian updates is undetermined, systematically pushing the marginal densities to lower likelihood regions at inference.

**Global-pose coupling.**  Augmenting torsional coordinates with a global pose $(R, p) \in \mathrm{SE}(3)$ (*i.e.* treating the parameter space as $\mathbb{T}^k \times \mathrm{SE}(3)$) *does not* in general restore a product space.

Torsional changes typically (i) shift the molecule's center-of-mass (CoM) and (ii) alter its orientation-dependent moments of inertia. As a result, the Jacobian acquires additional non-zero cross-terms between torsion and translation/rotation columns. Concretely:

- The *translation–torsion cross term* is proportional to the net displacement of the molecular CoM induced by a torsional $\phi_i$:

$$\frac{d}{d\phi_i}\mathrm{CoM}(\phi_i)\ \propto\ \sum_a m_a \frac{\partial x_a}{\partial \phi_i},$$

  where, for uniform masses, this reduces to the unweighted centroid shift). This quantity is generically non-zero, so torsions couple to non-zero translations about the CoM.

- The *rotation-torsion cross term* is proportional to the instantaneous change in angular momentum induced by the torsional displacement field:

$$\sum_a m_a\, x_a \times \frac{\partial x_a}{\partial \phi_i},$$

  which can be interpreted as the torsion-induced torque on the molecular frame. This term is generically non-zero, so torsions also couple to rotations.

Only in rare cases where both the CoM shift and the torque vanish for every torsion (or their mass-weighted analogues vanish under a physical metric), or in infinitesimally small time changes, do torsions decouple from global $\mathrm{SE}(3)$, yielding a quasi-block-diagonal metric and an approximate product structure $\mathbb{T}^k \times \mathrm{SE}(3)$. Although heuristics such as RMSD alignment can be applied to mitigate this issue, this is not a general solution and is only strictly true for infinitesimally small $\Delta\phi$): RMSD can only attempt to mitigate net body motions induced from torsional updates.

**Degenerate exceptions.**  If two torsions act on strictly disjoint atom sets (no downstream overlap), or only a single torsional bond exists such that it cannot form a branch or chain, the corresponding Jacobian columns have disjoint support and the associated blocks of $G_\psi$ may be diagonal; in such degenerate topologies the geometric term can factorise.

### C.2.2 RIGID FRAGMENTS

*Proof.* Rigid-fragment parametrisation induces a true product measure.

Partition a molecule into $m$ rigid, disjoint fragments with local coordinates $\tilde{\mathbf{x}}_{F_i} \in \mathbb{R}^{|\mathcal{G}_{F_i}| \times 3}, i \in \{1, \cdots, m\}$. The parameter space is

$$\mathcal{Z} = \mathrm{SE}(3)^m, \qquad \mathbf{z} = (z_1, \cdots, z_m), \qquad z_i = (R_i, p_i),$$

and the rigid-fragment parametrisation is the map

$$\varphi : \mathcal{Z} \to \mathbb{R}^{N \times 3}, \qquad \mathbf{z} \to \mathbf{x}, \qquad \mathbf{x}_{F_i} = R_i \tilde{\mathbf{x}}_{F_i} + p_i,$$

which by construction is block-wise rigid:

$$\mathbf{x} = \varphi(z_1, \ldots, z_m) = \bigoplus_{i=1}^{m} (R_i, p_i) \cdot \tilde{\mathbf{x}}_{F_i}.$$

Its Jacobian, Gram matrix and induced Riemannian volume form are

$$J_\varphi(z) = \frac{\partial \varphi}{\partial z} \in \mathbb{R}^{3N \times 6m}, \qquad G_\varphi(z) = J_\varphi^\top(z) J_\varphi(z) \in \mathbb{R}^{6m \times 6m}, \qquad d\mu_\varphi(z) = \sqrt{\det G_\varphi(z)} \, dz.$$

From the definition of the block-wise rigid mapping in $\varphi$ and the disjointness in the fragments, moving one fragment does not affect the Cartesian positions of any other fragment. Concretely the Jacobian has a block-diagonal structure

$$J_\varphi(z) = \begin{bmatrix} J_{11} & 0 & \ldots & 0 \\ 0 & J_{22} & \ldots & 0 \\ \vdots & & \ddots & \vdots \\ 0 & 0 & \ldots & J_{mm} \end{bmatrix}, \qquad J_{ii} = \frac{\partial \mathbf{x}_{F_i}}{\partial z_i} \in \mathbb{R}^{3|\mathcal{G}_{F_i}| \times 6}.$$

As a result, the Gram matrix is also block-diagonal:

$$G_\varphi(z) = \mathrm{diag}\big(J_1^\top J_1, \ldots, J_m^\top J_m\big),$$

so that the determinant factorises:

$$\det G_\varphi(z) = \prod_{i=1}^{m} \det \big(J_i^\top J_i\big).$$

Hence the induced volume element

$$d\mu_\varphi(z) = \sqrt{\det G_\varphi(z)} \, dz = \prod_{i=1}^{m} \sqrt{\det \big(J_i^\top J_i\big)} \, dz_i,$$

is *a product measure over fragments*.

Therefore, independent noise applied to each fragment in the parameter space remains independent in Cartesian space via the linear rigid mapping. This decoupling makes diffusion or generative modelling much simpler via the rigid body $\mathrm{SE}(3)$ parametrisation. $\qquad \square$

## D    FURTHER DETAILS ON FRAGMENTATION & CONFORMATIONAL ALIGNMENT

### D.1    HOLONOMIC CONSTRAINTS

Let $\mathbf{x} = (x_1, \ldots, x_{|\mathcal{G}_{\text{ligand}}|}) \in \mathbb{R}^{|\mathcal{G}_{\text{ligand}}| \times 3}$ be the Cartesian coordinates of the ligand atoms. The holonomic map

$$g(\mathbf{x}) = \big(g_1(\mathbf{x}), \ldots, g_m(\mathbf{x})\big)^\top$$

is built from scalar constraints that enforce bond lengths and bond angles inferred from the molecular graph $\mathcal{G}_{\text{ligand}}^{\text{2D}}$. Below we give simple valid forms which (softly) approximate the holonomic constraints assumed in the construct of our method.

**Bond-length constraint.**    For a bonded pair $(A, B)$ with equilibrium length $d_{AB} = d_0$ define

$$r_{AB} := x_A - x_B, \qquad g_{(AB)}(\mathbf{x}) = \|r_{AB}\| - d_0.$$

Thus $g_{(AB)}(\mathbf{x}) \approx 0$ enforces $\|r_{AB}\| \approx d_0$. The gradients (akin to vector field / forces) are

$$\frac{\partial g_{(AB)}}{\partial x_A} = \frac{r_{AB}}{\|r_{AB}\|}, \qquad \frac{\partial g_{(AB)}}{\partial x_B} = -\frac{r_{AB}}{\|r_{AB}\|},$$

and $\partial g_{(AB)}/\partial x_i = 0$ for $i \notin \{A, B\}$.

**Bond-angle constraint.**    For a bonded triplet $(A - B - C)$ with equilibrium angle $\tau_{ABC} = \tau_0$ let

$$r_{AB} = x_A - x_B, \qquad r_{CB} = x_C - x_B, \qquad u = \frac{r_{AB}}{\|r_{AB}\|}, \qquad v = \frac{r_{CB}}{\|r_{CB}\|}.$$

We use the cosine form

$$g_{(ABC)}(\mathbf{x}) = u \cdot v - \cos \tau_0,$$

so $g_{(ABC)}(\mathbf{x}) \approx 0$ enforces the desired angle. The partial derivatives are

$$\frac{\partial g_{(ABC)}}{\partial x_A} = \frac{1}{\|r_{AB}\|}(I - uu^\top)\, v, \qquad \frac{\partial g_{(ABC)}}{\partial x_C} = \frac{1}{\|r_{CB}\|}(I - vv^\top)\, u,$$

and

$$\frac{\partial g_{(ABC)}}{\partial x_B} = -\frac{\partial g_{(ABC)}}{\partial x_A} - \frac{\partial g_{(ABC)}}{\partial x_C}.$$

**Soft (harmonic) enforcement.**    Instead of imposing $g(\mathbf{x}) = 0$ as a hard constraint, we relax the requirements by instead assuming stiff harmonic penalties. For a bond and an angle the typical potentials are

$$E_{\text{bond}}(x_A, x_B) = \tfrac{1}{2}k_{AB}\big(\|r_{AB}\| - d_0\big)^2, \qquad E_{\text{angle}}(x_A, x_B, x_C) = \tfrac{1}{2}k_{ABC}\big(u \cdot v - \cos \tau_0\big)^2.$$

For large force constants $k_{AB}, k_{ABC} \gg 1$ the Boltzmann measure concentrates near $g(\mathbf{x}) = 0$ and approximates the hard-constrained manifold; for finite (but large) constants the potentials provide numerical stability and allow standard unconstrained sampling while strongly biasing configurations toward $\mathcal{M}_c$. These harmonic constraints are a fundamental component in physics-based and template-based conformer generators such as ETKDGv3 (Landrum, 2025). Crucially, this assumption is paramount to our SE(3) rigid-body framework as it allows us to safely assume that the variation in bond angles and lengths is negligible (up to a certain error rate, see D.3) when defining our local coordinates during sampling.

### D.2    PROOF OF LEMMA 1

Let $\overline{BC}$ be a torsional bond at the center of a set of dihedrals defined as $\phi_{A:BC:\mathcal{D}}$ and let $\mathcal{A} = \{A_p\}_{p \in F_B}$ denote the set of neighbour atoms on the $B$-side (connected to $B$), and $\mathcal{D} = \{D_q\}_{q \in F_A}$

denote the set of neighbour atoms on the $C$-side (connected to $C$). Suppose the following pairwise distances are defined as:

$$\forall A_p \in \mathcal{A}: \quad \|A_p - B_{\mathcal{A}}\|, \; \|B_{\mathcal{A}}(t) - C_{\mathcal{D}}(t)\|, \; \|A_p(t) - C_{\mathcal{D}}(t)\|,$$

$$\forall D_q \in \mathcal{D}: \quad \|B_{\mathcal{A}}(t) - C_{\mathcal{D}}(t)\|, \; \|C_{\mathcal{D}} - D_q\|, \; \|B_{\mathcal{A}}(t) - D_q(t)\|,$$

where gray terms denote cross-edges across fragments that are time-dependent and therefore varying according to the forward-backward diffusion process on SE(3). For clarity, we use the notation $B_{\mathcal{A}}(t)$ to represent the node $B$ of the dihedral $\phi_{\mathcal{A}:BC:\mathcal{D}}$ corresponding to fragment $\mathcal{A}$. This is important because $B$ and $C$ are duplicated (as dummy atoms) in the fragmentation process to preserve torsional bond information. An illustration is shown in Figure 5.

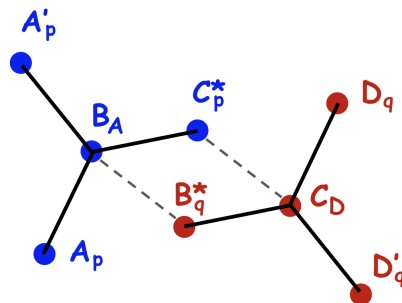

Figure 5: Visual helper for Lemma 1 showing corresponding atoms for fragment $\mathcal{A}$ shown in blue and fragment $\mathcal{D}$ denoted in red. For additional clarity, dummy atoms across torsional bond $\overline{BC}$ are marked with an asterisk.

Thus, during the forward noising process, $B_{\mathcal{A}}(t) \neq B_{\mathcal{D}}(t), \; \forall t \in (0,1]$, and $B_{\mathcal{A}}(t=0) = B_{\mathcal{D}}(t=0)$. Note that we absorb time for the other distance definitions as these are always preserved: distances are fixed through time by from the canonical fragment template $\mathcal{G}_F$ (i.e. due to fixed internal bond angles). To simplify notation, we will also absorb time for all other variables for the rest of the proof but keep the colouring.

For every $A_p$ the triangle $(A_p, B_{\mathcal{A}}, C_{\mathcal{D}})$ is rigidified via side-side-side constraints (SSS). Equivalently, for every $D_q$ the triangle $(B_{\mathcal{A}}, C_{\mathcal{D}}, D_q)$ is also rigidified. Subsequently, for every $A_p \in \mathcal{A}$ and $D_q \in \mathcal{D}$ the corresponding adjacent bond angles $\angle(A_p, B_{\mathcal{A}}, C_{\mathcal{D}})$ and $\angle(B_{\mathcal{A}}, C_{\mathcal{D}}, D_q)$ are uniquely determined by the fixed side lengths and therefore invariant under any motion that preserves the listed distances (in particular under any torsional update in $\delta\phi_{ABCD}$ that preserves those distances).

*Proof.* Fix any $A_p \in \mathcal{A}$. The triple of distances $\{\|A_p - B_{\mathcal{A}}\|, \|B_{\mathcal{A}} - C_{\mathcal{D}}\|, \|A_p - C_{\mathcal{D}}\|\}$ specifies triangle $(A_p, B_{\mathcal{A}}, C_{\mathcal{D}})$ up to rigid motion. By the law of cosines,

$$b^2 = a^2 + c^2 - 2ac\cos(b)$$

rearranging and plugging in our terms, we define the cosine of our angle as

$$\cos\angle(A_p, B_{\mathcal{A}}, C_{\mathcal{D}}) = \frac{\|A_p - B_{\mathcal{A}}\|^2 + \|B_{\mathcal{A}} - C_{\mathcal{D}}\|^2 - \|A_p - C_{\mathcal{D}}\|^2}{2\|A_p - B_{\mathcal{A}}\|\|B_{\mathcal{A}} - C_{\mathcal{D}}\|},$$

so the angle $\angle(A_p, B_{\mathcal{A}}, C_{\mathcal{D}})$ is uniquely determined from the fixed lengths. An identical argument applies for any $D_q \in \mathcal{D}$ using the triple $\{\|B_{\mathcal{A}} - C_{\mathcal{D}}\|, \|C_{\mathcal{D}} - D_q\|, \|B_{\mathcal{A}} - D_q\|\}$. Hence all listed adjacent bond angles are fixed whenever the cross-edge stated distances are fixed, and any motion preserving those distances (including a torsional rotation that leaves them invariant) preserves the angles. In particular, the distances $\|A_p - C_{\mathcal{D}}\|$ and $\|B_{\mathcal{A}} - D_q\| \; \forall \, (p,q) \in \mathcal{A}, \mathcal{D}$ are the additional *triangulation conditioning edges* that pseudo-enforce these invariances through conditioning jointly with the symmetrical *torsional conditioning edges* $\|B_{\mathcal{A}} - C_{\mathcal{D}}\|, \|B_{\mathcal{A}} - C_{\mathcal{D}}\|$ and the *known* (from equilibrium chemistry) immutable bond lengths $\|A_p - B_{\mathcal{A}}\| = d_{AB}$ and $\|C_{\mathcal{D}} - D_q\| = d_{CD}$. $\qquad\square$

**Implications and remarks.**   The distances $\|A_p - C_{\mathcal{D}}\|$ and $\|B_{\mathcal{A}} - D_q\|$ are *new conditioning edges* introduced by triangulation: they do not typically appear as edges of the 2D chemical graph but are required to convert adjacent triples into SSS triangles and thereby fix angles via the law of cosines. Fixing these triangulation edges removes the continuous freedom in the adjacent bond angles: the angles become functions of the fixed side lengths. Consequently, triangulation reduces the effective (not strictly but trivially by conditioning) local degrees of freedom. In the idealised SSS case (one triangle on each side), the dihedral (signed) degree of freedom about $\overline{BC}$ is constrained to a discrete choice (mirror sign) rather than a continuous rotation; in other words, continuous torsional variation that would change the triangulation edges is no longer allowed.

Note that using many $A_p$ and $D_q$ triangles generally over-determines the local geometry and can introduce redundant or mutually constraining distance equalities. In practice, we select a consistent subset of triangulation edges and remove duplicates to avoid incompatibility. See Section 2.2.3 for more details on how we ensure that we avoid over-constraining the system by removing dummy atoms during fragmentation merging.

### D.3   FRAGMENTATION & ALIGNMENT

Further statistics for the alignment procedure described in Section 2.2.1 are shown in Figure 6. By selecting the best-aligned pose from 5 randomized conformers, we obtain an average RMSD of 0.21 Å between the aligned $\mathbf{x}'_b$ and the ground-truth $\mathbf{x}_b$. This provides strong empirical evidence that sampling fragments from $\mathbf{x}_c$ does not push the model out of distribution at test time. Moreover, the small alignment errors act as a form of regularisation during training, with typical RMSDs below 0.5 Å for any aligned conformer. In contrast, aligning only a single conformer results in an average RMSD of 0.39 Å. In both settings, all 85 ligands from the Astex diverse set produced conformers that could be aligned below the 2 Å acceptance threshold. These results indicate that, during both training and sampling, SIGMADOCK remains well within the distribution of valid conformers.

To further ensure that our $\mathbf{x}'_b$ samples are not chemically implausible, we run iterative optimisation with PoseBuster checks in place. We find this to be helpful in reducing high-energy alignments, and as we show in Figure 7, this is exacerbated in circumstances where a single alignment optimisation attempt is taken, where changes in internal energies above 10Kcal/mol are observed (highly implausible stereochemistry).

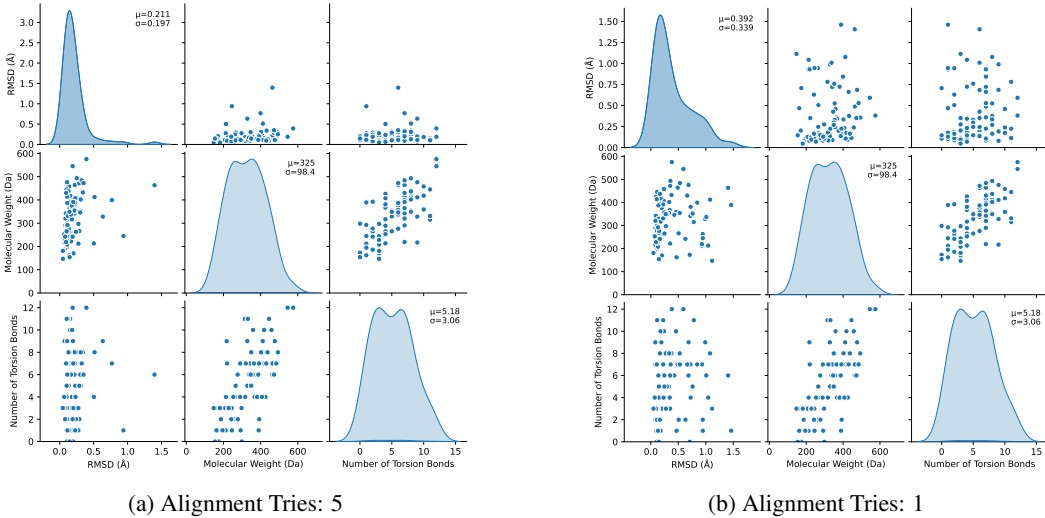

(a) Alignment Tries: 5             (b) Alignment Tries: 1

Figure 6: Pair-plot distributions across the alignment RMSDs (Å), molecular weight (Da), and number of torsional bonds across all 85 ligands in the Astex diverse set. We use the best (lowest RMSD) aligned pose from an initial set of conformation(s) generated from different random seeds.

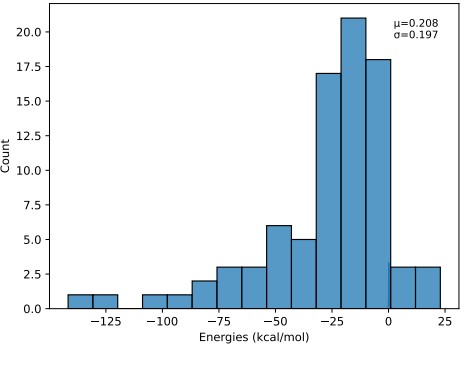 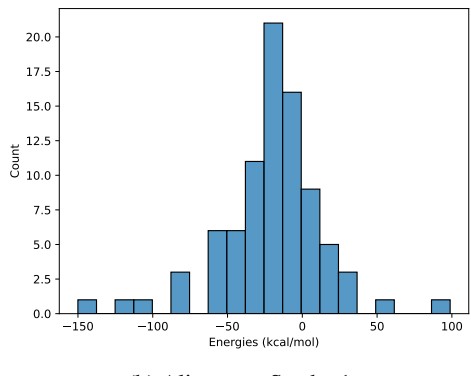

(a) Alignment Seeds: 5             (b) Alignment Seeds: 1

Figure 7: Histogram of internal energy differences (Kcal/mol) (bound pose vs. aligned pose) across all 85 ligands in the Astex diverse set. We use the best (lowest RMSD) aligned pose from an initial set of conformation(s) generated from different random seeds.

### D.4 FR3D

We provide a more concrete algorithmic breakdown of the logic behind FR3D in Algorithm 1. Note that at inference, we may choose different heuristics for `RecMerge`, allowing different valid-cut sets to be prioritised according to a selection criteria, *i.e.* by filtering out fragments with broken chiralities, or prioritising larger overall fragments. However, we leave this for future work and use the random fragmentation merging strategy for both training and inference.

We quantitatively contrast the differences between the fragments produced by FR3D and the classic $(k+1)$ approach, where all $k$ torsional bonds are split, through Figure 8.

**Degrees of freedom (DoF)** Let the ligand be partitioned into $m$ rigid fragments; the global pose is $z \in \mathrm{SE}(3)^m$ with $6m$ DoFs (3 rotation, 3 translation per fragment), which is far smaller than the $3N$ DoFs (coordinates) of all-atom point clouds ($N \gg m$). Naïve fragmentation by cutting all $k$ rotatable bonds yields $\hat{m} = k+1$ fragments, i.e., $6\hat{m}$ DoFs, which exceeds the torsional parameterisation ($k+6$). Our FR3D merging reduces $m$ empirically to $\approx \frac{2}{3}(k+1)$, lowering the raw coordinate count. Across each cut bond we introduce two $\mathrm{SE}(3)$-invariant *triangulation* distances (Sec. 2.2.3; Lemma 1) that fix adjacent bond angles and the torsional bond length while leaving the dihedral free. Denote by $e$ the number of cut bonds (edges) between fragments and stack all distance constraints into $c(z) \in \mathbb{R}^{2e}$. The hard-constrained manifold

$$\mathcal{M}_{\mathrm{frag}} = \{\, z \in \mathrm{SE}(3)^m : c(z) = 0 \,\}$$

has tangent dimension

$$\dim \mathcal{M}_{\mathrm{frag}} = 6m - \mathrm{rank}\big(J_c(z)\big).$$

For a *tree* fragment graph ($e = m-1$), two non-degenerate point-to-point distance constraints across a joint generically remove 5 relative DoF (leaving a single revolute DoF about the bond axis), so $\mathrm{rank}(J_c) = 5(m-1)$ and

$$\dim \mathcal{M}_{\mathrm{frag}} = 6m - 5(m-1) = m + 5.$$

Under naïve fragmentation ($m = k+1$) the hard (torsional) lower bound is $k+6$. With FR3D, each merge removes one cut and we drop one triangulation distance at the adjacent bond to avoid over-constraining the dihedral across adjacent fragments. Hence, the effective lower bound remains $k+6$ after merging, while the upper bound tightens to $6m$ such that $k+6 \leq \mathrm{DoF}_{\mathrm{eff}} \leq 6m$.

**Soft vs. hard constraints.** In practice, triangulation is injected as *soft*, $\mathrm{SE}(3)$-invariant edge features $c_t$ at each denoising step; no hard equalities are imposed. Thus trajectories concentrate near $\mathcal{M}_{\mathrm{frag}}$ as $t \to 0$, and the *effective* DoF lies between $k+6$ and $6m$. Consistent with Lemma 1, the only local DoF around a cut bond as $t \to 0$ is the dihedral $\Delta\phi_{ABCD}$. Empirically, removing these conditioning features reduces PB-validity by 12.8% (Table 1, Config A).

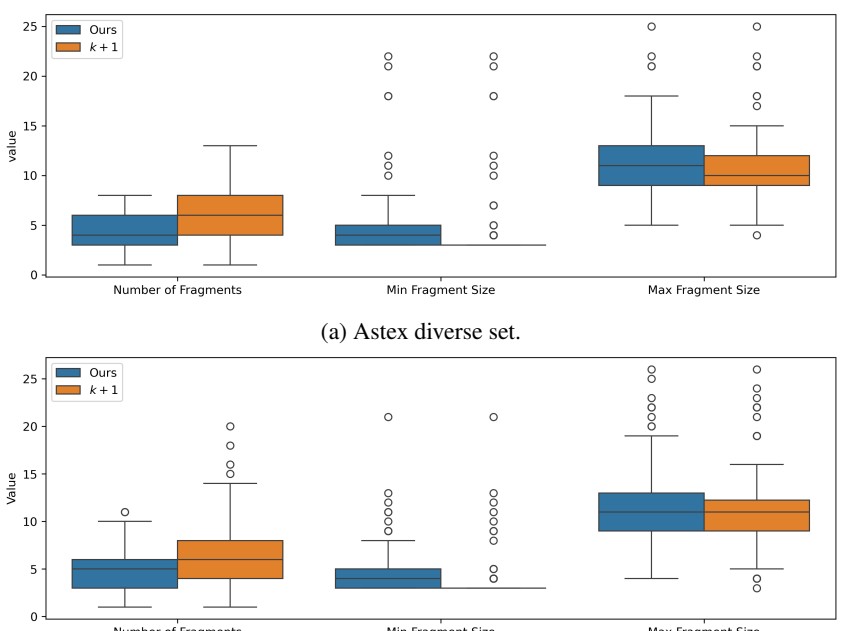

(a) Astex diverse set.

(b) PoseBusters set. Randomised sample of 100 ligands.

Figure 8: Whisker plots showing the number of fragments and the range (min and max) of fragment sizes (heavy atoms only) across the PoseBusters and Astex diverse set using either FR3D, or the classic $(k+1)$ fragmentation scheme.

# E  TRAINING DETAILS

## E.1  DATA PROCESSING

Here, we describe how we preprocess each sampled protein-ligand pair $(\mathcal{G}_{\text{ligand}}, \mathcal{G}_{\text{protein}}) \sim p_{\text{data}}$.

**Molecule parsing.**  We define proteins and ligands in raw form via the standard PDB and SDF formats. We parse ligand files with RDKit (Landrum, 2025) and force sanistisation to ensure correct kekulisation and featurisation. We parse protein files via BioPython (Cock et al., 2009), removing water molecules and other heteroatoms (ions, crystallisation aids, and other co-factors). For both, we remove all hydrogens (except, for ligands, those necessary to maintain correct valence). During training, we filter out ligands with more than 20 torsional bonds and/or those with higher molecular weight than 750 Daltons (excluding hydrogens). Since FR3D permutes across possible fragmentation merging options, this allows us to better scale training, whilst also maintaining our training distribution inside the desired range of drug-like compounds.

**Pocket selection.**  For a given protein-ligand pair, we define the pocket by including every residue that has at least one atom within a specified cut-off distance $d_0$ from any atom in the bound ligand pose. To help de-bias the selection of the pocket we use a stochastic cutoff $d_r = d_0 + \mathcal{N}(0, \sigma_r)$, where we use default values of $\sigma_r = 1, d_0 = 5$ for training and $\sigma_r = 0.5$ for sampling (all in Å). If there are multiple ligands present in the SDF file due to the presence of multiple chains, we randomly sample one. We note that we include residues from all protein chains as this is the closest representation of what the ligand sees during its binding event.

We define the pocket center of mass, which we define as the origin for our stationary distribution, by finding the center of mass of the pocket residues and adding an additional noise $\overline{x}_{\text{pocket}} \in \mathbb{R}^3 = \left( \frac{1}{|\mathcal{G}_{\text{pocket}}|} \sum_{x_i \in \mathcal{G}_{\text{pocket}}} x_i \right) + \mathcal{N}(0, \sigma_{\text{CoM}} \mathbf{I}_3)$. This is in an attempt to further de-bias the selection of the CoM away from the bound ligand's CoM and introduce robustness to the often arbitrary choice of the CoM in practical settings.

---

**Algorithm 1:** FR3D — Fragment REduction for Molecular Docking

---

**Input:** molecule mol, candidate torsions $\mathcal{C}$, seed $s$, optional max_iters
**Output:** reduced fragments $\{\mathcal{G}_{F'}\}$
init RNG with $s$
cuts $\leftarrow \mathcal{C}$                                                                  // start from all torsion cuts
$\mathcal{S} \leftarrow$ RecMerge$(\text{mol}, \text{cuts}, \text{sorted}(\mathcal{C}), 0)$                          // valid cut-sets
sample cuts uniformly from $\mathcal{S}$                                        // stochastic augmentation
frags $\leftarrow$ safe_fragment$(\text{mol}, \text{cuts})$                                                   // Sanitize
**foreach** $f \in$ frags **do**
     $f \leftarrow$ remove_overconstrained$(f)$
**return** frags

**Function** RecMerge $(\text{mol}, \text{cur}, \text{cand}, \text{start})$
     $\mathcal{V} \leftarrow \{\text{frozenset}(\text{cur})\}$
     **for** $i \leftarrow$ start **to** $|\text{cand}| - 1$ **do**
         $b \leftarrow \text{cand}[i]$
         **if** $b \in$ cur **then**
             new $\leftarrow$ cur $\setminus \{b\}$
             **if** ValidState $(\text{mol}, \text{new})$ **then**
                 add frozenset(new) to $\mathcal{V}$
                 $\mathcal{V} \leftarrow \mathcal{V} \cup$ RecMerge$(\text{mol}, \text{new}, \text{cand}, i+1)$
     **return** $\mathcal{V}$

**Function** ValidState $(\text{mol}, \text{cuts})$
     parts $\leftarrow$ safe_fragment$(\text{mol}, \text{cuts})$
     **foreach** $p \in$ parts **do**
         $p_{\text{clean}} \leftarrow$ remove_overconstrained$(p)$
         **if** $|p_{clean}| > 3$ **and** detect_torsions$(p_{clean})$ **then**
             **return false**
     **return true**

---

**Defining fragments.** For a sample $(\mathcal{G}_{\text{ligand}}, \mathcal{G}_{\text{protein}}) \sim p_{\text{data}}$, we generate a conformer via RDKit's ETKDGv3 (Landrum, 2025) from $\mathcal{G}_{\text{ligand}}^{2D}$, align to $\mathbf{x}_b$ via our Kabsch aligment procedure and split the fragments selected by FR3D, yielding $m$ independently SE(3) bodies. We initialise our space $\mathbf{z}^{(0)} = (\mathbf{p}^{(0)}, \mathbf{R}^{(0)}) \in \text{SE}(3)^m$ by trivially defining $\mathbf{p}^{(0)}$ as the CoMs of each fragment as described in 2.1. This canonical choice ensures our coordinate-frame-invariant pseudo-force construction outlined in G.4 is valid. Furthermore, since our training objective is invariant to the relative orientation of local coordinates $\tilde{\mathbf{x}}_F$, we initialise $\mathbf{R}^{(0)}$ at the identity.

**Data scaling.** We scale the global coordinate system by a factor of 2.7. This factor is derived from the isotropic dimensional scale $|\mathcal{M}_b|$, computed as the average standard deviation (in Å) of fragment CoMs relative to the pocket CoM across all ligands in the training dataset. The total normalisation of the coordinate system is therefore $\mathbf{x} \leftarrow \frac{\mathbf{x} - \overline{x}_{\text{pocket}}}{|\mathcal{M}_b|}, \mathbf{y} \leftarrow \frac{\mathbf{y} - \overline{x}_{\text{pocket}}}{|\mathcal{M}_b|}$. In effect, this scaling ensures that the stationary distribution (prior) $\mathbf{z}_T$ provides a faithful representation of the spread of states in $\mathbf{z}_0$, consistent with the variance-preserving form of the forward kernel on $\mathbb{R}^{m \times 3}$.

### E.2 ACCELERATED CACHING

We accelerate sampling of complex molecular graphs by creating a cached sampling algorithm as shown in Algorithm 2. Effectively, we use FR3D to process particular protein-ligand complexes individually to create a cache of size $C$ and then sample $R$ random batches of size $B$. Each time we sample a particular complex from the dataset we sample time $t \in [0, 1]$ independently such that we re-use a loaded complex $R$ times, artificially reducing the computational bottleneck by a factor $R/C$. By default we set $R$ to 8 and $C$ to 2, such that we speed up processing time by a factor of 4. We pick a relatively small factor to ensure we can sample across multiple noise levels with

sufficient diversity in $\mathcal{G}_{\text{dock}}$ such that we limit the risk of non-independent and identically distributed (iid) sampling.

---

**Algorithm 2:** CachedSampling — Stochastic cached batching with recycling

---

**Input:** Dataset $\mathcal{D}$, cache size $C$, batch size $B$, cycles $R$, seed $s$
**Output:** Batch of size $B$; each datapoint constructed by FRED()
Initialize random seed $s$
**while** *true* **do**
    $cache \leftarrow [\,]$
    **while** $|cache| < C$ **do**
        **if** *no more data* **then**
            **if** $|cache| = 0$ **then**
                **return**
            **break**
        $i \leftarrow$ next index
        **if** $x \leftarrow$ FRED$(i)$ *is invalid* **then** continue
        append(clone($x$), $cache$)
    **for** $r = 1$ **to** $R$ **do**
        shuffle($cache$; seed $= s$)
        **for** $i = 0, \ldots, B$ **do**
            yield $cache[i]$

---

### E.3 TRAINING

We build SIGMADOCK in Python 3.12, employing PyTorch, PyTorch Geometric and PyTorch Lightning as our core deep learning libraries. For our optimiser, we employ AdamW (Loshchilov & Hutter, 2019) with an L2 weight penalty of 0.1. We train for a maximum of 256 epochs using a batch size of 32 and apply early stopping with a patience of 50 epochs. We also use an Exponential Moving Average (EMA) to define our final parameters, using a weight decay of 0.999, as is common practice in diffusion models. We use linear warmup with cosine annealing as our learning rate scheduler strategy, with an initial (cold) learning rate of $1 \times 10^{-6}$, a maximum learning rate of $1 \times 10^{-4}$ warmed up over the first 16 epochs, and a final annealed learning rate of $1 \times 10^{-5}$. We train SIGMADOCK using 4 NVIDIA-A100s (80Gb) via distributed-data-parallel for 4 days, until convergence.

## F INFERENCE DETAILS

In this section, we present our sampling procedure for SIGMADOCK and how we filter generated samples to return the best samples for evaluation.

### F.1 SAMPLING

We outline our sampling procedure in Algorithm 3 below which supports both deterministic and stochastic generation. We include the use of noise annealing, as we find that it can help improve performance for stochastic generation. For our time discretisation between $t_{\max}$ and $t_{\min}$, we use the schedule from Karras et al. (2022). For $k = 0, \ldots, N - 1$, the $k$-th timestep is defined as

$$t_k = \left( t_{\max}^{1/\rho} + \frac{k}{N_{\text{steps}} - 1} \left( t_{\min}^{1/\rho} - t_{\max}^{1/\rho} \right) \right)^{\rho},$$

where $N_{\text{steps}}$ is the number of discretisation steps and $\rho > 0$ controls the spacing. When $\rho = 1$, this reduces to uniform discretisation; larger values of $\rho$ concentrate timesteps near $t_{\min}$, while smaller values spread them more evenly across the interval. By default, we use $N_{\text{steps}} = 25$, $t_{\min} = 0.002$, $t_{\max} = 1$, $\rho = 3$. For our noise annealing, we use the same schedule with $\gamma_{\min} = 0, \gamma_{\max} = 0.5$ and $\rho = 2$.

---

**Algorithm 3:** SIGMADOCK sampling

**Input:** Score model $s_\theta$, docking information $\mathcal{G}_{\text{dock}}$, time steps $0 \leq t_{\min} = t_0 < \ldots < t_n = 1$, noise annealing schedule $\{\gamma_i\}_{i=0}^n$ where $\gamma_i > 0$

**Output:** Predicted ligand coordinates $\mathbf{x}$

Sample $\mathbf{z}^{(1)} = (\mathbf{p}^{(1)}, \mathbf{R}^{(1)}) \sim q(\mathbf{z}) = \mathcal{N}(0, \mathbf{I}) \otimes \mathcal{U}_{\text{SO}(3)^m}$    // sample from the prior

**for** $i \leftarrow n$ **to** 1 **do**
   $\Delta t \leftarrow |t_i - t_{i-1}|$
   $[\hat{\mathbf{s}}^p, \hat{\mathbf{s}}^R] \leftarrow s_\theta(\mathbf{z}^{(t_i)}, t_i, \mathcal{G}_{\text{dock}})$        // compute scores
   $\mathbf{p}^{(t_{i-1})} \leftarrow \texttt{ReverseTranslations}(\mathbf{p}^{(t_i)}, \hat{\mathbf{s}}^p, t_i, \Delta t, \gamma_i)$
   $\mathbf{R}^{(t_{i-1})} \leftarrow \texttt{ReverseRotations}(\mathbf{R}^{(t_i)}, \hat{\mathbf{s}}^R, t_i, \Delta t, \gamma_i)$
   $\mathbf{z}^{(t_{i-1})} \leftarrow (\mathbf{p}^{(t_{i-1})}, \mathbf{R}^{(t_{i-1})})$

$\mathbf{x} \leftarrow \varphi(\mathbf{z}^{(t_{\min})})$
**return** $\mathbf{x}$

**Function** $\texttt{ReverseTranslations}(\mathbf{p}, \mathbf{s}^p, t, \Delta t, \gamma)$
   $\Delta \mathbf{p} \leftarrow \Delta t \cdot (\frac{1}{2}\beta(t)\mathbf{p} + \beta(t)\mathbf{s}^p)$
   $\Delta \epsilon \leftarrow \gamma \cdot \sqrt{\Delta t \cdot \beta(t)} \cdot \mathcal{N}(0, \mathbf{I})$
   $\mathbf{p}' \leftarrow \mathbf{p} + \Delta \mathbf{p} + \Delta \epsilon$
   **return** $\mathbf{p}'$

**Function** $\texttt{ReverseRotations}(\mathbf{R}, \mathbf{s}^R, t, \Delta t, \gamma)$
   $\mathbf{s}^R \leftarrow \mathbf{R}^{-1} \cdot \mathbf{s}^R$        // convert to Lie algebra
   $\mathbf{s}^R \leftarrow [\mathbf{s}^R]_{\mathbb{R}^3}$        // convert to $\mathbb{R}^3$ basis
   $g \leftarrow \frac{d}{ds}\sigma^2(s)|_{s=t}$
   $\Delta \mathbf{R} \leftarrow \Delta t \cdot g \cdot \mathbf{s}^R$
   $\Delta \epsilon \leftarrow \gamma \cdot \sqrt{\Delta t \cdot g} \cdot \mathcal{N}(0, \mathbf{I})$
   $\mathbf{R}' \leftarrow \mathbf{R} \cdot \exp([\Delta R + \Delta \epsilon]_\times)$        // convert back to SO(3)
   **return** $\mathbf{R}'$

---

**Computational cost of sampling.** Using an NVIDIA-A40 GPU for sampling, we achieve a generation speed (average runtime across the PoseBusters set) of 0.57s/mol per seed when using 20 discretisation steps and a batch size of 64. Depending on the number of seeds ($N_{\text{seed}}$) used in our SIGMADOCK ($N_{\text{seed}}$) parametrisation via our heuristic outlined in F.2, increasing $N_{\text{seed}}$ up-scales the runtime linearly (unless a distributed system was employed). Sampling speed is also linearly dependent on the number of reverse diffusion steps. However, we find diminishing returns with more

than 20-30 steps. We argue SIGMADOCK is highly computationally efficient in comparison to other deep learning methods, taking between 5.7-22.8s/mol for $N_{\text{seed}} = 10, 40$ respectively. See Table 2 in (Jiang et al., 2025) for a comparative analysis of other deep learning tools and their relative runtimes. Notably, AF3 (Abramson et al., 2024) is reported to have an average runtime of 16min/mol and DiffDock (Corso et al., 2022) is reported to have an average runtime of 72s/mol (this includes the respective number of seeds required by each method).

## F.2 CONFIDENCE MODEL

A common requirement of previous work in generative molecular docking is the use of a separately trained confidence model. This is used to filter a batch of generated samples, from ranking samples by their computed confidence value, in order to find the best generated poses and discard samples with poor chemical plausibility. However, this is a significant additional computational burden, from both the training of the confidence model and the need to apply this to every generated sample. For instance, Corso et al. (2022) evaluates the performance of their model from a batch of 40 generated samples for each protein-ligand pair, and Prat et al. (2024) creates a scoring function to filter a large pool of physics-based docked poses.

In contrast, due to the inductive biases that SIGMADOCK leverages, we find that generated samples from SIGMADOCK tend to be reliably chemically plausible. Hence, we *do not* require the use of a trained confidence mode, and more importantly, we *do not* employ energy minimisation techniques which alter the final coordinates of our predicted $\hat{\mathbf{x}}_b$. Specifically, using the force fields (Maier et al., 2015; Boothroyd et al., 2023) implemented in OpenMM (Eastman et al., 2017) for energy evaluation, Buttenschoen et al. (2024) shows the relevance of energy minimisation in previous deep learning approaches.

Instead, we use the simple and cheap (trivially parallel) heuristic of evaluating the binding energy of the protein-ligand system at a generated docked pose as our "confidence model" with Vinardo (Quiroga & Villarreal, 2016), where we take lower energies as corresponding to better samples. During sampling, we compute a total score per sample which also penalises for stereochemical inconsistencies including bond lengths, bond angles, tetrahedral chirality mismatch, internal steric clash, and minimum distance to the protein.

Formally, let $b_i \in \mathbb{R}$ denote the binding energy of sample $i$, for $i \in \{1, \cdots, N_{\text{seeds}}\}$ and let $p_i \in [0, 1]$ represent the average PoseBusters validity checks for the stereochemical properties specified above (1 = all checks are valid, 0 = all checks are invalid). The mixed score for a set of samples is defined as

$$s_i = - b_i \, p_i^{\beta},$$

where $\beta$ controls the strength of the total PoseBusters penalty (set to 4). Higher values of $s_i$ correspond to higher-confidence poses. We then use these scores to rank the sampled poses across $N_{\text{seeds}}$ when computing Top-$k$ metrics.

We believe our approach stands as a more fair test since relying on a different method to correct the coordinates of sampled bound states is not generally a form which agrees with the maximum likelihood objective of score models during sampling, and can be tuned at test time to artificially generate better performance metrics.

# G ARCHITECTURE

In this section, we provide an overview of our score architecture $s_\theta$ which takes the inputs $\mathbf{z} \in \mathrm{SE}(3)^m$, $t \in [0, 1]$ and $\mathcal{G}_{\mathrm{dock}} = (\mathcal{G}_{\mathrm{ligand}}^{\mathrm{2D}}, \{\mathcal{G}_{F_i}\}_{i=1}^m, \mathcal{G}_{\mathrm{protein}})$. We assume that the docking conditioning information $\mathcal{G}_{\mathrm{dock}}$ has already been preprocessed by the steps outlined in Section E.1.

## G.1 PROTEIN-LIGAND FEATURISATION

Here we describe the structure of $\mathcal{G}_{\mathrm{ligand}}^{\mathrm{2D}}$, $\{\mathcal{G}_{F_i}\}_{i=1}^m$, and $\mathcal{G}_{\mathrm{protein}}$, and how we extract relevant structural stereochemical information therein.

**Chemical features.** We featurise the atoms and bonds according to their chemical nature. Specifically, in addition to the coordinates, we extract the following atomic (node-wise) features:

- Atomic Number $\in \mathbb{Z}^+$.
- Degree $\in \mathbb{Z}^+$, number of adjacent atoms.
- Charge $\in \{0, 1, 2\}$ representing the formal charge: 0 if charge is neutral, 1 if charge is negative, 2 otherwise.
- Hybridicity $\in \{0, \cdots, 5\}$ representing the hybridisation type (Other, SP, SP2, SP3, SP3D, SP3D2) respectively.
- Implicit Valence $\in \mathbb{Z}$.
- Explicit Valence $\in \mathbb{Z}$.
- Is Aromatic $\in \{0, 1\}$. Boolean value determining if atom belongs to an aromatic chain.
- Is Ring $\in \{0, 1\}$. Boolean value determining if the atom belongs to a ring/cycle structure.
- Num Rings $\in \mathbb{Z}^+$. Number of ring systems the atom belongs to.
- Chirality Tag $\in \{0, 1, 2\}$. R(1)-S(2) featurisation of the chirality if atom is chiral, else 0.

Similarly, we extract the following bond (edge-wise) features:

- Bond Type $\in \{0, \cdots, 4\}$: undefined, single, double, triple, or aromatic bond.
- Is Conjugated $\in \{0, 1\}$. Boolean determining if the bond is a conjugated bond.
- Is In Ring $\in \{0, 1\}$. Boolean determining if bond is in a ring system.
- Stereo Feature $\in \{0, \cdots, 3\}$. Mapping bond stereo information (None, Any, Cis, Trans).

**Protein-specific features.** To further extract relevant information from the protein atoms we add an additional feature, for all atoms in the protein, flagging the amino acid it belongs to (from the standard set of 20 amino acids) and passing this through a learnable embedding function. We also create virtual nodes at the alpha-carbon $C_\alpha$ which have significant importance in our topological design. Furthermore, to remove redundancies and prevent over-smoothing across the protein residue, we add a depth-from-$C_\alpha$ distance encoding over the atoms in the chain.

**Fragment-specific features.** Using FR3D and the featurisation procedure outlined above, we define our rigid body fragments from a conformation (aligned bound pose during training, random conformer during sampling). On top of the chemically featurised local geometry obtained here, we add a set of virtual nodes in each fragment (denoted as $V_F$) defined by the following logic: if the fragment has no rings, place the virtual node in the center of mass of the fragment. Alternatively, place multiple virtual nodes at the center of each ring.

## G.2 INPUT GRAPH

We clarify that in the main body we use the notation $\mathcal{G}_{\mathrm{dock}}$ as our representation of the protein-ligand complex conditioning, and in this section we specify how $\mathcal{G}_{\mathrm{dock}}$ is processed into the input graph $\mathcal{G}_{\mathrm{input}}$ that we passing into our score architecture. We note that while our diffusion process and training objective is defined on the Lie group $\mathrm{SE}(3)^m$, following previous work, we design our score model $s_\theta$ to operate on the 3D coordinates of the protein-ligand system, defined by $\mathcal{G}_{\mathrm{protein}}$ and $\varphi(\mathbf{z})$, instead of on the group elements $\mathbf{z}$ directly.

The input graph $\mathcal{G}_{\text{input}}$ can be decomposed into the *global topology*, which does not change throughout the diffusion path, *i.e.* is immutable across all $p_t(\mathbf{z}(t), t | \mathcal{G}_{\text{dock}})$, $t \in [0, 1]$, and the *transient topology*, which is defined from the current fragment coordinates $\mathbf{x}(t) = \varphi(\mathbf{z}(t))$ and $\mathcal{G}_{\text{protein}}$. We make the important distinction: In our *global topology*, we are not assuming that the entries have fixed coordinates, but instead that the edge indices and edge types are fixed throughout. We provide the analogy of a spider's web, which you can stretch out (altering the coordinates) but will still maintain the same nodes and edges.

**Global topology.** The core representation of the global topology consists of the rigid fragments and static protein structure absorbed in $\mathcal{G}_{\text{dock}}$. We extend our global topology by creating edges across all virtual nodes:

- Cartesian product across all $C_\alpha$ under an 18Å distance cut-off,
- Cartesian product across all $(V_{F_i}, V_{F_j})$, $i \neq j$,
- Cartesian product across all $(V_F, C_\alpha)$,

forming a hierarchical and well-structured global message passing mechanism. We also add triangulation edges connecting fragment atoms together, as illustrated in Figure 9.

**Transient topology.** Our transient topology is dynamically defined and includes all protein-ligand interactions edges under a pre-determined cutoff of 4Å. We keep this cutoff relatively low to focus on important protein-fragment interactions which could otherwise be missed by the message passing mechanism over the global topology.

**Node encoding.** To distinguish between protein and ligand nodes we add further categorical features to the aforementioned list of chemical features. Namely, we add a node entity attribute which labels the node as either one of:

- Ligand (fragment) Atom
- Ligand (fragment) Anchor (beginning of torsional bond)
- Ligand (fragment) Dummy (inclusion of end of torsional bond)
- Fragment Virtual ($V_F$)
- Protein Atom
- Protein Virtual ($C_\alpha$)

For all nodes in our graph we encode time through a Fourier embedding as used in (Song et al., 2020). We then define an `AtomDiffusionEncoder` which processes the categorical features of each node (including node entity and chemical features) together with the time embedding to produce a feature embedding of dimension $f_{\text{node}} = 128$. Distinctively, the feature embedding of $V_{F_i}$ is computed as the average of the feature embeddings extracted from each individual atom connected to $V_{F_i}$. Note we connect to $V_{F_i}$ all atoms in fragment $F_i$ unless it has multiple rings, whereby we connect $V_{F_i}$ to all atoms in each ring system. This reduces over-squashing $V_{F_i}$ in large fragments.

**Edge encoding.** Similarly, we build a node entity attribute which labels edges as either one of:

- Ligand Bond
- Protein Bond
- Torsional Bond (across $(F_i, F_j)$, $i \neq j$)
- Fragment Triangulation
- Protein-Ligand Interaction
- Fragment Virtual to fragment atom $V_{F_i}$ to Fragment Atom $F_i$
- Fragment-Virtual to Protein-Virtual ($V_F - C_\alpha$)
- Fragment Virtual-Virtual ($V_F - V_F$)
- Protein Virtual-Virtual ($C_\alpha - C_\alpha$)

For all edges, we define a distance embedding function using an adapted Fourier distance encoder, `TotalFourierSmearing`, for edges defined as part of the *global topology*, and a Bessel basis

with smooth polynomial cutoff, `RadialEmbeddingBlock`, for edges defined as part of the *transient topology*. All edges compute distance encodings based on the true distance evaluated from $(\mathbf{x}_t, \mathbf{y}_t)$ except the fragment triangulation edges. For these edges we define the distance as an offset from equilibrium: All distances are computed from the current atomic positions $(\mathbf{x}_t, \mathbf{y}_t)$, with the exception of fragment triangulation edges. For these edges, we instead define the distance as an offset from the equilibrium length:

$$d_{ij}(t) = \big|\, \|\mathbf{x}_i(t) - \mathbf{x}_j(t)\| - \|\mathbf{x}_i(0) - \mathbf{x}_j(0)\| \,\big|,$$

where $\mathbf{x}_i(0), \mathbf{x}_j(0)$ denote the equilibrium reference positions extracted from a random conformer sampled from RDKit.

An example of $G_{\text{input}}$ (excluding interaction and virtual edges for clarity) is shown in Figure 9.

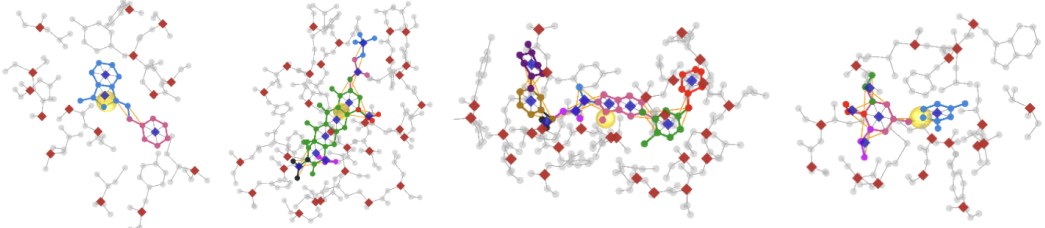

Figure 9: Example of SIGMADOCK representing the bound pose as fragments, (color coded) on the protein pocket (gray) for complexes with PDB-LIG codes 5SAK-ZRY, 7NR8-UOE, 6YR2-T1C, 6XG5-TOP respectively. Virtual nodes (blue diamonds for each fragment, red diamonds for protein $C_\alpha$) are depicted. Fragment-protein interactions, $C_\alpha - C_\alpha$, and $C_\alpha - V_F$ edges are hidden for clarity. We keep the connections between fragment virtual nodes and their corresponding fragment atoms. Triangulation edges are shown in orange. The pocket CoM is depicted via the golden sphere with a radius of 1Å. Note how the pocket CoM is not always indicative of the bound pose CoM, offering lower bias during sampling.

### G.3 BACKBONE MODEL

Given our input graph $\mathcal{G}_{\text{input}}$ constructed as described above, we pass this into our backbone model. This is a modified[14] version of the EquiformerV2 (Liao et al., 2023) architecture where the output head predicts atom-wise forces. The modification that we make to the EquiformerV2 architecture is to remove the bias terms within the MLPs (*i.e.* `RadialFunction`) that process edge features for dynamic edges in $\mathcal{G}_{\text{input}}$ (*i.e.* edges which we include if the neighbouring nodes are within some cutoff distance); we also use a different MLP for each edge type. We recall from above, that the edges features for these dynamic edges are computed, from passing the distance between the neighbouring nodes through a Bessel basis expansion and a polynomial envelop for the cutoff distance, so that these features smoothly decay to zero as the cutoff distance is approached. Therefore, the effect of removing the bias terms is to ensure that the messages along these dynamic edges, computed within the equivariant graph attention blocks (as well as the edge degree embedder), also smoothly decay to zero as the cutoff distance is approached. In addition, our radial basis function has the desired property of ensuring the gradient norms smoothly decay to zero as the edge distance approaches the cutoff. This is to ensure smoothness in the prediction of forces as we perturb $\mathbf{z}$ which could cause sudden changes in the topology of the input graph $\mathcal{G}_{\text{input}}$ from dynamic edges being added/removed. All our ablations and main results are run with the same number of trainable parameters (14.9M).

### G.4 PREDICTION HEAD

To ensure invariance with respect to the choice of local coordinates, we adapt the $\mathrm{SO}(3)$-equivariant prediction head from Jin et al. (2023); Guan et al. (2023) which utilises the Newton-Euler equations from rigid-body mechanics. In particular, for each fragment $\mathcal{G}_F$ with global input coordinates $\mathbf{x}_F = (x_{F,1}, \ldots, x_{F,|\mathcal{G}_F|}) \in \mathbb{R}^{|\mathcal{G}_F| \times 3}$ and $\mathrm{SE}(3)$ coordinates $(p_F, R_F)$, let

---

[14]Although strictly within the architecture module, we consider the featurisation of the nodes and edges in our input graph, which we previously describe, as separate from the core EquiformerV2 architecture.

$\mathbf{f} = (f_1, \ldots, f_{|\mathcal{G}_F|}) \in \mathbb{R}^{|\mathcal{G}_F| \times 3}$ where $f_i$ denotes the output from our backbone model for the $i$th node in the fragment; the total force $\mathbf{F}_F$ and the torque $\boldsymbol{\tau}_F$ on $\mathcal{G}_F$ are defined as

$$\mathbf{F}_F = \sum_{i=1}^{|\mathcal{G}_F|} f_i, \qquad \boldsymbol{\tau}_F = \sum_{i=1}^{|\mathcal{G}_F|} (x_{F,i} - p_F) \times f_i.$$

We then predict the score function for $\mathcal{G}_F$ with the score components $[\mathbf{s}_\theta^p, \mathbf{s}_\theta^R]$ defined by:

$$\mathbf{s}_\theta^p = \frac{1}{\sqrt{1 - \alpha_t}} \cdot \frac{1}{|\mathcal{G}_F|} \mathbf{F}_F, \quad \mathbf{s}_\theta^R = -\frac{\partial_\omega f_0(\omega, \sigma(t))}{\omega f_0(\omega, \sigma(t))} [\mathbf{I}_F^{-1} \boldsymbol{\tau}_F]_\times R_F.$$

where $\omega = \left\| \mathbf{I}_F^{-1} \boldsymbol{\tau}_F \right\|_2$ and $\mathbf{I}_F = \sum_{i=1}^{|\mathcal{G}_F|} \|x_{F,i} - p_F\|^2 \mathbf{I} - (x_{F,i} - p_F)(x_{F,i} - p_F)^\top$ denotes the inertia matrix of the fragment. We see that $\mathbf{s}_\theta^p$ is essentially parametrised for $\epsilon$-prediction and that $\mathbf{s}_\theta^R$ is constructed to match the form of the conditional score; this is done to improve the conditioning of the score matching loss.

# H SYMMETRIES

As part of SIGMADOCK, we are concerned with two main symmetries of our model. Firstly, we note that an issue of working with fragments naively is that due to their heterogenous construction (i.e. fragments can have various different topologies and number of atoms etc.), we can not define a canonnical orientation for their local coordinates. This is not an issue with other fragment-based models, such as AlphaFold2 (Jumper et al., 2021), as in their setting, all fragments take the same form (e.g. protein backbone frames have a canonical axis defined by the $(N - C_\alpha - C)$ atoms). Formally, consider a fragment $\mathcal{G}_F$ with some choice of local coordinates $\tilde{\mathbf{x}}_F$. For any choice of $R_s \in$ SO(3), we can define another valid choice of local coordinates given by $\tilde{\mathbf{x}}'_F = R_s \cdot \tilde{\mathbf{x}}_F$; we denote the fragment defined by this alternative choice by $\mathcal{G}'_F$. Therefore, if we observe the global coordinates of the fragment are $\mathbf{x}_F = p_F + R_F \cdot \tilde{\mathbf{x}}_F$, then under the fragment $\mathcal{G}_F$, this pose is parametrised by $(p_F, R_F)$, whereas under the fragment $\mathcal{G}'_F$, this pose is parametrised by $(p_F, R_F R_s^\top)$ instead, since $\tilde{\mathbf{x}}_F = p_F + R_F R_s^\top \cdot \tilde{\mathbf{x}}'_F$. To prevent issues from the orientation of local coordinates not being well-defined, we have designed SIGMADOCK to ensure that our training objective and sampling procedure are invariant with respect to the choice of orientations for local coordinates.

Secondly, we note the kernel $p_\theta(\mathbf{z}|\mathcal{G}_{\text{dock}})$ defined by sampling from SIGMADOCK should ideally be *stochastically equivariant* (Bloem-Reddy & Teh, 2020; Cornish, 2024; Zhang et al., 2024) with respect to SO(3), i.e.

$$p_\theta(d\mathbf{z}|R_0 \cdot \mathcal{G}_{\text{dock}}) = R_0 \cdot p_\theta(d\mathbf{z}|\mathcal{G}_{\text{dock}}), \qquad \forall R_0 \in \text{SO}(3),$$

where $R_0 \cdot \mathcal{G}_{\text{dock}}$ should be understood as applying the rotation $R_0$ to the coordinates $\mathbf{y}$ of $\mathcal{G}_{\text{protein}} \in \mathcal{G}_{\text{dock}}$, and the right-hand side denotes the push-forward under the mapping given by applying $R_0$ (where we define the action $R_0 \cdot \mathbf{z} = R_0 \cdot (p_{F_i}, R_{F_i})_{i=1}^m = (R_0 \cdot p_{F_i}, R_0 R_{F_i})_{i=1}^m$). In random variable notation, this corresponds to the statement saying that the sample $\mathbf{z}' = R_0 \cdot \mathbf{z}$ where $\mathbf{z} \sim p_\theta(\mathbf{z}|\mathcal{G}_{\text{dock}})$ follows the same distribution as $\hat{\mathbf{z}} \sim p_\theta(\mathbf{z}|R_0 \cdot \mathcal{G}_{\text{dock}})$. Intuitively, this means that the distribution $p_\theta(\mathbf{z}|\mathcal{G}_{\text{dock}})$ of fragments (and hence, the ligand, as it is easy to see that $\varphi(R_0 \cdot \mathbf{z}) = R_0 \cdot \varphi(\mathbf{z})$) should be transformed in a consistent way as we change the orientation of the protein pocket that SIGMADOCK is conditioned on. To enforce this symmetry, we have designed SIGMADOCK to use an SO(3)-equivariant architecture for our score model.

We note that stochastic equivariance is more commonly presented in the literature (Hoogeboom et al., 2022) in terms of conditional densities $p(x|y)$ which obey the condition $p(g \cdot x|g \cdot y)$ for all $g \in G$ where $G$ is some group under consideration. However, this definition implicitly assumes that the Jacobian of the group action has unit determinant and does not account for the general case presented in Bloem-Reddy & Teh (2020); Cornish (2024). Further, we note that it is surprising to us that, to the best of our knowledge, this particular symmetry has not been remarked on (or proved to hold as we do later on) in the literature on diffusion-based molecular docking, as this condition provides justification for using an equivariant architecture for the score model. Specifically, we find prior work appears to use such architectures without analysing the properties they induce on the diffusion model and sampling.

To show that SIGMADOCK does indeed satisfy the symmetry requirements that we have outlined above, and stated in Theorem 2, we provide a proof of this fact below in Appendix H.1.

## H.1 PROOF OF THEOREM 2

*Proof.* We divide the proof into three sections: (i) proving invariance of our training objective; (ii) proving invariance of our sampling procedure; and (iii) proving $p_\theta(\mathbf{z}|\mathcal{G}_{\text{dock}})$ is stochastically equivariant with respect to SO(3).

**Invariance of training objective.** Consider two different fragmentations $\{\mathcal{G}_{F_i}\}_{i=1}^m$ and $\{\mathcal{G}'_{F_i}\}_{i=1}^m$ of a ligand with different choices for the orientations of local coordinates—i.e. $\mathcal{G}_F$ has local coordinates $\tilde{\mathbf{x}}_F$ and $\mathcal{G}'_F$ has local coordinates $\tilde{\mathbf{x}}'_F = R_{s,F} \cdot \tilde{\mathbf{x}}_F$ where $R_{s,F} \in$ SO(3) for $F \in \{F_i\}_{i=1}^m$. Then the global 3D coordinates $\mathbf{x}$ of the ligand has different representations for each choice of fragmentations. However, no matter the choice of fragmentation, the backbone of our score model $s_\theta$ (with $\mathcal{G}_{\text{dock}}$ and $\mathcal{G}'_{\text{dock}}$ being the corresponding docking information formed the two choices) will output forces for each fragment that are independent of this, as we have designed the backbone to operate over the global 3D coordinates of our protein-ligand system, i.e. $\mathbf{x} = \varphi(\mathbf{z})$ and $\mathbf{y}$.

Next, we focus on the rotational scores for a single fragment with different orientations: $\mathcal{G}_F$ and $\mathcal{G}'_F$. We let $(p_F^{(0)}, R_F^{(0)})$ denote the SE(3) parametrisation at $t = 0$ for $\mathcal{G}_F$, therefore $(p_F^{(0)}, R_F^{(0)} R_{s,F}^\top)$ is the parametrisation under $\mathcal{G}'_F$. Therefore, the sampling distribution for training is given by $R_F^{(t)} \sim \mathcal{IG}_{\mathrm{SO}(3)}(R_F^{(t)}; R_F^{(0)}, \sigma^2(t))$ for $\mathcal{G}_F$ and is given by $R_F^{'(t)} \sim \mathcal{IG}_{\mathrm{SO}(3)}(R_F^{(t)}; R_F^{(0)} R_{s,F}^\top, \sigma^2(t))$ for $\mathcal{G}'_F$. We note that in random variable notation, we have the expression $R_F^{(t)} = \hat{R} R_F^{(0)}$ where $\hat{R} \sim \mathcal{IG}_{\mathrm{SO}(3)}(\hat{R}, \mathbf{I}, \sigma^2(t))$ and $R_F^{'(t)} = \hat{R} R_F^{(0)} R_{s,F}^\top$. This implies that $R_F^{'(t)} \stackrel{d}{=} R_F^{(t)} R_{s,F}^\top$. Further, under the sampling distribution for training for $\mathcal{G}'_F$, we can express the conditional score in the objective as

$$
\begin{aligned}
\nabla_R \log p_{t|0}(R_F^{'(t)} | R_F^{(0)} R_{s,F}^\top) &= \nabla_R \log p_{t|0}(R_F^{(t)} R_{s,F}^\top | R_F^{(0)} R_{s,F}^\top) \\
&= C(t) R_F^{(t)} R_{s,F}^\top \log\left(R_{s,F}(R_F^{(0)})^\top R_F^{(t)} R_{s,F}^\top\right) \\
&= C(t) R_F^{(t)} R_{s,F}^\top R_{s,F} \log\left((R_F^{(0)})^\top R_F^{(t)}\right) R_{s,F}^\top \\
&= C(t) R_F^{(t)} \log\left((R_F^{(0)})^\top R_F^{(t)}\right) R_{s,F}^\top \\
&= \nabla_R \log p_{t|0}(R_F^{(t)} | R_F^{(0)}) R_{s,F}^\top,
\end{aligned}
$$

where the third equality is due to the trivial identity: $\log(ABA^\top) = A \log(B) A^\top$ for $A, B \in \mathrm{SO}(3)$ and we define

$$
C(t) = \frac{\partial_\omega f_0(\omega(t), \sigma(t))}{w(t) f_0(\omega(t), \sigma(t))},
$$

where $\omega(t) = \omega\left(R_{s,F}(R_F^{(0)})^\top R_F^{(t)} R_{s,F}^\top\right)$ and this equals $\omega\left((R_F^{(0)})^\top R_F^{(t)}\right)$ from the definition of the function $\omega$.

Further, let $\mathbf{z} = ((p_{F_i}, R_{F_i}))_{i=1}^m$ and $\mathbf{z}' = ((p_F, R_{F_i} R_{s,F_i}^\top))_{i=1}^m$ be the parametrisation under $\{\mathcal{G}_{F_i}\}_{i=1}^m$ and $\{\mathcal{G}'_{F_i}\}_{i=1}^m$ respectively for a ligand pose $\mathbf{x}$, then the rotational score for a fragment $\mathcal{G}_F$ has the following relationship

$$
s_\theta^R(\mathbf{z}', t, \mathcal{G}'_{\mathrm{dock}})_F = s_\theta^R(\mathbf{z}, t, \mathcal{G}_{\mathrm{dock}})_F \cdot R_{s,F}^{-1},
$$

due to the fact that the predicted forces for $\mathcal{G}_F, \mathcal{G}'_F$ are the same and the only impact that the different choices of orientation makes is in the rotational prediction head.

Turning our focus to the translational scores, it is easy to see that the definition of the translational components $p_F$ (hence, the sampling for training, the prediction of scores and the training objective for the translational scores) are independent of the choice of orientations as these are disentangled.

Finally, by considering the training objective of the rotational scores under $\mathcal{G}'_F$, we have

$$
\begin{aligned}
&\mathbb{E}\left[\left\| s_\theta^R(\mathbf{z}'^{(t)}, t, \mathcal{G}'_{\mathrm{dock}})_F - \nabla_R \log p_{t|0}(R_F^{(t)} R_{s,F}^\top | R_F^{(0)} R_{s,F}^\top) \right\|_{\mathrm{SO}(3)}^2\right] \\
&= \mathbb{E}\left[\left\| s_\theta^R(\mathbf{z}^{(t)}, t, \mathcal{G}_{\mathrm{dock}})_F \cdot R_{s,F}^\top - \nabla_R \log p_{t|0}(R_F^{(t)} | R_F^{(0)}) \cdot R_{s,F}^\top \right\|_{\mathrm{SO}(3)}^2\right] \\
&= \mathbb{E}\left[\left\| s_\theta^R(\mathbf{z}^{(t)}, t, \mathcal{G}_{\mathrm{dock}})_F - \nabla_R \log p_{t|0}(R_F^{(t)} | R_F^{(0)}) \right\|_{\mathrm{SO}(3)}^2\right],
\end{aligned}
$$

where $\mathbf{z}'^{(t)}$ is formed from sampling from $p_{\mathrm{data}}$, extracting the SE(3) parametrisation under $\{\mathcal{G}'_{F_i}\}_{i=1}^m$ and applying the forward sampling kernel, and the final equality is due to the form of the SO(3) inner product. The above result demonstrates that the the training objective of SIGMADOCK is invariant to the choice of orientations for local coordinates, as the final equality has the form of the training objective of the rotational scores under $\mathcal{G}_F$.

**Invariance of sampling.** Under the same setup as before, let $\mathbf{z}^{(t)} = ((p_{F_i}^{(t)}, R_{F_i}^{(t)}))_{i=1}^m$ and $\mathbf{z}'^{(t)} = ((p_F^{(t)}, R_{F_i}^{(t)} R_{s,F_i}^\top))_{i=1}^m$ be the SE(3) parametrisation of the same ligand pose $\mathbf{x}$ under $\{\mathcal{G}_{F_i}\}_{i=1}^m$ and $\{\mathcal{G}'_{F_i}\}_{i=1}^m$. We consider the sampling step for $\mathbf{z}'^{(t)}$ to the next time step $t' < t$. For a

given fragment $\mathcal{G}'_F$, the update step has the form (for clarity, (WLOG) we set all coefficients used in sampling such as $\Delta t \cdot g$ and $\gamma \cdot \sqrt{\Delta t \cdot g}$ to be one):

$$R_F^{(t)} R_{s,F}^\top \exp\left(R_{s,F}(R_F^{(t)})^\top s_\theta^R(\mathbf{z}'^{(t)}, t, \mathcal{G}'_{\text{dock}})_F + Z\right)$$

$$= R_F^{(t)} R_{s,F}^\top \exp\left(-C(t) R_{s,F}(R_F^{(t)})^\top [\mathbf{I}_F^{-1}\tau_F]_\times R_F^{(t)} R_{s,F}^\top + Z\right)$$

$$= R_F^{(t)} R_{s,F}^\top R_{s,F} \exp\left(-C(t)(R_F^{(t)})^\top [\mathbf{I}_F^{-1}\tau_F]_\times R_F^{(t)} + R_{s,F}^\top Z R_{s,F}\right) R_{s,F}^\top$$

$$= R_F^{(t)} \exp\left((R_F^{(t)})^\top s_\theta(\mathbf{z}^{(t)}, t, \mathcal{G}_{\text{dock}}) + R_{s,F}^\top Z R_{s,F}\right) R_{s,F}^\top,$$

where $Z = \sum_{i=1}^3 \delta_i \mathbf{e}_i$ where $\delta_i \sim \mathcal{N}(0,1)$ are iid and $\mathbf{e}_i$ are the canonical basis of the Lie algebra $\mathfrak{so}(3)$. We can show that

$$[\sum_{i=1}^3 \delta_i R_{s,F}^{-1}\mathbf{e}_i R_{s,F}]_{\mathbb{R}^3} = \sum_{i=1}^3 \delta_i[R_{s,F}^{-1}\mathbf{e}_i R_{s,F}]_{\mathbb{R}^3}$$

$$= \sum_{i=1}^3 \delta_i R_{s,F}^{-1}\hat{\mathbf{e}}_i$$

$$= R_{s,F}^{-1} \cdot \mathcal{N}(0, \mathbf{I}_3)$$

$$\sim \mathcal{N}(0, \mathbf{I}_3)$$

where $\hat{\mathbf{e}}_i$ denotes the standard basis vectors of $\mathbb{R}^3$. This is a consequence of the self-adjointness of SO(3) which implies that $[R \cdot v]_\times = R[v]_\times R^{-1}$ where $v \in \mathbb{R}^3$ and $R$ acts on the vector. Therefore, we have $R_{s,F}^\top Z R_{s,F} \overset{d}{=} Z$. This allows us to conclude that distribution of the update step considered in Euclidean space (via the mapping $\varphi$) under the fragmentation $\{\mathcal{G}'_{F_i}\}_{i=1}^m$ is the same as the update step given by the fragmentation $\{\mathcal{G}_{F_i}\}_{i=1}^m$. Hence, the sampling procedure of SIGMADOCK is invariant with respect to the choice of orientations for local coordinates.

**Proof of stochastic equivariance**   We first recall the following notation $\mathbf{z} = ((p_{F_i}, R_{F_i}))_{i=1}^m$ and $\mathcal{G}_{\text{dock}} = (\mathcal{G}_{\text{ligand}}^{\text{2D}}, \{\mathcal{G}_{F_i}\}_{i=1}^m, \mathcal{G}_{\text{protein}})$, where $\mathcal{G}_{\text{protein}} = (\mathbf{y}, \mathbf{v}_y, \mathbf{b}_y)$. For $R_0 \in$ SO(3), we define the following group actions by $R_0 \cdot \mathbf{z} = ((R_0 \cdot p_{F_i}, R_0 R_{F_i}))_{i=1}^m$ and $R_0 \cdot \mathcal{G}_{\text{dock}} = (\mathcal{G}_{\text{ligand}}^{\text{2D}}, \{\mathcal{G}_{F_i}\}_{i=1}^m, R_0 \cdot \mathcal{G}_{\text{protein}})$ where $R_0 \cdot \mathcal{G}_{\text{protein}} = (R_0 \cdot \mathbf{y}, \mathbf{v}_y, \mathbf{b}_y)$. It is important to note that $R_0 \cdot \mathcal{G}_{\text{dock}}$ only acts on the 3D coordinates $\mathbf{y}$ of the protein pocket, and does not change the local coordinates for fragments $\{\mathcal{G}_{F_i}\}_{i=1}^m$. We note that this group action represents the global rotation of the protein-ligand system defined by $\mathbf{z}$ and $\mathcal{G}_{\text{dock}}$.

Let $p_\theta(\mathbf{z}^{(t)}|\mathcal{G}^{\text{dock}})$ denote the marginal distribution of samples at time $t$ generated by SIGMADOCK under the docking information $\mathcal{G}_{\text{dock}}$. In order to prove stochastic equivariance, we can equivalently show that if $p_\theta(\mathbf{z}^{(t)}|\mathcal{G}_{\text{dock}})$ is stochastically equivariant with respect to the above group action, then this implies that $p_\theta(\mathbf{z}^{(t')}|\mathcal{G}_{\text{dock}})$, where $t' < t$, is also stochastically equivariant. Therefore, as $p_\theta(\mathbf{z}^{(1)}|\mathcal{G}_{\text{dock}})$ is trivially stochastically equivariant, since this corresponds to the prior over $\mathbf{z}$ which is independent of the orientation of $\mathbf{y}$, the desired result follows by induction.

To show the inductive step, we note by basic probability that we have the following expression (with densities considered over the product of Lebesgue and Haar measures):

$$p_\theta(\mathbf{z}^{(t')}|\mathcal{G}_{\text{dock}}) = \int p_\theta(\mathbf{z}^{(t')}|\mathbf{z}^{(t)}, \mathcal{G}_{\text{dock}})p_\theta(\mathbf{z}^{(t)}|\mathcal{G}_{\text{dock}})\mathrm{d}\mathbf{z}^{(t)},$$

where $p_\theta(\mathbf{z}^{(t')}|\mathbf{z}^{(t)}, \mathcal{G}_{\text{dock}})$ denotes the kernel induced by sampling the next time step $t'$ from $\mathbf{z}^{(t)}$. We next consider showing the conditional density formulation of stochastic equivariance (this is valid as the action of rotations has unit Jacobian),

$$p_\theta(R_0 \cdot \mathbf{z}^{(t')}|R_0 \cdot \mathcal{G}_{\text{dock}}) = \int p_\theta(R_0 \cdot \mathbf{z}^{(t')}|\mathbf{z}^{(t)}, R_0 \cdot \mathcal{G}_{\text{dock}})p_\theta(\mathbf{z}^{(t)}|R_0 \cdot \mathcal{G}_{\text{dock}})\mathrm{d}\mathbf{z}^{(t)}$$

$$= \int p_\theta(R_0 \cdot \mathbf{z}^{(t')}|R_0 \cdot \mathbf{z}^{(t)}, R_0 \cdot \mathcal{G}_{\text{dock}})p_\theta(R_0 \cdot \mathbf{z}^{(t)}|R_0 \cdot \mathcal{G}_{\text{dock}})\mathrm{d}\mathbf{z}^{(t)}$$

$$= \int p_\theta(R_0 \cdot \mathbf{z}^{(t')}|R_0 \cdot \mathbf{z}^{(t)}, R_0 \cdot \mathcal{G}_{\text{dock}})p_\theta(\mathbf{z}^{(t)}|\mathcal{G}_{\text{dock}})\mathrm{d}\mathbf{z}^{(t)},$$

where the second equality is due to a change of variables and using the fact that rotations has unit Jacobian, and the third equality is due to our inductive hypothesis.

We can express the conditional density $p_\theta(R_0 \cdot \mathbf{z}^{(t')}|R_0 \cdot \mathbf{z}^{(t)}, R_0 \cdot \mathcal{G}_{\text{dock}})$ in terms of the product of $p_\theta(R_0 \cdot \mathbf{p}^{(t')}|R_0 \cdot \mathbf{z}^{(t)}, R_0 \cdot \mathcal{G}_{\text{dock}})$ and $p_\theta(R_0 \cdot \mathbf{R}^{(t')}|R_0 \cdot \mathbf{z}^{(t)}, R_0 \cdot \mathcal{G}_{\text{dock}})$. For the former, we have (for clarity, (WLOG) we set all constants to one):

$$
\begin{aligned}
p_\theta(R_0 \cdot \mathbf{p}^{(t')}|R_0 \cdot \mathbf{z}^{(t)}, R_0 \cdot \mathcal{G}_{\text{dock}}) &= \mathcal{N}(R_0 \cdot \mathbf{p}^{(t')}; R_0 \cdot \mathbf{p}^{(t)} + s_\theta^p(R_0 \cdot \mathbf{z}^{(t)}, t, R_0 \cdot \mathcal{G}_{\text{dock}}), \mathbf{I}) \\
&= \mathcal{N}(R_0 \cdot \mathbf{p}^{(t')}; R_0 \cdot (\mathbf{p}^{(t)} + s_\theta(\mathbf{z}^{(t)}, t, \mathcal{G}_{\text{dock}})), \mathbf{I}) \\
&= \mathcal{N}(p^{(t')}; \mathbf{p}^{(t)} + s_\theta(\mathbf{z}^{(t)}, t, \mathcal{G}_{\text{dock}}), \mathbf{I}) \\
&= p_\theta(\mathbf{p}^{(t')}|\mathbf{z}^{(t)}, \mathcal{G}_{\text{dock}}),
\end{aligned}
$$

where the second equality uses the fact that the forces predicted by EquiformerV2 are SO(3)-equivariant (we note that this is with respect to the input coordinates which are $R_0 \cdot \mathbf{x}$ and $R_0 \cdot \mathbf{y}$ in this case) and the prediction head for translations is trivially SO(3)-equivariant, and the third equality is due to the Gaussian density being isotropic.

For the latter term, we first consider the random variable representation of $R_F'^{(t')} \sim p_\theta(\cdot|R_0 \cdot \mathbf{z}^{(t)}, R_0 \cdot \mathcal{G}_{\text{dock}})$ for an individual fragment (the distribution on $\mathbf{R}^{(t')}$ factorises over fragments independently). This is given from the update step for sampling (for clarity, (WLOG) we set all constants to one) as

$$
\begin{aligned}
R_F'^{(t')} &= R_0 R_F^{(t)} \exp\left((R_F^{(t)})^\top R_0^\top s_\theta(R_0 \cdot \mathbf{z}^{(t)}, t, R_0 \cdot \mathcal{G}_{\text{dock}}) + Z\right) \\
&= R_0 R_F^{(t)} \exp\left((R_F^{(t)})^\top s_\theta(\mathbf{z}^{(t)}, t, \mathcal{G}_{\text{dock}}) + Z\right),
\end{aligned}
$$

where $Z = \sum_{i=1}^3 \delta_i \mathbf{e}_i$ where $\delta_i \sim \mathcal{N}(0,1)$ are iid. The second equality is using the fact that the forces predicted by EquiformerV2 are SO(3)-equivariant and Guan et al. (2023) shows that $\mathbf{I}_F^{-1}\tau_F$ is SO(3)-equivariant and then applying $[R \cdot v]_\times = R[v]_\times R^{-1}$. Further, we note that the term $R_F^{(t)} \exp\left((R_F^{(t)})^\top s_\theta(\mathbf{z}^{(t)}, t, \mathcal{G}_{\text{dock}}) + Z\right)$ is the random variable representation of $R_F^{(t')} \sim p_\theta(\cdot|\mathbf{z}^{(t)}, \mathcal{G}_{\text{dock}})$. Therefore, we can view $R_F'^{(t')}$ as a change of variables of $R_F^{(t')}$ by the transformation $R_0$. Hence, we can conclude that $p_\theta(R_0 \cdot R_F^{(t')}|R_0 \cdot \mathbf{z}^{(t)}, R_0 \cdot \mathcal{G}_{\text{dock}}) = p_\theta(R_F^{(t')}|\mathbf{z}^{(t)}, \mathcal{G}_{\text{dock}})$ and more generally $p_\theta(R_0 \cdot \mathbf{R}^{(t')}|R_0 \cdot \mathbf{z}^{(t)}, R_0 \cdot \mathcal{G}_{\text{dock}}) = p_\theta(\mathbf{R}^{(t')}|\mathbf{z}^{(t)}, \mathcal{G}_{\text{dock}})$.

Since, we have shown that $p_\theta(R_0 \cdot \mathbf{z}^{(t')}|R_0 \cdot \mathbf{z}^{(t)}, R_0 \cdot \mathcal{G}_{\text{dock}}) = p_\theta(\mathbf{z}^{(t')}|\mathbf{z}^{(t)}, \mathcal{G}_{\text{dock}})$, following from what we had before, we can conclude that

$$
\begin{aligned}
p_\theta(R_0 \cdot \mathbf{z}^{(t')}|R_0 \cdot \mathcal{G}_{\text{dock}}) &= \int p_\theta(R_0 \cdot \mathbf{z}^{(t')}|R_0 \cdot \mathbf{z}^{(t)}, R_0 \cdot \mathcal{G}_{\text{dock}})p_\theta(\mathbf{z}^{(t)}|\mathcal{G}_{\text{dock}})d\mathbf{z}^{(t)} \\
&= \int p_\theta(\mathbf{z}^{(t')}|\mathbf{z}^{(t)}, \mathcal{G}_{\text{dock}})p_\theta(\mathbf{z}^{(t)}|\mathcal{G}_{\text{dock}})d\mathbf{z}^{(t)} \\
&= p_\theta(\mathbf{z}^{(t')}|G_{\text{dock}}).
\end{aligned}
$$

Hence, we can conclude that $p_\theta(\mathbf{z}|\mathcal{G}_{\text{dock}})$ is stochastically equivariant with respect to SO(3).

$\square$

# I EXTENDED RESULTS

## I.1 GENERATED SAMPLES

**Trajectories.** Sample trajectories for four randomised protein-ligand pairs in the Astex diverse set are displayed in Figure 10.

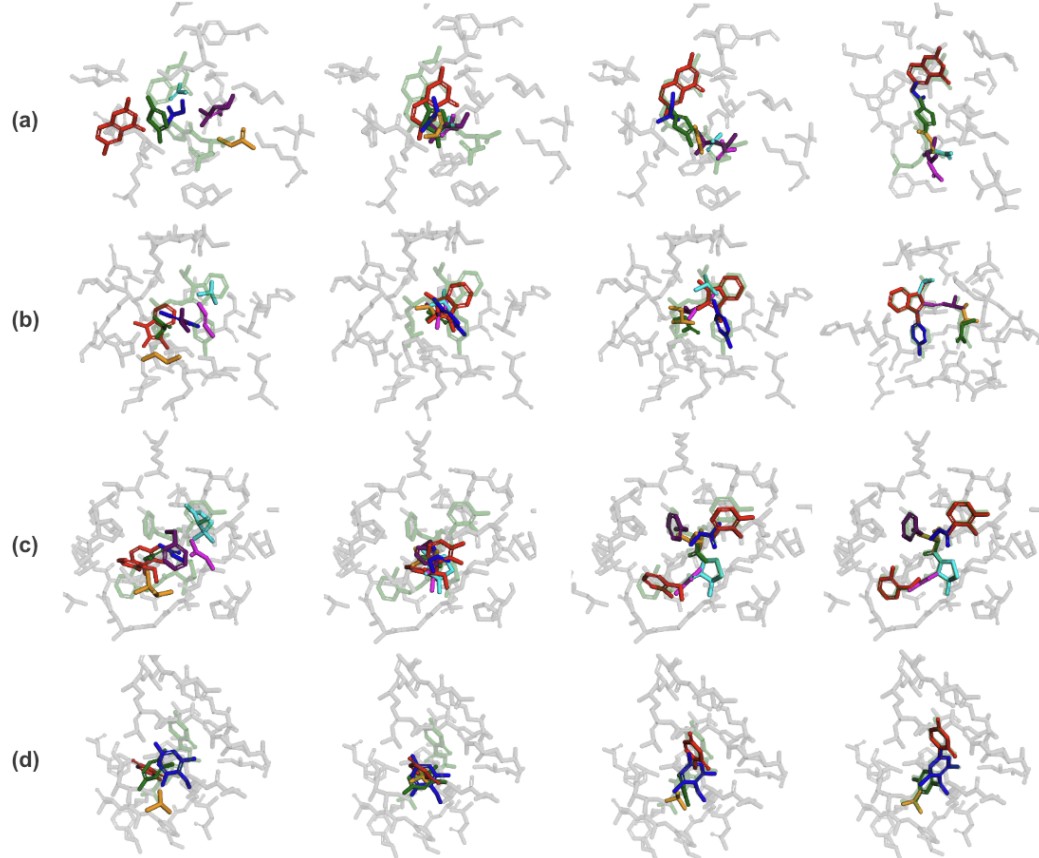

Figure 10: Sample trajectories of protein-ligand pairs 1HVY-D16 (**a**), 1HWI-115 (**b**), 1KZK-JE2 (**c**), and 1N46-PFA (**d**), using the default sampling scheme outlined in F. The respective RMSDs between the sampled and bound pose are 1.2, 0.31, 0.33, 0.88 Å . We illustrate discretisation-steps 0, 3, 10 and 20 (out of 20), where the zeroth step represents samples from the stationary distribution. We show the reference bound pose in light green. For **a** and **b** we tilt the view in the last frame to improve visibility of the fit.

**RMSD is an imperfect metric.** In Figure 11 we show 4 examples of generated poses from the PoseBusters set where the (symmetry-corrected) RMSD between the generated pose and the bound pose is higher than the threshold of 2Å . According to this metric, these are deemed as failed samples. However, we argue that the generated poses seem stereochemically valid, as they recover similar interactions. Namely, in (**a**), ligand O88 is highly symmetric; the generated pose yields very similar interactions and induced fit, even when SIGMADOCK predicts the pseudo-rotated conformation, yet the RMSD is 10.2Å. A similar argument can be made for ligand XN7 in (**b**), where we observe the $(C = O)$ forming the correct interaction (hydrogen bond acceptor) with the protein. In our third example (**c**) we see how the sampled pose very tightly recovers the true pose, yet the non-interacting tail of the ligand containing a cycloproply group is predicted to be elsewhere. However, given the binding pocket is open, this cyclopropyl should freely rotate (molecules are dynamic). This unfairly labels the sampled pose as incorrect, with an RMSD of 4.2Å.

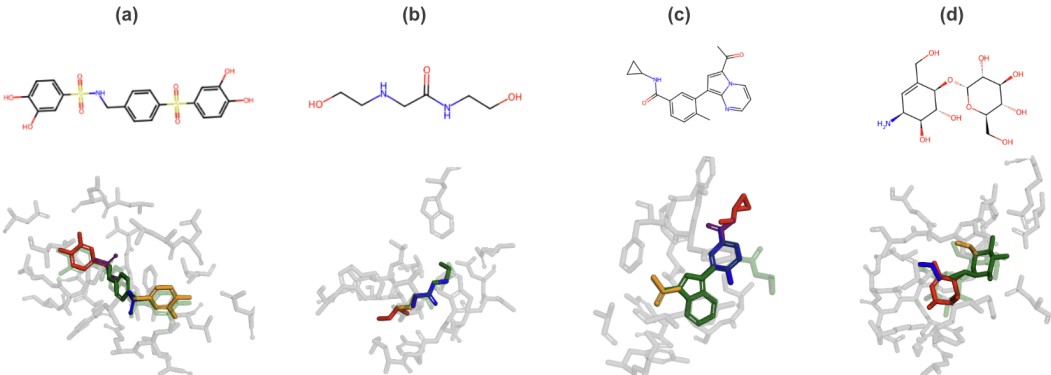

Figure 11: 2D visualizations of ligands (top) and their corresponding generated poses (bottom) across PDB–ligand pairs from the PB set: 7FRX–O88 (**a**, 10.2 Å RMSD), 7KZ9–XN7 (**b**, 4.7 Å RMSD), 6Y7L–QMG (**c**, 4.2 Å RMSD), and 7MGY–ZD1 (**d**, 5.8 Å RMSD). The true bound pose is shown in faint green.

## I.2 Sample Performance & Efficiency

In the main body we use Top-1 (%) as our key performance metric. However, it is also important to assess the *sample efficiency* of our generator. We measure sample efficiency by looking at the rate of change in Top-$k$ as we vary the number of independent samples ($N_{\text{seeds}}$). We report our results in Figure 12.

Notably, our Top-$k$ success rates increase rapidly with only a few seeds, indicating high sample efficiency. This is especially true when considering PB-validity, as a few random seeds rapidly ensures better samples are prioritised according to our heuristic. Conversely, the Top-$k$ accuracies for (RMSD < 2) and (RMSD < 2 & PB-Valid) converge to similar values as $N_{\text{seeds}}$ increases, and are practically indistinguishable when $N_{\text{seeds}} > 20$. Finally, we consider the *Oracle* performance, which represents the empirical maximum success rate achievable by perfectly selecting the best sample from the pool of $N_{\text{seeds}}$. At $N_{\text{seeds}} = 20$, the Oracle reaches over 90% success for the (RMSD < 2 Å) metric and just under 90% for the combined (RMSD < 2 Å & PB-Valid) metric. The relatively large gap ($\sim 10\%$) between this empirical ceiling and our practical Top-1 performance quantifies the potential for improvement in re-ranking the generated candidates, and we leave this for future work.

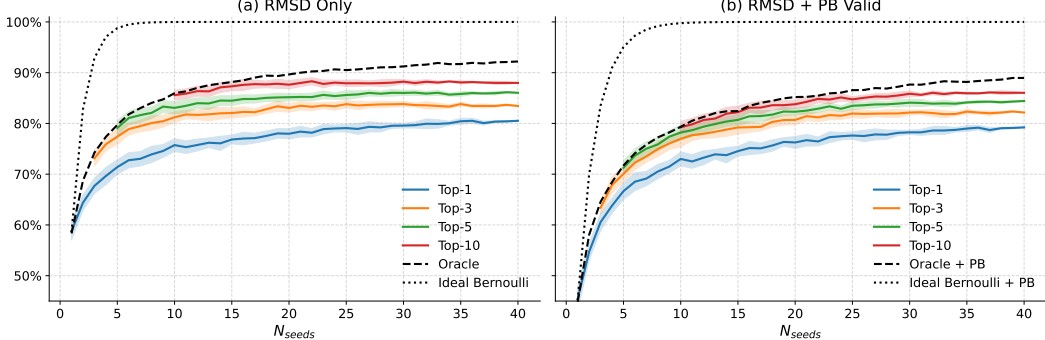

Figure 12: Top-$k$ success rate as a function of the pool sample size $N_{\text{seeds}}$. Solid Top-$k$ lines represent the mean success rates across (40 - $N_{\text{seeds}}$) permutations, with shaded areas representing the standard deviation. The Oracle reflects the empirical *maximum* success rate attainable across the $N_{\text{seeds}}$ samples, equivalent to Top-$N_{\text{seeds}}$. The ideal Bernoulli line represents the empirical optimal, assuming independent sampling probability.

## J LIMITATIONS

### J.1 CURRENT LIMITATIONS AND FUTURE WORK

**Intentionally small training set.** SIGMADOCK was deliberately trained on PDBBind(v2020) to enable fair method-level comparison with the intended PoseBusters train-test split, and to highlight methodological improvements rather than raw performance. This choice clearly reduces the model's out-of-distribution generalisability. In the future, we will look at scaling training with additional curated complexes, applying data augmentation and transfer learning / pre-training strategies, or combining with larger foundation models to improve robustness.

**Dependence on the pocket center at inference.** SIGMADOCK requires a user-specified pocket center to define the search region. Although we debias the choice of the geometric center from the ligand center, and make SIGMADOCK relatively robust to the choice of pocket center during training (see Appendix E.1), when the true center is misspecified, the search space (and therefore predicted samples) can be biased. For fairness, we note that many physics-based and deep-learning-based competing tools introduce a greater bias, such as a strict bounding box around the bound ligand (Trott & Olson, 2010; Zhou et al., 2023b; Jiang et al., 2025). This explicitly restricts the search space over a biased search region $(\Delta x, \Delta y, \Delta z)$, providing information about the structural fit of the bound ligand. Further work could be performed to (i) explore the sensitivity over the choice of pocket centers; (ii) incorporate a different product structure with rigid body translations that separates the requirement of a defined pocket center.

**Chirality challenges from fragmentation.** Although we implement a chirality-preserving scheme by attempting fragmentation merging operations which try to keep chiral centers during sampling, there are cases where no valid reconstruction preserves the original stereochemistry and a chiral center may be altered during fragment linking. Currently, we simply apply post-filtering over the generated $N_{\text{seeds}}$ samples to discard undesired stereoisomers (as performed in Abramson et al. (2024)). Future works could try to explore ways in which SIGMADOCK might better capture stereochemical priors to further ensure chirality is preserved.

**Sensitivity to cofactors and receptor flexibility.** Performance degrades when relevant co-factors are omitted from the input. This indicates the model does not merely memorise unphysical poses but relies on explicit physical context. In future work, we will consider including cofactors and flexible side chains jointly with our $SE(3)^m$ fragments.

**Restricted evaluation protocol (re-docking only).** All reported results are based on re-docking benchmarks. We intentionally restrict the evaluation of SIGMADOCK to re-docking in order to emphasise its key contribution on a task not yet dominated by deep learning methods. These results demonstrate the model's ability to recover bound poses from known pocket conformations, but they do not fully capture its generalisation to more challenging scenarios. While many protein targets have relatively rigid pockets with limited structural change, important future directions include training and assessing performance on cross-docking and apo-structure docking tasks.

### J.2 FURTHER COMMENTS

**AlphaFold3.** It is evident that the task of co-folding is a more complex task than that of protein-ligand docking, wherein there exists a larger set of degrees of freedom (larger dimensionality) to model. This is especially true in our setup, where we assume a rigid protein structure (fair and commonly used in practice (Kamuntavičius et al., 2024), but not a strictly holistic assumption). However, for full visibility, and to support our narrative, we would like to stress that AF3 (Abramson et al., 2024) reports values of up to 84% in their Top-1 metric in the PoseBusters set, without applying rigorous test-train leakage filtering steps. This is evident when comparing the bucketisation of the protein-ligand pairs in the PoseBusters test set, as illustrated in Table 5.

Unexpectedly, AF3's reported accuracy decreases as test–train similarity increases – contrary to our findings and to prior reports (Škrinjar et al., 2025). Despite AF3 evidently training with substantially higher train–test overlap, our Top-1 accuracy is comparable overall and matches or exceeds AF3 across most of the test set (match on $[30, 95)$, surpass on $(95, 100]$). For a fair comparison, we

Table 5: Stratification of the original PDBbind(v2020) vs. AF3 train-test splits on PoseBusters(v2). Sequence similarity split values extracted from AF3 Extended Data Fig. 4c (Abramson et al., 2024). (*) We observe a mismatch between the averaged sequence similarity results (80.2%) vs the reported average performance in AF3's Extended Data Fig. 4e (84.4%).

| Seq. Similarity (%) | Bucket counts | | | Results (Top-1 & PB-Valid (%)) | | |
| | Original | AF3 | $\Delta$ | Ours | AF3 | $\Delta$ |
| --- | --- | --- | --- | --- | --- | --- |
| $[0, 30)$ | 109 | 38 | +71 | 72 | 87 | −16 |
| $[30, 95)$ | 76 | 83 | −7 | 79 | 82 | −3 |
| $[95, 100]$ | 123 | 187 | −64 | 87 | 78 | +9 |
| **Total / Avg** | **308** | **308** | – | **79.9** | **80.2*** | – |

deliberately train SIGMADOCK on a reduced dataset PDBBind(v2020) with ∼19k datapoints, and do not model cofactors. Nevertheless, we still achieve AF3-level performance. Contrary to common concerns about the physicochemical coherence of generative deep-learning models for structure prediction, we observe superior generalisation with a compact model (14.9M vs. ∼500M parameters for AF3) and a set of well defined inductive biases. It is therefore reasonable to expect that SIGMADOCK will further improve with more data, larger models, and added flexibility, and surpass AF3 on more challenging settings such as cross-docking and flexible-receptor protein–ligand docking.

**Difficulties in comparing approaches.** Fair comparison across learning-based docking methods is complicated by heterogeneity in (i) training data and preprocessing, (ii) pocket definitions, and (iii) post-processing pipelines. Differences in (i) affect both the risk of train–test leakage and the apparent ability to generalise (see our comments on AlphaFold3 above). For (ii), pockets are often defined from the bound pose; for example, Zhou et al. (2023a) uses a bounding box around the ground-truth ligand, which may introduce information unavailable in realistic deployments and thus overestimate performance. For (iii), many methods refine generated poses via energy minimisation, sometimes substantially improving reported metrics (Buttenschoen et al., 2024); while sensible in practice, such steps make it harder to attribute gains to the core model and add non-trivial computational overhead. For instance, Alcaide et al. (2025) reports results that rely on the post-processing of Alcaide et al. (2023) as implemented in their code and discussed in their issue tracker [15].

In this work, we mitigate these confounders by training on an intentionally small dataset, to match the setting of Corso et al. (2022) and decouple performance from data scale; by reducing the coupling between pocket definition and the bound pose via the stages in Appendix E; and by reporting metrics directly from SIGMADOCK without structurally altering post-processing. We hope these choices encourage clearer reporting and more comparable benchmarks for deep-learning–based molecular docking.

---

[15]See `https://github.com/deepmodeling/Uni-Mol/tree/main/unimol_docking_v2` with the `--steric-clash-fix` option.

