# OpenReview forum: "SigmaDock: Untwisting Molecular Docking with Fragment-Based SE(3) Diffusion"
_ICLR.cc/2026/Conference — ICLR 2026 Poster_

### Official Review · Reviewer_75zT · 2025-10-31

**Soundness:** 3
**Presentation:** 3
**Contribution:** 3
**Rating:** 6
**Confidence:** 5

**Summary:**

This paper introduces a novel fragmentation scheme, leveraging inductive biases from structural chemistry, to decompose ligands into rigid-body fragments. Building on this decomposition, this paper presents SIGMADOCK, an SE(3) Riemannian diffusion model that generates poses by learning to reassemble these rigid bodies within the binding pocket.

**Strengths:**

1. The presentation of this paper is clear and insightful
2. The experimental results are impressive (could be further enhanced by incorporating more experiments in the main-text)
3. This fragment method which is common in protein structure generation and Structure-based drug design (SBDD) is carefully extended into molecular docking field as an effective approximation for co-folding methods to strike a balance in effectiveness and effiency.

**Weaknesses:**

1. The proposed methods is limited in pocket-aware docking setting, which requires pocket identification in advance.
2. The method doesn’t model the pocket flexibility which is crucial for pocket-aware docking task. Since apo→holo pocket structure transformation is very common.
3. As this method is more closed to the protein-ligand complex structure prediction task (or even SBDD, as fragments-based models are very common in SBDD) , more comparison against complex structure prediction is appreciated.

**Questions:**

1. How to perform score estimation for Uniform distribution on SO(3)? This distribution is convenient for flow matching method. Why not use the Isotropic Gaussian Distribution for diffusion model?
2. Torsional diffusion could handle tasks like side-chain packing and how could the framework of SigmaDock handle this?

---

> ### Author Response · Authors · 2025-11-23
> **Clarifying main methodological contribution, practical HTVS focus and path to flexible side-chain docking.**
>
> Thank you for the thoughtful and technical feedback. We address each point below and have incorporated the corresponding changes in the revision.
>
> ## Weakness 1: “The proposed methods is limited in pocket-aware docking…”
> This is a deliberate design choice to focus on one of the most common and critical tasks in drug discovery. While blind docking is useful for binding site identification, the pocket-aware setting is the primary task for high-throughput virtual screening (HTVS) and lead optimisation. In practice, a binding site is often known from experimental evidence or computational prediction: practitioners typically employ tools such as FPocket to detect candidate pockets, or rely on structural evidence via crystallography of a few active compounds for that protein / homologues. *We now detail this in the introduction (page 2, last 2 paragraphs) to distinguish our setup and differences (practical) between identification from pose generation, and have further clarified the intended task of re-docking.*
>
> ## Weakness 2: “The method doesn’t model the pocket flexibility…”
> We thank the reviewer for highlighting this key distinction and agree that side-chain flexibility matters. While incorporating flexible side chains is straightforward in our framework, we want to highlight that our main contribution is methodological. Using an SE(3) Riemannian diffusion model over rigid fragments, we are the first generative model to match/surpass classical dockers on the rigid-receptor PoseBusters benchmark with high physical plausibility, and we substantially outperform other DL methods which assume the same construct (rigid receptor, training dataset), without requiring minimisation. We view this as a strong stepping stone toward flexible docking, which our framework naturally supports, and leave this exciting research direction as future work.
>
> ## Weakness 3: ”As this method is more closed to the protein-ligand complex structure…”
> We added a focussed comparison to AlphaFold3 in the main text (p 10, Table 4) and an extended analysis in Appendix J.2. Here, we demonstrate how we reach AF3-level performance with significantly lower test-train leakage, using a fraction of their training data, and orders of magnitude faster inference. We have also added an additional ablation (p 10, Table 3) to better characterise SigmaDock's robustness to larger pocket definitions, a common Achilles heel for search-based docking methods. For completeness, we also added a discussion on section 3.2 including the more recent DiffDock-L, where they report a Top-1 of 50% (RMSD only, no PB checks) whilst training on (BindingMOAD + PDBBind); in contrast, our method reaches a Top-1 (RMSD & PB-Valid) of ~80% whilst training on PDBBind only (lower test-train leakage & half the amount of data for training).
>
> ## Question 1
> Thank you for pointing out this source of confusion. To be precise, we do not estimate the score of the uniform distribution. The uniform distribution $U_{SO(3)^m}$ is the stationary distribution (prior) at $t=T$. The forward process (SDE in Sec 2.3) uses the standard Brownian motion on $SO(3)$, for which the tractable forward kernel is the Isotropic Gaussian on SO(3), $p_{t|0} = IG_{SO(3)}$, as you pointed out. We then train our model $s_\theta$  by regressing on the score of this conditional $IG_{SO(3)}$ density, $\nabla_R \log p_{t|0}$. The exact equations are in Appendix C (p18-19). Hence, as suggested, our process is indeed based on the IGSO(3) density. We have left these details in the appendix due to space constraints.
>
> ## Question 2
> Our current framework does not currently handle side-chain packing, as we operate under the rigid-receptor assumption (noted in our limitations on Appendix J.1). Torsional models are typically adopted for this, but as we argue in Theorem 1 / Appendix C.2, they suffer from 'entangled' dynamics. Indeed, a very promising future direction for our method, which our framework is uniquely suited for, is to extend the $SE(3)^m$ model by treating flexible protein side-chains as additional rigid-body fragments to be diffused. We have clarified this in different sections of our manuscript to clarify this distinction (introduction, conclusion, limitations).
>
> ## Closing Remarks
> We thank you again for your constructive feedback. We hope these clarifications and inclusions, particularly regarding the justification for our diffusion process and the practical reasons for our problem scope, have fully addressed your concerns and demonstrated the significance of our methodological contribution (and future developments).
>
> With these considerations, we kindly ask the reviewer to reconsider raising our rating towards acceptance.

---

### Official Review · Reviewer_v3fT · 2025-11-01

**Soundness:** 3
**Presentation:** 3
**Contribution:** 4
**Rating:** 8
**Confidence:** 3

**Summary:**

This paper proposes SigmaDock, a novel SE(3) Riemannian diffusion model for molecular docking that operates on rigid-body transformations of ligand fragments. Compared to prior torsional models, this approach has the advantage of simplifying the design and training of the diffusion model. The key contributions are a novel ligand fragmentation scheme and the design of the diffusion process and model architecture. The model shows impressive results on the reported docking benchmark.

**Strengths:**

* The paper addresses the relevant and challenging task of molecular docking, which has many applications in drug discovery. Also, the paper is very well-written.
* The key idea of decomposing the ligand into fragments and defining the diffusion process over rigid-body transformations over these fragments is sound and interesting.
* The experimental results are impressive.

**Weaknesses:**

* This paper is inspired by and similar in spirit to DiffDock [1]. The authors should compare the two methods in more detail and highlight the key differences between the two diffusion processes.
* All the reported experiments use a model trained on a single dataset. It is unclear how the model would perform on other datasets/benchmarks.

**References**

[1] Corso, Gabriele, et al. "Diffdock: Diffusion steps, twists, and turns for molecular docking." arXiv preprint arXiv:2210.01776 (2022).

**Questions:**

1. Regarding the conformational manifold, where are these holonomic constraints used in the literature? Can you provide references?
2. The ablation study in Table 1 is not clear, e.g., the base method, which should be the main point of comparison, is missing from the table.
3. How is the knowledge about the docking pocket leveraged by the model during training and inference?
4. This work assumes the protein conformation does not change during docking (rigid docking). Can the model be extended to model protein conformation changes as well?
5. In line 24 (Abstract), the baseline metric should be 32.8% instead of 30.8% (according to Figure 4).
6. In line 485 (Footnote 10), I believe RMSD should be "above", not "below", 2A.

---

> ### Author Response · Authors · 2025-11-23
> **Addressing questions & clarifying differences between torsional & SE(3) diffusion framework & discussion on fair test & future work.**
>
> Thank you for the thoughtful and supportive review. We address each point below and clarify the requested comparisons.
>
> ## Weakness 1: “This paper is inspired by and similar in spirit to DiffDock…”
> The key methodological difference is the diffusion space. DiffDock operates in torsional product space $\mathbb{T}^k \times SE(3)$. As we summarise in Sec. 2.2.2 (Theorem 1, extension in Appendix C.2), this leads to a highly entangled, non-product measure: a simple noise in torsion space becomes a complex, correlated noise in the 3D Cartesian space where interactions are observed. Our key novelty is diffusing disjoint rigid fragments in $SE(3)^m$, restoring a product structure, which yields a simpler and better-conditioned learning problem. This superior formulation is the primary reason for our 6.3x higher Top-1 PB-validity over the original DiffDock (79.9% vs 12.7%). *We have clarified this crucial distinction in Section 2.2.2. of the revised manuscript and add an explicit reference to Appendix C.2 which details the differences between the two diffusion processes further.*
>
> ## Weakness 2: “All the reported experiments use a model trained on a single dataset…”
>
> This is an important point. Our aim for this work is a methodological one: to introduce a novel framework and validate it fairly against other methods on the standard PoseBusters benchmark, which uses PDBBind(v2020) for training. We aimed to show that our proposed method is the key differentiator, not the training data.  We agree that extending this to larger datasets to study data scaling is an excellent direction for future work, but we decided this was beyond the scope of this initial methodological paper. *We have added further discussion on the comparison of our method to AF3 in the main text (p 10, Table 4) and further analysis in Appendix J.2.*
>
> We have also added a discussion *(on section 3.2, top of p. 9)* where we elaborate that the more recent DiffDock-L, which trains on a much larger dataset (BindingMOAD + PDBBind v2020), reports a Top-1 (RMSD-only) success rate of 50%. Yet, whilst training on a significantly reduced portion of the data (just PDBBind v2020), we still greatly outperform this, achieving ~80% Top-1 & PB-Valid. We hope these additional results further strengthen our paper.
>
> ## Questions
> - (1) **Holonomic Constraints**: Bond lengths / angles are treated as holonomic constraints extensively in computational chemistry / physics and MD (see SHAKE (Ryckaert et al., 1977) or ETKDG (Riniker-Landrum, 2015). We added references to the main text (footnote 4) for completeness. We empirically validate this assumption in Appendix D.3 , where we show that we can align RDKit-generated conformers (which respect these constraints) to the ground-truth bound pose with a negligible average RMSD of 0.21Å. This proves that our rigid-fragment assumption is a valid and powerful simplification, and is a necessary condition to motivate our approach.
> - **(2) Ablation study Table 1**: The base method is the SigmaDock ($N_{seeds}$ = 40) row. We have clarified what the default method is in the caption.
> - **(3) Knowledge of Pocket**: We detail our pocket definition in Appendix E.1 (p28). Briefly: during training we construct a stochastic pocket by selecting residues with any heavy atom within $d_{r} = d_0 + \mathcal N(0, \sigma_r)$ of any ligand atom; the pocket center is the CoM of the selected residues plus Gaussian perturbation $\mathcal N (0, \sigma_{CoM} I_3)$. We realise this is an important distinction, and have added more details to the main text in 2.5, with an additional reference to the appendix. We added a new pocket-sensitivity sweep ($d_{pocket} \in [4-7]$A) and show stable performance with larger pocket sizes, which is critical for practical scenarios where the exact binding environment is more uncertain. This is now present in page 10, Table 3.
> - **(4) Protein Flexibility**: This is a key limitation of our current work (noted in Appendix J.1), and is a clear direction for future research. Our fragment-based $SE(3)^m$ framework provides a strong foundation for this extension, as flexible side-chains can be naturally treated as additional rigid-body fragments to be diffused.
> - **(5) & (6)**: Thank you for catching these. We have corrected both typos in the revised manuscript.
>
> We appreciate the positive assessment and the helpful feedback which have helped strengthen our paper further. In light of these revisions, we hope that, given your excellent contribution score, you might reconsider your rating towards strong acceptance.

---

### Official Review · Reviewer_EyZ8 · 2025-11-01

**Soundness:** 4
**Presentation:** 3
**Contribution:** 3
**Rating:** 6
**Confidence:** 3

**Summary:**

This paper introduces SIGMADOCK, a novel deep-learning approach for molecular docking. The method moves away from traditional torsional diffusion models by first decomposing a ligand into a set of rigid-body fragments. It then employs a Riemannian diffusion model on the SE(3) group to learn how to reassemble these fragments within the protein's binding pocket. Key contributions include a novel fragmentation scheme (FR3D) to reduce the system's degrees of freedom, the use of soft geometric triangulation constraints as an inductive bias, and a tailored SO(3)-equivariant architecture. The authors report that SIGMADOCK achieves state-of-the-art performance on the PoseBusters and Astex benchmarks, claiming it is the first deep learning method to surpass classical physics-based docking tools under the specified train-test split.

**Strengths:**

1.   The paper is well-written, clearly motivated, and theoretically sound. It provides a strong argument for the limitations of torsional models and convincingly presents the fragment-based approach as a superior alternative.
2.   The proposed method is novel and principled. The combination of the FR3D fragmentation scheme, soft geometric constraints, and an SE(3) diffusion process introduces strong and chemically-aware inductive biases into the model.
3.   The empirical results are highly impressive. SIGMADOCK demonstrates a significant performance leap over previous deep learning methods and, notably, classical docking software on challenging, temporally-split benchmarks. The high rate of chemically plausible outputs (PB-validity) without post-hoc minimization is a major strength.
4.   The paper includes a thorough ablation study that validates the contribution of each key component of the model, alongside an insightful analysis of failure cases related to co-factors.

**Weaknesses:**

1.   The paper argues that its fragmentation scheme (FR3D) helps manage the degrees of freedom, but the analysis could be more thorough.

2.  The definition of the protein binding pocket is a critical input for any docking model, yet it is not discussed in the main paper. Typically, this involves selecting atoms within a certain radius of the ground truth ligand, making this radius a key hyperparameter. The sensitivity of SIGMADOCK's performance to this pocket definition is not analyzed. An ablation study (such as anlyais in PoseBusters paper for UniMol) showing how performance changes with different pocket size thresholds would provide valuable insight into the model's robustness and its potential applicability to scenarios where the binding site is less precisely defined.

**Questions:**

See above

---

> ### Author Response · Authors · 2025-11-23
> **Response to Reviewer: Clarification of FR3D & New Ablation on Pocket Size.**
>
> We are grateful for your positive and encouraging feedback, including your assessment that the paper is theoretically sound, novel, and highly impressive. We address your suggestions below and have incorporated the corresponding changes in the revision.
>
> ## Weakness 1: “The paper argues that its fragmentation scheme…”
> Empirically, FR3D reduces the number of fragments from the naïve (k+1) to approximately $\tfrac{2}{3}(k+1)$ ($\approx$ 34% fewer fragments), simplifying optimization. Trivially we reduce the DoFs by reducing the number of fragments. Extensions (pseudo-reductions) follow from the triangulation conditioning. Table 1 (Config C) shows that using naïve (k+1) fragmentation instead of FR3D causes a 6.2% drop in Top-1 (RMSD<2A & PB-Valid), confirming FR3D’s contribution. Similarly, removing triangulation conditioning yields a similar result. *A more detailed analysis supporting the empirical evidence and clarifying the DoFs is now presented in Appendix D.4 and referred to in the main text (second paragraph section 2.2.3).*
>
> ## Weakness 2: “The definition of the protein binding pocket is a critical input…”
> We agree that the pocket definition is critical. Our model was already designed with this exact concern in mind, and we have now performed the suggested ablation to confirm our model's robustness.
>
> **Clarification**: As detailed in Appendix E.1 (p28), we do not use a simple fixed-radius pocket. Our training procedure is explicitly designed for robustness:
> - We define the pocket using residues within a stochastic cutoff: $d_{r} = d_{0} + \mathcal{N}(0,1)$ from any ligand atom. The pocket center is set to the CoM of the selected residues and is further perturbed: $\bar x_{pocket} = \text{CoM}(\text{residues})$ + $\mathcal{N}(0,\sigma_{CoM}I_3)$. Importantly, we do not center the pocket using the bound ligand’s CoM to avoid leakage / bias. We point out this choice often used in the literature, which we also further discuss in Appendix J.1.
> - Our training-time stochasticity debiases the definition and encourages invariance to the precise choice of center and radius, improving robustness at inference. *We have now added a brief summary of the pocket definition in the main text (Section 2.5) and added an explicit reference to Appendix E1, and thank you for pointing out this potential point of ambiguity*.
>
> **New Ablation (reviewer suggestion)**: We sweep the inference-time pocket cutoff $d_{r}$ from 4 to 7 Å, and report Top-1 (RMSD<2Å & PB-Valid) across this range, offering a large margin for the definition of the pocket. Performance remains stable over the range covered by the training distribution, with a moderate degradation only when moving substantially beyond it, which is also expected as the search space expands due to the growing train-test mismatch (7A is 2$\sigma$ away from training mean). We also provide pocket volume statistics to quantify the change induced by the sweep. Results and discussion are included in p.9, Table 3.
>
> We hope these additions have addressed your initial concerns and further strengthen our work; thank you for your constructive insights. Given your positive evaluation of the method’s novelty and rigour, we kindly hope you consider the revisions & clarifications provided merit a stronger recommendation towards acceptance.

---

### Official Review · Reviewer_n4Mh · 2025-11-02

**Soundness:** 3
**Presentation:** 4
**Contribution:** 1
**Rating:** 4
**Confidence:** 4

**Summary:**

The paper develops a diffusion model for rigid-protein, known-pocket molecular docking. The ligand is fragmented into rigid bodies by breaking torsional bonds and the pose is generated by denoised over the SE(3) manifold for each fragment. The model is trained and evaluated on PDBBind, where the model surpasses the success rates of Gold, Vina, and DiffDock.

**Strengths:**

* The paper does a good job of motivating the problem formulation. The analysis of the right space to diffuse over for molecular docking is thoughtful, with theoretical results supporting the conceptual arguments.

* The technical exposition and proofs in the appendix are clear and nicely done.

* The preservation of performance at low sequence similarity to the training set is very nice to see.

**Weaknesses:**

**Significance**
* The topic is somewhat stale, with wide consensus on Euclidean diffusion and co-folding for molecular docking. While contributions that challenge the consensus are welcome, they should provide a compelling value proposition rather than retread problem formulations that are no longer of primary interest.

* The historical interest in docking to rigid receptors largely stems from computational considerations and works focusing only on this task should not be encouraged, as holo structures are not available in practice and sidechain interactions to accomodate ligands are always of interest in scoring poses.

* In particular, the additional complexity of manifold diffusion needs to be justified by marked improvements in performance relative to simpler alternatives, which the paper does not convincingly show.

**Experiments**
* Both the dataset (PDBBind) and the task (pocket-guided docking) make comparisons to state of the art co-folding models difficult. In the absence of such evaluations, it is impossible to judge the significance of the contribution.

* If authors are comparing against DiffDock, they should at least include DiffDock-L. Many of the other baselines are also quite outdated (TankBind, UniMol).

**Questions:**

No specific questions.

---

> ### Author Response · Authors · 2025-11-23
> **Response to Main Weaknesses (1 & 2): Re-docking scope, practical relevance, fair comparison & method development.**
>
> We appreciate the thoughtful review and the positive notes on soundness, presentation, and theory. Our scope is *not to replicate co-folding* (beyond our academic resources), but to tackle fast, physically-plausible, pocket-conditioned docking at screening scale, a setting where deep learning methods have historically underperformed classical tools, via our methodological innovations. We have clarified this scope and strengthened comparisons as detailed below.
>
> ## Weakness 1: “The topic is somewhat stale…”
> - **Relevance.** We respectfully but strongly disagree with the assessment that this topic is “stale”, and we hope we can politely raise a case against this. While co-folding is a popular and powerful new paradigm, the problem of fast, physically-plausible generative docking for tasks like High-Throughput Virtual Screening (HTVS) and lead optimisation remain largely unsolved due to the computational infeasibility of applying co-folding models to the scale required in these common use cases. Our method directly targets this practical regime.
>
> - **Failure of Generative Docking Methods**; Prior DL methods failed on this task, achieving Top-1 performances up to 12.7% (PB-Val) and 38.0% (RMSD only) for DiffDock, 50% on DiffDock-L (RMSD only) and up to 50.7% (RMSD only) and 32.8%(PB-Val) on more recent generative methods (ReDock). Conversely, physics-based methods (Vina, Gold) are far superior (~55-60% PB-Val). The PoseBusters benchmark was designed specifically to highlight this field-wide failure. *We have now included a discussion on DiffDock-L’s performance in the main text (p. 9, beginning of section 3.2)*. We appreciate you raising this point as we believe these results further evidence our proposed method (we generalise significantly better (Top-1 & PB-Val of ~80%) with 2x less data and lower test-train leakage). We also added a detailed comparison to AF3 in the main text (Table 4) and Appendix J.2.
>
> - **Our Contribution.** To our knowledge, SigmaDock is the first generative method to surpass these classical tools on the fair PoseBusters benchmark, without requiring energy minimisation or post-processing on generated poses. We believe our SE(3) Riemannian diffusion model does not represent an incremental step; it is a novel solution that successfully tackles the precise problem the PoseBusters benchmark identified. We do not claim to solve flexible-receptor docking; rather, we show a principled, efficient framework that finally surpasses classical methods for re-docking and forms a strong methodological base for future and more general/practical extensions.
>
> - **Methodological Premise.** We don’t believe the claimed consensus on treating point clouds without equivariance to be a gnerally accepted consensus [1], but a simplified (yet computationally expensive) design choice on recent co-folding methods (such as AF3) that we believe is not well posed (AF3 do no ablations on this). These methods leverage massive models (AF3 has ~500M parameters) and expensive training  (AF3 augments a state via 48 rototranslations to “learn” equivariance). Our work, in contrast, presents a principled, built-in SE(3)-equivariant framework that is theoretically sound and exploits geometric symmetries in chemistry. We are concerned with this statement as we believe our value proposition is compelling, well motivated, and evidenced based on our theoretical analysis and our results. We also ablate some of the novel contributions in Table 1.
>
> ## Weakness 2: “The historical interest in docking to rigid receptors…”
> We agree the task of docking toward the holo-receptor (re-docking) is simpler, and side-chain flexibility is oftentimes important for faithful modeling protein-ligand binding. However, our focus on the re-docking task is a deliberate methodological choice:
>
> - Re-docking is the long-established, standard benchmark for validating the algorithmic correctness of a docking method. We aimed to prove, on a fair apples-to-apples basis, that our method could finally solve this problem better than classical tools.
> - You highlighted "holo structures are not available in practice." While true for de novo target discovery, our task is aimed at HTVS and lead optimisation, where rigid-receptor docking is standard for the entire pharmaceutical industry, precisely because it is computationally tractable. Furthermore, docking to apo structures is a different and often inconclusive task; a failure to dock to a rigid apo conformation does not mean the docking tool fail since the conformation need not be binding-competent. We leave this for future work as we believe a faithful assessment requires extensive scientific consideration.
>
> We have clarified the scope of our paper in the revised text (Introduction, Results) addressing W1&W2 and appreciate your insights, which have helped us narrow down our main message & contributions further.
>
> [1]: Pearl: A Foundation Model for Placing Every Atom in the Right Location, Dobles et al. (2025).

---

> ### Author Response · Authors · 2025-11-23
> **Response to Weakness 3 (Motivation behind SE(3) Riemannian Diffusion) & Clarifying Experiments & Comparison to CoFolding/DiffDock-L**
>
> ## Weakness 3: “In particular, the additional complexity of manifold diffusion…”
> We believe the added complexity in our framework is the key novelty which reduces the effective complexity, yielding a simpler learning problem.
>
> - Other methods we benchmark against use atomic (point-cloud) representations or the torsional manifold diffusion parametrisation (DiffDock/ReDock), hence we believe we directly demonstrate our contribution empirically. Furthermore, we argue that the additional complexity of manifold diffusion is precisely what justifies our performance. The “simpler” torsional alternative forces the model to learn a non-local, highly nonlinear inverse problem. In contrast point-cloud models don't exploit geometric symmetries (redundancies) in structural chemistry. As we show in Theorem 1 (Appendix C.2), our SE(3) diffusion framework is in fact provably simpler to the torsional diffusion alternative, which is the fundamental point which drives the stark increase in performance with respect to DiffDock, ReDock, DiffDock-L, and so on. Additional geometric inductive biases also improve our performance, as we detail in our Ablations (Table 1.).
> - A direct comparison to co-folding models is difficult as it is beyond the scope of our current work. However, *we have added a more detailed comparison to AF3, from our original inclusion in section 3.2. This is now in Appendix J.2, and we have added further discussion in the main text (Table 4).* Importantly, we show that *we achieve AF3-level performance without test-train leakage*, which we believe to be an important result backing up our proposed manifold-diffusion method: we generalise better with less data and with a fraction of the compute and sampling time.
>
> With this perspective, we hope you may kindly reconsider your contribution score (1: poor), and most importantly, consider increasing your current rating (4: marginally below acceptance threshold) towards an accept. We hope you can appreciate our detailed methodological work and find the proposed framework relevant, rigorous, and interesting for future research.
>
> ## Experiments
> **“Both the dataset (PDBBind) and the task (pocket-guided docking) make comparisons to state of the art co-folding models difficult. In the absence of such evaluations, it is impossible to judge the significance of the contribution.”**
> - We restate that our motivation for our experimental setup of re-docking to rigid structures from the Astex and Posebusters test set and restricting our training dataset to PDBBind(v2020) is that this allows for a fair scientific evaluation of our proposed methodology, since this setup has seen scientific consensus (i.e. Posebusters) as a suitable way to benchmark new methods.
> - This standardised setup allows us to understand whether our improved results are essentially due to our new methodology instead of other confounding variables such as using a larger dataset. This is exactly why we do not directly compare against co-folding models as such models have been trained using a vastly different setup using much larger datasets, compute and assumptions. Moreover, as we do observe such a large gain in performance for SigmaDock under this standardised setup, we are confident that this represents a scientifically significant result (which would not necessarily be the case if we were not so careful in designing our experiments).
> - Additionally, even when we do compare against co-folding models such as AlphaFold 3 on the Posebusters test set (see the newly added Appendix J.2 and Table 4), we observe that our results (Top-1 79.9%) are comparable with AlphaFold 3 (Top-1 84%). This is despite SigmaDock using vastly less data and compute to train compared to AlphaFold 3, and with significantly lower test-train leakage. Therefore, we would like to kindly push back against the claim that the significance of our paper is impossible to judge when we observe such clear results.
>
> **"If authors are comparing against DiffDock, they should at least include DiffDock-L."** We have added a discussion (on section 3.2, top of p. 9) including results from the more recent DiffDock-L, which trains on a much larger dataset (BindingMOAD + PDBBind v2020), reporting a Top-1 (RMSD-only) success rate of 50%. Yet, whilst training on a significantly reduced portion of the data (just PDBBind v2020), we still greatly outperform this, achieving ~80% Top-1 & PB-Valid. *We hope these additional results further strengthen our paper.*

---

### Official Review · Reviewer_xTvN · 2025-11-04

**Soundness:** 3
**Presentation:** 3
**Contribution:** 3
**Rating:** 6
**Confidence:** 4

**Summary:**

The authors propose a fragment based docking method (sigmadock). The ligand is first broken into fragments that are then coordinately placed in a bound configuration against a fixed protein target. The main motivation for the approach follows prior work in terms of using rotatable bonds as the main degrees of freedom for the ligand beyond its global rotation and translation relative to a fixed protein target (presumably holo structure). Instead of operating on the angles directly as in prior work, the authors fragment the ligand across rotatable bonds (also heuristically reducing the number of fragments), and define a forward diffusion as a product measure across the fragments. A SE(3) equivariant de-noiser is then learned to iteratively refine noisy fragment rototranslations back into their bound configuration. The method is tested against (older) alternatives on PoseBusters set.

**Strengths:**

The motivation for fragment based docking is clear and compelling. The mathematical derivations are clear and straightforward albeit appear mostly in the appendices, not as part of the main paper.

**Weaknesses:**

There are a number of unsubstantiated and inconsistent claims, starting with the abstract. The presentation needs to be revised; any claims must accompany clear empirical or theoretical support. E.g., in section 2.2.2, the authors say that "... breaking the product structure [leads] to ill-conditioned and often degenerate dynamics during training and sampling". What's the evidence for ill-conditioned and degenerate dynamics? All claims of this kind require either supporting evidence or be removed. My rating assumes that the authors will make such changes.

The authors claim that co-folding methods are slow but such methods account for protein structural changes following binding. Presumably the authors start with a holo structure that is already altered towards accepting the ligand in question (?), thus making the task easier and unsuitable for fair screening across apo structures (or holo structures corresponding to different ligands). Reaching AF3 performance now does not make the method "the first" deep learning method to surpass physics based docking.

Comparisons are not to the state of the art methods. E.g., DiffDock (2023) has already been surpassed by its own refinements and by other methods. AF3 or Boltz-1x/2 (with pseudo-physical steering) would constitute more appropriate comparisons among many others. If the main argument is speed vs accuracy, then the authors should compare across the two axes, not exclude state of the art methods from comparison.

**Questions:**

Is the fixed protein structure a holo structure? How would the method perform relative to a slightly different apo structure?

Soft triangulation constraints are used across fragments to preserve bond lengths and angles. However, since fragments are refined coordinately as a set, presumably there will be some discrepancy between them about the corresponding rotatable bonds and angles. How are these resolved in the final answer?

How is ligand chirality handled? Are cyclic structures explicitly excluded?

theorem 1 seems vague and inflated. Naturally the mapping from torsional angles to cartesian coordinates is non-linear and thus does not lead to a product measure over fragments. What does "highly entangled" refer to?

does the forward diffusion begin with an aligned structure to the ground truth pose, i.e., selecting approximate torsion angles that best align with the ground truth bound pose? Otherwise internal fragment bond lengths etc would not necessarily be consistent with those from RdKit.

the reference to Jin et al. 2023 for their SE(3) equivariant prediction head is a little vague. What exactly is used from that reference?

---

> ### Author Response · Authors · 2025-11-23
> **Addressing Reviewer Weaknesses (clarifying scope, theory, and fair comparisons in re-docking)**
>
> Thank you for your constructive feedback. We have revised the manuscript to address the main weaknesses you outlined, which we believe significantly strengthens the paper.
>
> ## Weakness 1: “There are a number of unsubstantiated and inconsistent claims…”
>
> This is one of the core theoretical motivations for our work, which we expanded in Sec. 2.2.2 and Appendix C.2. Here, we argue that our method defines a provably simpler and better-conditioned learning problem. We have clarified this theoretical motivation in Section 2.2.2 of the revised manuscript.
>
> Torsional Models (Appendix C.2.1): In torsional models, a change in one angle causes non-local Cartesian displacements of atoms far down the chain. A simple, independent noise model in torsion space becomes a complex, correlated, and anisotropic noise model in 3D Cartesian space (where interactions are observed). This is the ill-conditioned dynamic we refer to in the main text. In contrast, our method is factorisable by construction (disjoint $SE(3)^m$ fragments): moving one fragment does not affect any other, leading to simpler dynamics.
>
> Moreover, our approach greatly outperforms models adopting the torsional diffusion framework, providing strong empirical evidence for on the training & sampling limitations behind torsional diffusion.
>
> ## Weakness 2: “The authors claim that co-folding methods are slow but…”
> We agree this distinction should be made explicit. In the introduction (penultimate paragraph, p.2) we now state that our rigid-receptor, holo-structure setup is a deliberate methodological choice to address the re-docking problem. This protocol is the long-standing benchmark for validating docking methods, and the PoseBusters temporal split is designed for this setting. For an apples-to-apples comparison, we therefore evaluate against methods trained and tested under the same protocol (e.g., UniMol, TankBind, ReDock, DiffDock/DiffDock-L when used in holo re-docking). Within this scope, our main contribution is to be, to our knowledge, the first DL method on the intended PoseBusters split to surpass classical physics-based docking on the stricter Top-1 metric (RMSD<2 Å and PB-Valid).
>
> Apo evaluation is a valuable but different benchmark. Strictly, apo conformations need not be binding-competent. Hence, docking to a single rigid apo snapshot can therefore confound algorithmic performance with receptor state choice and inflate false negatives. Rigorous apo studies typically require ensembles of apo states and/or flexible-receptor protocols (e.g., side-chain sampling or induced-fit), which are outside our current scope. We now state more explicitly (intro, limitations, discussion) that extending SigmaDock to flexible side-chains (naturally handled as additional fragments in our $SE(3)^m$ framework) is a key direction for future work.
>
> An assessment of apo-docking via ESM is not necessarily rigorous; there is extensive literature showing that, due to the training data (highly flexible structures often require artificial binders to stabilise loops and/or other regions to resolve the structure via crystallography), synthetically generated “apo” structures can be similar to their holo counterparts and/or miss the multimodal nature therein [1].
>
> [1] Assessing Structures and Conformational Ensembles of Apo and Holo Protein States… (Raisinghan, et al., 2024).
>
> ## Weakness 3: “Comparisons are not to the state of the art methods…”
> **Other Baselines**. The 12.7-32.8% baseline refers to the original DiffDock (12.7%) and the more recent ReDock (32.8%), which extend DiffDock’s torsional framework to a pocket-conditioned setup. While recent reports for DiffDock-L claim a Top-1 performance of 50% (RMSD only), these results are not directly comparable as they train on a larger dataset (BindingMOAD + PDBBind). In contrast, our 79.9% (RMSD < 2Å & PB-Valid) score remains significantly above concurrent methods, crucially without any post-hoc minimisation, and holding the original split for the PoseBusters set (train on PDBBind only). For completeness, we have added a discussion on DiffDock-L in the main text (p 9, section 3.2), as further strengthens our contributions. We thank you for pointing this out!
>
> **Co-folding methods**. As co-folding models are trained with substantially larger and more diverse datasets, a direct comparison is complex. However, for completeness, we have added a summarised comparison in the main text (p.10, Table 4) with a further extension in Appendix J.2. Although we recognise there is future work to make SigmaDock a general tool (such as flexible side-chains), we make the following observations:
>
> - AF3's test set has significant train-test leakage (>60% of test data having >95% sequence similarity). Hence, whilst matching their performance, we show better generalisation
> - We achieve this with a fraction of the data and compute (our inference is up to 200x faster), making it suitable for HTVS where co-folding models are computationally infeasible

---

> ### Author Response · Authors · 2025-11-23
> **Addressing Reviewer Questions**
>
> **Holo Structure**: Please see our response to Weakness 2 above.
>
> **Soft Triangulation Constraints**: These are not resolved post-hoc. They are a conditioning signal fed directly into the score model as a dynamic edge feature (Sec 2.4). The mismatch $\Delta d_{A,C}(x_t, t) = ||A(t)-C(t)|| - d^\text{ref}_{A,C}$ is an input, and the model learns to minimise this mismatch during denoising. As shown in Table 1 (Config A), removing this signal causes a 12.8% drop in PB-validity, proving it is a critical inductive bias. Reference distances are computed from the initial conformer generated by RDKit and there is no discrepancy in the final result since we use dummy atoms across torsional bonds, which are ignored in the molecular construction, where torsional bonds are uniquely defined from the "anchors." *We have clarified this potential point of confusion in the main text (2.4, conditioning paragraph).*
>
> **Chirality & Cyclic structures**: This is a key challenge (noted in Appendix J.1). Our FR3D scheme (Appendix D.4)  stochastically attempts to preserve chiral centres. However, when this is not possible, we use post-filtering to remove invalid (according to the input smiles) stereoisomers, which is standard practice (also used by AF3). We observe only ~3-4% of samples failing this check. Cyclic structures are not excluded and are part of training (albeit large cycles are uncommon (1-2%)). However, we agree that the rigidity assumption for large, flexible cycles is less robust, and a simple solution which we will explore in future work, is to split such cycles into smaller fragments. We appreciate the observation!
>
> **Theorem 1**: Please see our response to Weakness 1 above.
>
> **Forward Diffusion start**: Yes, the process begins with a conformer aligned to the ground truth. This is key for the model to be robust to the initial choice of conformer, which is fundamental for generalisation. *We clarify this in Appendix D.3 & E.1*. To be precise: during training we observe the ground-truth bound ligand pose $x_b$, generate a conformer $x_c$ (via RDKit), and find the optimal alignment $x_b'$ (via rigid roto-translation + torsional updates). $x_b'$ has a negligible avg. RMSD of 0.21A to $x_b$, as shown in D.3. $x_b'$ is then fragmented to get the $t=0$ pose $z^{(0)}$. At test time we show how, when using $x_b$ instead of $x_b’$ (Table 1G), we do get better performance (86.4% top-1), simply because of small local perturbations. Nevertheless, we acknowledge our main result with $x_b’$ as it is unbiased. *We have now clarified this further in the main text (Last sentence of 2.2.1), and thank you for raising this potential source confusion.*
>
> **Prediction Head**. This is detailed in Appendix G.4 (p35). We adapt the SO(3)-equivariant prediction head from Jin et al. for our denoising diffusion process, which is based on Newton-Euler equations. This head predicts equivariant forces ($F_F$) and torques ($\tau_F$) on each fragment, which is essential to guarantee Theorem 2: our training objective is invariant to the arbitrary choice of local coordinate axes for each fragment. *We have clarified this in section 2.4 (before Theorem 2) as we agree it is an important distinction.*

---

### Meta-Review · Area_Chair_S5pG · 2026-01-05

**Summary:**

In this submission, the authors developed a fragment-based SE(3) diffusion model for the molecular re-docking problem. The proposed method decomposes ligands into rigid-body fragments and performs re-docking by generating poses for the fragments using an SE(3) Riemannian diffusion model. Experiments verify the usefulness of the proposed method to some extent. Four of five reviewers appreciate the contributions of this work, and the main concerns are about the task setting and the implementation details. Only one reviewer expresses concerns about the method's novelty and significance.

**Reviewer Concerns:**

The concerns can be coarsely categorized into two groups:

1. The mild concerns about the technical details and the task settings.

2. The critical concerns about the work's novelty and significance (provided by Reviewer n4Mh)

The authors made efforts to resolve the concerns, including providing more explanations and experimental results.
I think the most concerns in the first category should have been resolved.
To my knowledge, even at this stage, re-docking is not fully resolved, and its technical approaches remain diverse. Therefore, personally, I think the second kind of concern is a little bit too harsh.

**Reviewer Scores:**

I believe the reviewers should have maintained their scores if they had participated fully in the discussion.

---

### Decision · Program_Chairs · 2026-01-26

Accept (Poster)